# ANALYZING AND IMPROVING THE OPTIMIZATION LANDSCAPE OF NOISE-CONTRASTIVE ESTIMATION

**Bingbin Liu, Elan Rosenfeld, Pradeep Ravikumar, Andrej Risteski**
Machine Learning Department
Carnegie Mellon University
`{bingbinl,elan,pradeepr,aristesk}@cs.cmu.edu`

## ABSTRACT

Noise-contrastive estimation (NCE) is a statistically consistent method for learning unnormalized probabilistic models. It has been empirically observed that the choice of the noise distribution is crucial for NCE's performance. However, such observations have never been made formal or quantitative. In fact, it is not even clear whether the difficulties arising from a poorly chosen noise distribution are statistical or algorithmic in nature. In this work, we formally pinpoint reasons for NCE's poor performance when an inappropriate noise distribution is used. Namely, we prove these challenges arise due to an ill-behaved (more precisely, flat) loss landscape. To address this, we introduce a variant of NCE called *eNCE* which uses an exponential loss and for which *normalized gradient descent* addresses the landscape issues *provably* when the target and noise distributions are in a given exponential family.

## 1 INTRODUCTION

Noise contrastive estimation (NCE) is a method for learning parameterized statistical models (Gutmann & Hyvärinen, 2010; 2012). To estimate a distribution $P_*$, NCE trains a discriminant model to distinguish between samples of $P_*$ and a known distribution $Q$ of our choice, often referred to as the "noise" distribution. If the function class for the discriminant model is representationally powerful enough, the optimal model learns the density ratio $p_*/q$, from which we can extract the density $p_*$ since $q$ is known (Menon & Ong, 2016; Sugiyama et al., 2012). Compared to the well-studied maximum likelihood estimation (MLE), NCE avoids calculating the (often intractable) partition function,[1] while maintaining the asymptotic consistency of MLE (Gutmann & Hyvärinen, 2012).

It is empirically well-documented that the choice of the noise distribution $Q$ is crucial to both the statistical and algorithmic efficiency of NCE (Gutmann & Hyvärinen, 2010; 2012; Rhodes et al., 2020; Goodfellow et al., 2014; Gao et al., 2020). However, it has been observed in practice that even when following the standard guidelines for choosing $Q$, NCE can still yield parameter estimates far from the ground truth (Rhodes et al., 2020; Goodfellow et al., 2014; Gao et al., 2020). Most recently, Rhodes et al. (2020) identified a phenomenon they call the "density chasm," observing empirically that NCE performs poorly when the KL divergence between $P_*$ and $Q$ is large. One example is when $P_*, Q$ are both tightly concentrated unimodal distributions with faraway modes; the region between the two modes will have a small density under both distributions, thus forming a "chasm". While it makes intuitive sense that NCE does not perform well under such settings—since disparate $Q$ and $P_*$ are easy to distinguish and do not require the model to learn much about $P_*$ in order to do well on the classification task—there has not been a theoretical analysis of this phenomenon. In fact, it is not even clear whether the difficulty is statistical or algorithmic in nature.

In this work, we formally study the challenges for NCE with a fixed $Q$ with a focus on distributions in an exponential family. We show that when the noise distribution $Q$ is poorly chosen, the loss landscape can become extremely flat: in particular, even when $P^*$ and $Q$ are two univariate Gaussian with unit variance, the loss gradient and curvature can become exponentially small in the difference in their means. We prove that this poses challenges for standard first order and even

---

[1]The partition function is also known as the normalizing constant of an unnormalized density, such that the density after normalization will integrate to 1.

second-order optimization methods, forcing them to take an exponential number of steps to converge to a good parameter estimate. Thus, standard approaches to minimizing convex functions such as gradient descent—or even more advanced techniques such as momentum or Newton's method—are not suited to the NCE objective unless $Q$ is close to $P_*$ in KL sense.

To remedy this issue, we study an alternative method for optimizing the NCE objective. We consider instead Normalized Gradient Descent (NGD) whereby the gradient is normalized to have unit norm at each time step. Perhaps surprisingly, we prove that this small modification can overcome the problem of poor curvature in the Gaussian example. In general, we show the number of steps for NGD to converge to a good solution for the NCE loss depends on the *condition number* $\kappa$ of the Hessian of the loss at the optimum—the growth of this condition number is unclear for $P^*$ and $Q$ when they belong to an exponential family.

To address this, we propose the *eNCE* loss, a variant to NCE that replaces the log loss in NCE with an exponential loss, and we show that the resulting condition number is polynomial in the dimension and the parameter distance between $P^*$ and $Q$ when they belong to an exponential family. Our proposed change of loss and optimization algorithm *together* form the first solution that provides a provable polynomial rate for learning the parameters of the ground truth distribution. Theoretically, both NCE and eNCE can potentially suffer from numerical issues during optimization when $P^*$ and $Q$ are far—this is an interesting direction for future work. Nonetheless, we find this to be a simple and effective fix to the flatness of the loss landscape in many settings, as evidenced by experimental results on synthetic and MNIST dataset.

## 1.1 RELATED WORK

NCE and its variants have inspired a large volume of research in NLP (Mnih & Teh, 2012; Mnih & Kavukcuoglu, 2013; Dyer, 2014; Kong et al., 2020) as well as computer vision (Oord et al., 2018; Hjelm et al., 2018; Henaff, 2020; Tian et al., 2020). It has been observed empirically that NCE with a fixed noise $Q$ is often insufficient for learning good generative models. The predominant class of approaches that have been proposed to overcome this issue aim to do so by not using a fixed $Q$ but by iteratively solving multiple NCE problems with an updated $Q$, or equivalently updated discriminators. This includes the famous generative adversarial network (GAN) by Goodfellow et al. (2014), which uses a separate discriminator network updated throughout training. In a similar vein, Gao et al. (2020) also aimed to increase the discriminative power as the density estimator improves, and parameterize $Q$ explicitly with a flow model. More recently, Rhodes et al. (2020) proposed the telescoping density ratio estimation, or TRE, which sidesteps the chasm by expanding $p_*/q$ into a series of intermediate density ratios, each of which is easier to estimate, leading to strong empirical performance—though their work carries no formal guarantees.

With respect to a fixed $Q$, it remains an open question about what formally are the nature of the challenges posed by a poorly chosen $Q$, which could be statistical and/or algorithmic. Various previous works have analyzed the asymptotic behavior of NCE and its variants (Gutmann & Hyvärinen, 2012; Riou-Durand et al., 2018; Uehara et al., 2020), but these do not provide guidance on the finite step convergence of NCE or its common variants. The improvements to NCE in prior works are all borne out by the empirical observations of NCE practitioners, rather than motivated by theory, which is precisely the aim of this work.

Finally, we would like to note that prior work has proposed "generalized NCE" (Pihlaja et al., 2010; Gutmann & Hirayama, 2011; Uehara et al., 2020), which relates the NCE objective to minimizing the Bregman divergence. Generalized NCE says that we can design a family of training objectives by using different convex functions to define the Bregman divergence, and the proposed eNCE is an instance of the generalized NCE objective. The difference between these prior work and ours is again the different focuses on asymptotic behavior versus finite step convergence.

## 2 PRELIMINARIES

**The NCE objective**    Let $P_*$ denote an unknown distribution in a parametric family $\{P_\theta\}_{\theta \in \Theta}$, for some bounded convex set $\Theta$, with $P_* = P_{\theta_*}$. Our goal is to estimate $P_*$ via $P_\theta$ for some $\theta \in \Theta$ by solving a noise contrastive estimation task. The noise distribution $Q$ belongs to the same parametric family with parameters $\theta_q \in \Theta$, so that $Q = P_{\theta_q}$. We use $p_\theta, p_*, q$ to denote the probability density

functions (pdfs) of $P_\theta$, $P_*$, and $Q$; we may omit $\theta$ in $P_\theta$, $p_\theta$ when it is clear from the context and write $P, p$ instead. Given $P_*$ and $Q$, the NCE loss of $P$ is defined as follows:

**Definition 2.1** (NCE Loss). *The NCE loss of $P_\theta$ w.r.t. data distribution $P_*$ and noise $Q$ is:*

$$L(P_\theta) = -\frac{1}{2}\mathbb{E}_{P_*} \log \frac{p_\theta}{p_\theta + q} - \frac{1}{2}\mathbb{E}_Q \log \frac{q}{p_\theta + q}. \tag{2.1}$$

The NCE loss can be interpreted as the binary cross-entropy loss for the classification task of distinguishing the data samples from the noise samples. Moreover, the NCE loss has a unique minimizer:

**Lemma 2.1** (Gutmann & Hyvärinen 2012). *The NCE objective in Definition 2.1 is uniquely minimized at $P = P_*$, provided that the support of $Q$ covers that of $P_*$.*

**Exponential family.** We focus our attention on the exponential family, where the pdf for a distribution with parameter $\theta$ is $p_\theta(x) = \exp\left(\theta^\top \tilde{T}(x) - A(\theta)\right)$, with $\tilde{T}(x)$ denoting the sufficient statistics and $A(\theta)$ the log partition function. [2] The partition function is treated as a parameter in NCE, so we use $\tau$ to denote the extended parameter, i.e. $\tau := [\theta, \alpha]$ where $\alpha$ is the estimate for the log partition function. We accordingly extend the sufficient statistics as $T(x) = [\tilde{T}(x), -1]$ to account for the log partition function. The pdf with the extended representation is now simply $p_\tau(x) = \exp(\tau^\top T(x))$. We will use the notation $P_\theta$ and $P_\tau$ interchangeably. We will also use $\tau(\theta)$ to denote the log-partition extended parameterization when the log partition function $\alpha$ properly normalizes the distribution specified by $\theta$.

A compelling reason for focusing on the exponential family is the observation that the NCE loss is convex in the parameter $\tau$:

**Lemma 2.2** (NCE convexity). *For exponential family $p_{\theta,\alpha}(x) = h(x)\exp(\theta^\top \tilde{T}(x) - \alpha)$, the NCE loss is convex in parameter $\tau := [\theta, \alpha]$.*

Lemma 2.2 has been stated under more general settings by Uehara et al. (2020); an alternative self-contained proof is included in Appendix A for completeness.

Recall that $\Theta$ denotes the set of parameters without the extended coordinate for the log partition function. We assume the following on distributions supported on $\Theta$:

**Assumption 2.1** (Bounded parameter norm). $\|\theta\|_2 \le \omega$, $\forall \theta \in \Theta$.

**Assumption 2.2** (Lipschitz log partition function). *Assume the log partition function is $\beta_Z$-Lipschitz, that is, $\forall \theta_1, \theta_2 \in \Theta$, $|\log Z(\theta_1) - \log Z(\theta_2)| \le \beta_Z \|\theta_1 - \theta_2\|$.*

**Assumption 2.3** (Bounded singular values of the population Fisher matrix). *There exist $\lambda_{\max}, \lambda_{\min} > 0$, such that $\forall \theta \in \Theta$, we have $\sigma_{\max}(\mathbb{E}_\theta[T(x)T(x)^\top]) \le \lambda_{\max}$, and $\sigma_{\min}(\mathbb{E}_\theta[T(x)T(x)^\top]) \ge \lambda_{\min}$.*

**Assumption 2.4** (Smooth change in the Fisher matrix). *Assume the maximum and minimum singular values of the Fisher matrix change smoothly. Namely, there exist constants $\gamma_{\max}, \gamma_{\min} > 0$ s.t.*

$$\|\nabla_\theta \sigma_{\max}(\mathbb{E}_\theta[T(x)T(x)^\top])\| \le \gamma_{\max}, \quad \|\nabla_\theta \sigma_{\min}(\mathbb{E}_\theta[T(x)T(x)^\top])\| \le \gamma_{\min}.$$

Assumptions 2.2-2.4 can be viewed as smoothness assumptions on the first, second and third order derivatives of the log partition function, and can be viewed as introducing *structural parameters* of the distributions. For example, distributions with flatter tails will have a larger $\lambda_{\max}$, which then translates to a slower rate in the results; distributions closer to being singular will have a smaller $\lambda_{\min}$, etc. In particular, Assumption 2.3 says the singular values of the Fisher matrix $\mathbb{E}_\theta[T(x)T(x)^\top]$ should be bounded from above and below. It can be shown that the Fisher matrix is proportional to the Hessian of the NCE objective when using $Q = P_*$, which means Assumption 2.3 can be interpreted as saying the NCE task can be solved efficiently under the optimal choice of $Q$.

## 3 OVERVIEW OF RESULTS

We first provide an informal overview of our results, focusing on learning of exponential families.

---

[2] Another common format of the exponential family PDF is $p_\theta(x) = h(x)\exp\left(\theta^\top T(x) - A(\theta)\right)$ where $h(x)$ is a non-negative function. Such $h(x)$ could be absorbed into $\tilde{T}(x)$ and $\theta$ with corresponding coordinates $\log(h(x))$ and 1.

**Flatness of population landscape:** Our first contribution is a negative result identifying a key source of difficulty for NCE optimization to be an ill-behaved *population* landscape. We show that due to an extremely flat landscape, gradient descent or Newton's method with *standard choices* of step sizes will need to take an exponential number of steps to find a reasonable parameter estimate.

We emphasize that though Gaussian mean estimation is a trivial task, its simplicity *strengthens the results above*: we are proving a *negative* result so that failures with a simpler setup means a stronger result. Moreover, the results only apply to *standard* choices of step sizes, such as inversely proportional to the smoothness for gradient descent, or to the ratio between the smoothness and strong convexity for Newton's method. This does not rule out the possibility that a cleverly designed learning rate schedule or a different algorithm would work efficiently; the results are however still meaningful since gradient descent with standard step sizes is the most common choice in practice.

**Overcoming flatness using normalized gradient descent:** Our second contribution is to show that the flatness problem can be solved by a simple modification to gradient descent if the loss is well-conditioned. Specifically, we show that the convergence rate for *normalized gradient descent* is polynomial in the parameter distance and $\kappa_*$, the *condition number* of the Hessian at the optimum. One immediate consequence is that in Gaussian mean estimation, for a target error of $\delta \in (0, \frac{1}{\beta_Z}]$ in parameter distance, NCE optimized with NGD achieves a rate of $O(\frac{1}{\delta^2})$, which is the same as the optimal rate achieved by MLE.

The remaining question is then whether $\kappa_*$ is polynomial in the parameters of interests. We show that $\kappa_*$ can be related to the Bhattacharyya coefficient between $P_*$ and $Q$, which indeed grows polynomially in parameter distance under certain assumptions as detailed in Section 5.2.

**Polynomial condition number for the eNCE loss:** Our third and final contribution is that if we modify the NCE objective slightly—namely, use the exponential loss in place of the log loss—then the condition number at the optimum is guaranteed to be polynomial. We call this new objective *eNCE* . Combined with the NGD result, we get that running NGD on the eNCE objective achieves a polynomial convergence guarantee.

We then provide empirical evidence on synthetic and MNIST dataset that eNCE with NGD performs comparatively with NGD on the original NCE loss, and both outperform gradient descent.

## 4 FLATNESS OF THE NCE LOSS

In this section, we study the challenges posed to NCE when using a badly chosen fixed $Q$. The main thrust of the results is to show that both algorithmic and statistical challenges can arise because the NCE loss is *poorly behaved*, particularly for first- and second-order optimization algorithms: when $P_*, Q$ are far, the loss landscape is extremely flat near the optimum. In particular, the gradient has exponentially small norm and the strong convexity constant decreases exponentially fast, limiting the convergence rate of the excess risk. We further show that when moving from $P = Q$ to $P = P_*$, the loss drops from $\Theta(1)$ to a value that is exponentially small in terms of the distance between $P_*$ and $Q$. Consequently, common gradient-based and second order methods will take exponential number of steps to converge.

An important note is that our analysis is at the population level, implying that the hardness comes from the landscape itself regardless of the statistical estimators used.

**Setup – Gaussian mean estimation**: For the negative results in this section, let's consider an exceedingly simple scenario of 1-dimensional, fixed-variance Gaussian mean estimation. We will demonstrate the enormous difficulty of achieving a good parameter estimate, even for such a simple problem—this bodes ill for NCE objectives corresponding to more complex models in practice, which certainly pose a much more difficult challenge. In particular, let $P_*, Q, P$ be Gaussians with identity variance. Let $\theta_*, \theta_q, \theta$ denote the respective means, with $\theta_*$ being the target mean that NCE aims to estimate. When the covariance is known to be 1, we can denote $h(x) := \exp(-\frac{x^2}{2})$, and parametrize the pdf of a 1d Gaussian with mean $\theta$ as $p(x) = h(x)\exp(\langle \tau(\theta), T(x) \rangle),$[3] where the parameter is $\tau(\theta) := [\theta, \frac{\theta^2}{2} + \log\sqrt{2\pi}]$ and the sufficient statistics are $T(x) := [x, -1]$. [4] We will shorthand $\tau(\theta)$ when it is clear from the context.

---

[3] Thus, we are setting $h$ to be the base measure for the exponential family we are considering.

[4] Recall that the last coordinate $-1$ acts as a sufficient statistic for the log partition function.

*Without loss of generality, we will assume $\theta_q = 0$, and $\theta_* > 0$, and denote $R := \theta_* - \theta_q$. We will write $\tau_* := \tau(\theta_*) = [R, \frac{R^2}{2} + \log\sqrt{2\pi}]$, and $\tau_q := \tau(\theta_q) = [0, \log\sqrt{2\pi}]$. As a clarification, the results stated in this section will be in terms of $R$, hence the asymptotic notations $\Omega, O$ never hide dominating dependency on $R$.* [5]

## 4.1 Properties of the NCE loss

We first describe several properties of the NCE loss that will be useful in the analysis of first- and second-order algorithms.

To start, we show that the dynamic range of the loss is large: that is, the optimal NCE loss is exponentially small as a function of $R$; on the other hand, if $\theta$ is initialized close to $\theta_q$, the initial loss would be on the order of a constant. Precisely:

**Proposition 4.1** (Range of NCE loss). *Consider the 1d Gaussian mean estimation task with mean $\theta_*, \theta_q \in \mathbb{R}$, and a known variance of 1. Denote $R := |\theta_q - \theta_*|$ where $R \gg 1$, Then, the loss at $\theta = \theta_q$ is $\log 2$, while the minimal loss $L_*$ is $L_*(R) = c\exp(-R^2/8)$ for some $c \in [\frac{1}{2}, 2]$.*

The next shows we *need to* decrease the loss to be on an order comparable to the optimum value. Namely, the loss is very flat close to $\theta_*$, thus in order to recover a $\theta$ close to $\theta_*$, we have to reach a very small value for the loss. Precisely:

**Proposition 4.2.** *Under the same setup as Proposition 4.1, for a given $\delta \in (0,1)$, if the learned parameter $\tau$ satisfies $\|\tau - \tau^*\|_2 \le \delta$, then $L(\tau) - L(\tau^*) \le R\exp(-R^2/8)\,\delta^2$.*

The way we will leverage Propositions 4.1 and 4.2 to prove lower bounds is to say that if the updates of an iterative algorithm are too small, the convergence will take an exponential number of steps.

Proposition 4.2 is proven via the Taylor expansion at $\theta^*$: since the gradient is 0 at $\theta_*$, we just need to bound the Hessian at $\theta_*$. We show:

**Lemma 4.1** (Smoothness at $P = P^*$). *Under the same setup as Proposition 4.1, the smoothness at $P = P_*$ is upper bounded as $\sigma_{\max}(\nabla^2 L(\tau_*)) \le \frac{R}{\sqrt{2\pi}}\exp(-R^2/8)$.*

We will also need a bound on the strong convexity constant (i.e. smallest singular value) at $P = P^*$:

**Lemma 4.2** (Strong convexity at $P = P^*$). *Under the same setup as Proposition 4.1, the minimum singular value at $P = P_*$ is $\sigma_{\min}^*(\nabla^2 L(\tau_*)) = \Theta\left(\frac{1}{R}\exp\left(-\frac{R^2}{8}\right)\right)$.*

Finally, in order to estimate the choice of the step size for standard optimization methods, we will also need a bound of the smoothness at $P = Q$:

**Lemma 4.3** (Smoothness at $P = Q$). *Under the same setup as Proposition 4.1, the smoothness at $P = Q$ is lower bounded as $\sigma_{\max}(\nabla^2 L(\tau_q)) \ge \frac{R^2}{2}$.*

Lemma 4.1, 4.2 are proved in Appendix D.4, and Lemma 4.3 is proved in Appendix D.5.

## 4.2 Lower bounds on first- and second-order methods

With the landscape properties at hand, we are now ready to provide lower bounds for both first-order and second-order methods. For first-order methods, we show that:

**Theorem 4.1** (Lower bound for gradient-based methods). *Let $P_*, Q, P$ be 1d Gaussian with variance 1. Assume $\theta_q = 0, \theta_* > 0$ without loss of generality, and assume $R := \theta_* - \theta_q \gg 1$. Then, gradient descent with any step size $\eta = o(1)$ from an initialization $\tau = \tau_q$ will need an exponential number of steps to reach some $\tau'$ that is $O(1)$ close to $\tau_*$.*

Note, the maximum step size $\eta = o(1)$ the theorem applies to is actually a loose bound: the standard setting of step size for gradient descent is $\eta \le 1/\lambda_M$ for $\lambda_M := \max_{\theta \in \Theta} \sigma_{\max}(\nabla^2 L(\tau(\theta)))$, which is $\Omega(R^2)$ by Lemma 4.3. Theorem 4.1 helps explain why NCE with a far-away $Q$ fails in practice, if we set the budget for the number of updates to be polynomial.

---

[5]For example, for $R \gg 1$, $R\exp(R^2) = O(\exp(R^2))$, but the constant in $O(1)$ will not depend on $R$.

A natural remedy to the drastically changing norms of the gradients is to use methods that can properly precondition the gradient. This motivates the use of second order methods, which adapt to the geometry of the loss and hence can potentially perform more competitively.

Unfortunately, standard second-order approaches are again of no help, and the number of steps required to converge remains exponential. Consider Newton's method with updates of the form $\eta(\nabla^2 L)^{-1}\nabla L$. At first glance, this looks like it may solve the issue of a flat gradient, since the Hessian $\nabla^2 L$ may also be exponentially small hence canceling out with the exponentially small gradient. However, the flatness of the landscape forces us to take an exponentially small step size $\eta$, resulting in the following claim:

**Theorem 4.2** (Lower bound for Newton's method). *Let $P_*, Q, P$ satisfy the same conditions as in theorem 4.1. Let $\lambda_\rho := \min_{\theta \in \Theta} \sigma_{\min}(\nabla^2 L(\tau_\theta))$, $\lambda_M := \max_{\theta \in \Theta} \sigma_{\max}(\nabla^2 L(\tau_\theta))$. Then, running the Newton's method with step size $\eta = O(\frac{\lambda_\rho}{\lambda_M})$ from an initialization $\tau = \tau_q$ will need an exponential number of steps to reach some $\tau'$ that is $O(1)$ close to $\tau_*$.*

Again, the condition $\eta = O\left(\frac{\lambda_\rho}{\lambda_M}\right)$ follows the typical step size choice for Newton's method, i.e. the step size should be upper bounded by the ratio between the *global* strong convexity constant and the *global* smoothness of the function, which is exponentially small for this setup by Lemma 4.2, 4.3.

## 5 Normalized gradient descent for well-conditioned losses

We have seen that due to an ill-behaved landscape, NCE optimized with standard gradient descent or Newton's method will fail to reach a good parameter estimate efficiently, even on a problem as simple as Gaussian mean estimation, and even with access to the population gradient.

In this section, we will show that a close relative of gradient descent, *normalized gradient descent* (NGD), despite its simplicity, provides a fix to the flatness problem to exponential family distributions when the Hessian of the loss is well-conditioned close to the optimum.

Precisely, recall that the NGD updates for a loss function $L$ is $\tau_{t+1} = \tau_t - \eta \frac{\nabla L(\tau_t)}{\|\nabla L(\tau_t)\|_2}$. We assume that in a neighborhood around $\tau_*$, the change in the shape of the Hessian $\boldsymbol{H}$ is moderate: [6]

**Assumption 5.1** (Hessian in a neighborhood of $\tau_*$). *Under assumption 2.2 with constant $\beta_Z$, assume that for any $\tau$ such that $\|\tau - \tau_*\|_2 \leq \frac{1}{\beta_Z}$, it holds that $\sigma_{\max}(\boldsymbol{H}(\tau)) \leq \beta_u \cdot \sigma_{\max}(\boldsymbol{H}(\tau_*))$, and $\sigma_{\min}(\boldsymbol{H}(\tau)) \geq \beta_l \cdot \sigma_{\min}(\boldsymbol{H}(\tau_*))$, for some constant $\beta_u, \beta_l > 0$.*

The main result of this section states that NGD can find a parameter estimate efficiently for exponential families, where the number of steps required is polynomial in the distance between the initial estimate and the optimum:

**Theorem 5.1.** *Let $L$ be any loss function that is convex in the exponential family parameter and satisfies Assumptions 5.1 and 2.1 - 2.4. Furthermore, let $P_*, Q$ be exponential family distributions with parameters $\tau_*, \tau_q$ and let $\kappa_*$ be the condition number of the Hessian at $P = P_*$. Then, for any $0 < \delta \leq \frac{1}{\beta_Z}$ and parameter initialization $\tau_0$, with step size $\eta \leq \sqrt{\frac{\beta_l}{\beta_u \kappa_*}}\delta$, performing NGD on the population objective $L$ guarantees that after $T \leq \frac{\beta_u \kappa_*}{\beta_l} \cdot \frac{\|\tau_0 - \tau_*\|^2}{\delta^2}$ steps, there exists an iterate $t \leq T$ such that $\|\tau_t - \tau_*\|_2 \leq \delta$.*

The main technical ingredient for proving Theorem 5.1 is the following Lemma:

**Lemma 5.1.** *Suppose Assumptions 2.2 and 5.1 hold with constants $\beta_Z$, $\beta_u$ and $\beta_l$. Let $L$ be a convex function with minimizer $\tau_*$, and let $g := \nabla L(\tau)$. For any $\delta \leq \frac{1}{\beta_Z}$, let $\gamma = \sqrt{\frac{\beta_l}{\beta_u \kappa_*}}\delta$. Then for all $\tau$ s.t. $\|\tau - \tau_*\|_2 \geq \delta$, we have $L(\tau_* + \gamma \frac{g}{\|g\|}) \leq L(\tau)$.*

Lemma 5.1 explains the dependency on $\kappa_*$ in the NGD convergence rate. The intuition of the proof is that in a small neighborhood around $\tau_*$, the set of parameters that have the same loss form an "ellipsoid", and by Taylor expansion, any two points in the same set will have a distance-to-$\tau_*$ ratio upper bounded roughly by $\sqrt{\kappa_*}$. The details are in Appendix C.2.

---

[6] As a concrete example, we will show in the next section that a variant of NCE satisfies both conditions.

*Proof sketch for Theorem 5.1*: the proof leverages two observations: the convexity of NCE loss on exponential family parameters, and that the Hessian in a neighborhood around $\tau_*$ changes moderately as stated in Assumption 5.1. One can then show there exists a global constant $\gamma$ such that for any $\tau_t$ satisfying $\|\tau_t - \tau_*\|_2 \geq \delta$, one step of NGD update guarantees a decrease of $\gamma^2$ in the squared error, which means NGD must have found an $\delta$-close estimate within $\frac{\|\tau_0 - \tau_*\|^2}{\gamma^2}$ steps. The full proof is deferred to Appendix C.1.

## 5.1 EXAMPLE: 1D GAUSSIAN MEAN ESTIMATION

It is relatively straightforward to check that NGD addresses the flatness problem faced by Gaussian mean estimation we considered in Section 4:

**Corollary 5.1.** *Let $P_*, Q$ be 1d Gaussian with covariance 1 and mean $\theta_* = R$ where $R \ll 1$, and $\theta_q = 0$. For any given $\delta \leq \frac{1}{R}$ and initial estimate $\tau_0 = \tau_q$, NGD can find an estimate $\tau$ such that $\|\tau - \tau_*\|_2 \leq \delta$, with at most $O(\frac{R^6}{\delta^2})$ steps.*

Intuitively, the effectiveness of NGD comes from the crucial observation that though the magnitude for the loss and derivatives can be exponentially small, they share the same exponential factor, making normalization effective. Formally, it can be shown that $\frac{\beta_u}{\beta_l} = O(1)$ (Appendix D.6). Corollary 5.1 then follows from Theorem 5.1 and the curvature and strong convexity from Lemma 4.1, 4.2.

## 5.2 BOUNDS ON THE CONDITION NUMBER OF NCE

The convergence rate in Theorem 5.1 depends on $\kappa_*$, the condition number of the NCE Hessian at the optimum, and Hessian-related constants $\beta_u, \beta_l$ in Assumption 5.1. We now show that under the setup of Theorem 5.1, $\kappa_*$ and $\beta_u, \beta_l$ can be related to the *Bhattacharyya coefficient* between $P_*$ and $Q$, which is a similarity measure defined as $\mathrm{BC}(P_*, Q) := \int_x \sqrt{p_*(x)q(x)} dx$. As a result, we get the following convergence guarantee:

**Theorem 5.2.** *Suppose Assumptions 2.1, 2.3 hold with constants $\omega$, $\lambda_{\max}$, and $\lambda_{\min}$. Consider a NCE task with data distribution $P_1$ and noise distribution $P_2$, parameterized by $\theta_1, \theta_2 \in \Theta$ respectively. Then for any given $\delta \leq \frac{1}{R}$ and initial estimate $\tau_0 = \tau_q$, NGD finds an estimate $\tau$ such that $\|\tau - \tau_*\|_2 \leq \delta$ within $T \leq C \cdot \frac{1}{\mathrm{BC}(P_*, Q)^3} \frac{\|\tau_0 - \tau_*\|^2}{\delta^2}$ steps, where $C := 18 \exp\left(\frac{2}{\beta_Z}\right) \cdot \left(\frac{\lambda_{\max}}{\lambda_{\min}}\right)^3 \cdot \min\left\{\frac{2\lambda_{\max}^2}{\lambda_{\min}^2}, \frac{2\lambda_{\min} + \gamma_{\max}\|\bar{\delta}\|}{\lambda_{\min} - \gamma_{\min}\|\bar{\delta}\|}\right\}$.*

In particular, when $P_*, Q$ are not too far, we can further show a lower bound on $\mathrm{BC}(P_*, Q)$:

**Lemma 5.2.** *For $P_1, P_2$ parameterized by $\theta_1, \theta_2 \in \Theta$, if $\|\theta_1 - \theta_2\|_2^2 \leq \frac{4}{\lambda_{\max}}$, then $\mathrm{BC}(P_1, P_2) \geq \frac{1}{2}$.*

The proofs of Theorem 5.2 and Lemma 5.2 rely on analyzing the geodesic on the manifold of square root densities $\sqrt{p}$ equipped with the Hellinger distance as a metric; the details are deferred to Appendix C.3 and C.4. It is also worth noting that Theorem 5.2 only requires $\|\theta_1 - \theta_2\|$ to be smaller than a constant, rather than tending to zero as usually required for analyses using Taylor expansions.

Finally, we would like to note that although our analysis can be tightened, it is unlikely to remove such dependency since NGD only uses first-order information. [7] Moreover, the condition number $\kappa_*$ also affects the practical use of Newton-like methods, since matrix inversion is widely known to be sensitive to numerical issues when the matrix is extremely ill-conditioned. It is an interesting open question whether a non-standard preconditioning approach might be amenable to this setting.

## 6 ANALYZING ENCE : NCE WITH AN EXPONENTIAL LOSS

The previous section proved that NGD can serve as a simple fix to overcome the flatness problem of NCE for well-conditioned losses. However, though we showed $\kappa_*$ has a polynomial growth when the distributions $P, Q^*$ are sufficiently close —it is unclear how $\kappa_*$ behaves beyond this threshold.

---

[7]In the next section, we will that the condition number is provably polynomial in $\|\theta_* - \theta_q\|$ for a variant of the NCE loss.

In this section, we introduce a slight modification to the NCE objective, which we call the *eNCE* objective, for which $\kappa_*$ depends *polynomially* on some class-related constants. This means though eNCE may still suffer from the flatness problem, *eNCE and NGD together provide a solution that guarantees a polynomial convergence rate*.

Towards formalizing this, the eNCE loss is defined as:

**Definition 6.1** (eNCE Loss). *Let* $\varphi(x) := \log \sqrt{\frac{p(x)}{q(x)}}$, *and* $l(x, y) := \exp(-y\varphi(x))$ *for* $y \in \{\pm 1\}$. *The eNCE loss of* $P_\theta$ *w.r.t. data distribution* $P_*$ *and noise* $Q$ *is:*

$$L_{\exp}(P_\theta) = \frac{1}{2}\mathbb{E}_{x \sim P_*}[l(x, 1)] + \frac{1}{2}\mathbb{E}_{x \sim P_*}[l(x, -1)] = \frac{1}{2}\int_x p_* \sqrt{\frac{q(x)}{p(x)}} + \frac{1}{2}\int_x q\sqrt{\frac{p(x)}{q(x)}}. \quad (6.1)$$

It can be checked easily that the minimizing $\varphi$ learns $\varphi(x) = \frac{1}{2}\log\frac{p_*}{q}$. Moreover, each $\varphi$ is associated with an induced distribution $p$, defined by $p(x) = \exp(\varphi(x))q(x)$.

*Relation to NCE*: Same as the original NCE loss (referred to as "NCE" below), eNCE learns to solve a distinguishing task between samples from $P_*$ or $Q$. The difference lies only in the losses, which have analogous forms: the NCE loss described in Def. 2.1 can be rewritten in the same form with $l(x, y) := \log \frac{1}{1+\exp(-y\psi(x))}$ and $\psi(x) := \log\frac{p(x)}{q(x)}$.

The main advantage of the exponential loss is that the Hessian at the optimum is now guaranteed to be well-conditioned:

**Lemma 6.1** (Polynomial condition number for eNCE loss). *Under Assumption 2.3 with constants* $\lambda_{\max}, \lambda_{\min}$, *the condition number of the eNCE Hessian at the optimum is bounded by* $\kappa_* \leq \frac{\lambda_{\max}}{\lambda_{\min}}$.

We can also show that eNCE satisfies part (ii) of Assumption 5.1. Due to space considerations, the proof is deferred to Appendix B.1.

**Lemma 6.2.** *Under assumption 2.2 with constant* $\beta_Z$, *for any unit vector* $\boldsymbol{u}$ *and constant* $c \in [0, \frac{1}{\beta_Z}]$, *the maximum and minimum singular values of* $\boldsymbol{H}(\tau_* + c\boldsymbol{u})$ *satisfy assumption 5.1 with constants* $\beta_u = 2\exp(1) \cdot \frac{\lambda_{\max}}{\lambda_{\min}}$, $\beta_l = \frac{1}{2\exp(1)} \cdot \frac{\lambda_{\min}}{\lambda_{\max}}$.

Lemma 6.1 and Lemma 6.2 together imply the Hessian is well-conditioned around the optimum. Combined with Theorem 5.1, we have the main result of this section:

**Theorem 6.1.** *Let* $P_*, Q$ *be exponential family distributions with parameters* $\tau_*, \tau_q$ *under Assumption 2.1-2.4. Let* $\lambda_{\max}, \lambda_{\min}$ *be constants for Assumption 2.3. For any given* $\delta \leq \frac{1}{\beta_Z}$ *and parameter initialization* $\tau_0$, *performing NGD on the eNCE objective guarantees that when taking* $T \leq 4\exp(2) \cdot \frac{\lambda_{\max}^3}{\lambda_{\min}^3} \cdot \frac{\|\tau_0 - \tau_*\|^2}{\delta^2}$ *steps, there exists an iterate* $t \leq T$ *such that* $\|\tau_t - \tau_*\|_2 \leq \delta$.

*Proof.* Theorem 6.1 follows directly from Theorem 5.1, using the condition number bound from Lemma 6.1 and constants from 6.2. □

## 6.1 PROOF OF LEMMA 6.1

It can be checked that the Hessian at $P = P_*$ is $\boldsymbol{H}_* := \nabla^2 L(P_*) = \frac{1}{4}\int_x \sqrt{p_*q}\nabla\log p(\nabla\log p)^\top$. Recall that $\theta_*, \theta_q, \tilde{T}$ denote the parameters and sufficient statistics without the partition function coordinate, and $\tau_*, \tau_q, T$ denote the extended version with the partition function, e.g. $\tau_* = [\theta_*, \log Z(\theta_*)]$, $T(x) = [\tilde{T}(x), -1]$. Then, we can rewrite $\boldsymbol{H}_*$ as:

$$\begin{aligned}
\boldsymbol{H}_* &= \frac{1}{4}\int_x \sqrt{p_*q}T(x)T(x)^\top = \frac{1}{4}\int_x \exp\left(\frac{(\tau_* + \tau_q)^\top}{2}T(x)\right)T(x)T(x)^\top \\
&= \frac{1}{4}\int_x \exp\left(\frac{(\theta_* + \theta_q)^\top}{2}\tilde{T}(x) - \frac{1}{2}\log Z(\theta_*) - \frac{1}{2}\log Z(\theta_q)\right)T(x)T(x)^\top \\
&= \frac{1}{4}\underbrace{\frac{Z\left(\frac{\theta_* + \theta_q}{2}\right)}{\sqrt{Z(\theta_*)Z(\theta_q)}}}_{B(P_*, Q)}\int_x \frac{\exp\left(\left(\frac{\theta_* + \theta_q}{2}\right)^\top\tilde{T}(x)\right)}{Z\left(\frac{\theta_* + \theta_q}{2}\right)}T(x)T(x)^\top dx = \frac{B(P_*, Q)}{4}\mathbb{E}_{\frac{\theta_* + \theta_q}{2}}[TT^\top].
\end{aligned} \quad (6.2)$$

The Lemma then follows from $\lambda_{\min}\boldsymbol{I} \preceq \mathbb{E}_{\frac{\theta_* + \theta_q}{2}}[TT^\top] \preceq \lambda_{\max}\boldsymbol{I}$ by Assumption 2.3.

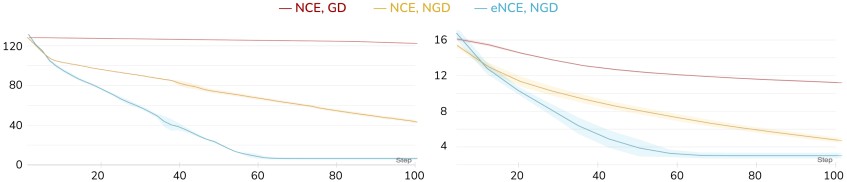

Figure 1: Results for estimating 1d (left) and 16d (right) Gaussians, plotting $\min_{t\in[T]}\|\tau_* - \tau_t\|_2$ ($y$-axis) against the number of updates $T$ ($x$-axis). In both cases, when using NCE, normalized gradient descent ("NCE, NGD", yellow) largely outperforms gradient descent ("NCE, GD", red). When using NGD, the proposed eNCE ("eNCE, NGD", blue) decays faster than the original NCE loss. The results are averaged over 5 runs, with shaded areas showing the standard deviation.

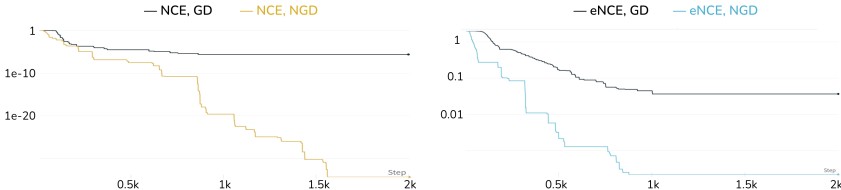

Figure 2: Results on MNIST, plotting loss value ($y$-axis, log scale) against update steps ($x$-axis). The left plot shows NCE optimized by GD (black) and NGD (yellow), and the right shows eNCE optimized by GD (black) and NGD (blue). It can be seen that NGD outperforms GD in both cases.

## 7    EMPIRICAL VERIFICATION

To corroborate our theory, we verify the effectiveness of NGD and eNCE on Gaussian mean estimation and the MNIST dataset. For MNIST, we use a ResNet-18 to model the log density ratio $\log(p/q)$, following the setup in TRE (Rhodes et al., 2020).

**Results:** For Gaussian data, we run gradient descent (GD) and normalized gradient descent (NGD) on the NCE loss and eNCE loss. Figure 1 compares the best runs under each setup given a fixed computation budget (100 update steps), where "best" is defined to be the run with the lowest loss on fresh samples. The plots show the minimum parameter distance $\min_{t\in[T]}\|\tau_* - \tau_t\|_2$ for each step $T$. We find that NGD indeed outperforms GD, and that the proposed eNCE sees a further improvement over NCE while additionally enjoying provable polynomial convergence guarantees. For MNIST, we can no longer compare parameter distances since $\tau_*$ is unknown. Instead, we compare the result of optimization directly in terms of loss achieved, again under a fixed computation budget (2K steps). The results are shown in Figure 2, with NGD converging significantly faster for both NCE and eNCE. We note that eNCE can be numerically unstable, especially when $P_*, Q$ are well separated. We include implementation details in Appendix E.1 and additional results in Appendix E.2.

## 8    CONCLUSION AND DISCUSSIONS

We provided a theoretical analysis of the algorithmic difficulties that arise when optimizing the NCE objective with an uninformative noise distribution, stemming from an ill-behaved loss landscape. Our theoretical results are inspired by empirical observations in prior works (Rhodes et al., 2020; Gao et al., 2020; Goodfellow et al., 2014) and provide the first formal explanation on the nature of the optimization problems of NCE. Our negative results showed that even on the simple task of Gaussian mean estimation, and even assuming access to the population gradient, gradient descent and Newton's method with standard step size choice still require an exponential number of steps to reach a good solution.

We then proposed modifications to the NCE loss and optimization algorithm, whose combination results in the first provably polynomial convergence rate for NCE. The loss we propose, eNCE, can be efficiently optimized using normalized gradient descent and empirically outperforms existing methods. We hope these theoretical results will help identify promising new directions in the search for simple, effective, and practical improvements to noise-contrastive estimation.

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

APPENDIX

We will fist provide missing proofs for the eNCE results in section 6 in section B. Section C provides proofs for NGD convergence on the NCE loss, and section D proves the negative results of NCE in section 4). Additional notes on the experiments are provided in section E.1.

**Notation**: We will use $a \lesssim b$ to denote $a = O(b)$ with $O$ hiding a constant less than 2. Similarly, $a \gtrsim b$ denotes $a = \Omega(b)$ where $\Omega$ hides a constant greater than $\frac{1}{2}$.

## A    PROOF OF CONVEXITY OF NCE (LEMMA 2.2)

As a preliminary, let's first prove that the NCE loss is convex in exponential family parameters. Recall that the NCE loss is

$$L(P) := \frac{1}{2}\mathbb{E}_{P_*} \log \frac{p+q}{p} + \frac{1}{2}\mathbb{E}_Q \log \frac{p+q}{q} \tag{A.1}$$

where $p(x) = p(\tau^\top T(x))$. The gradient and Hessian of the NCE loss are:

$$
\begin{aligned}
\nabla_\tau p(x) =& p(x) \cdot T(x), \\
\nabla L(\tau) =& \frac{1}{2}\nabla \left[ \mathbb{E}_* \log \frac{p+q}{p} + \mathbb{E}_Q \log \frac{p+q}{q} \right] \\
=& \frac{1}{2}\left[ \mathbb{E}_* \frac{p}{p+q}\frac{p-p-q}{p^2}\nabla_\tau p + \mathbb{E}_Q \frac{q}{p+q}\frac{1}{q}\nabla_\tau p \right] = \frac{1}{2}\int_x \frac{q}{p+q}(p-p_*)T(x)dx, \\
\nabla^2 L(\tau) =& \frac{1}{2}\int_x \left( -\frac{q(p-p_*)}{(p+q)^2}\nabla_\tau p + \frac{q}{p+q}\nabla_\tau p \right)T(x)dx \\
=& \frac{1}{2}\int_x \frac{q}{p+q} \cdot \frac{p_*+q}{p+q} \cdot p \cdot T(x)T(x)^\top dx = \frac{1}{2}\int_x \frac{(p_*+q)pq}{(p+q)^2}T(x)T(x)^\top dx.
\end{aligned} \tag{A.2}
$$

Hence the Hessian is PSD at any $\tau$.

## B    PROOFS FOR SECTION 6 (ENCE)

We first write down the loss, gradient and Hessian of eNCE   for the exponential family:

$$
\begin{aligned}
L_{\exp}(P) =& \frac{1}{2}\int_x p_*\sqrt{\frac{q}{p}} + \frac{1}{2}\int_x q\sqrt{\frac{p}{q}} \\
\nabla L_{\exp}(P) =& \frac{1}{4}\int_x \sqrt{q}\left( \sqrt{p} - \frac{p_*}{\sqrt{p}} \right)\nabla \log p \\
\nabla^2 L_{\exp}(P) =& \frac{1}{4}\int_x \sqrt{q}\left( \sqrt{p} - \frac{p_*}{\sqrt{p}} \right) \cdot \nabla^2 \log p + \frac{1}{8}\int_x \sqrt{q}\left( \sqrt{p} + \frac{p_*}{\sqrt{p}} \right)\nabla \log p(\nabla \log p)^\top \\
=& \frac{1}{8}\int_x \sqrt{q}\left( \sqrt{p} + \frac{p_*}{\sqrt{p}} \right)\nabla \log p(\nabla \log p)^\top \\
=& \frac{1}{8}\int_x p_*\sqrt{\frac{q}{p}}T(x)T(x)^\top + \frac{1}{8}\int_x q\sqrt{\frac{p}{q}}T(x)T(x)^\top.
\end{aligned} \tag{B.1}
$$

Note that the Hessian is always PSD, which means $L_{\exp}$ is convex in the parameters of the exponential family.

## B.1 PROOF OF LEMMA 6.2

We directly calculate the Hessian at some $\tilde{\tau} := \tau_* + c\boldsymbol{u}$ for some $c \leq \frac{1}{\beta_Z}$ and $\|\boldsymbol{u}\|_2 = 1$, using the expression in equation B.1:

$$
\begin{aligned}
\nabla^2 L_{\exp}(\tilde{\tau}) &= \int_x \left( p_* \sqrt{\frac{q}{\tilde{p}}} + q \sqrt{\frac{\tilde{p}}{q}} \right) T(x) T(x)^\top \\
&= \int_x \left[ \exp \left( \langle \tau_* + \frac{\tau_q - \tilde{\tau}}{2}, T(x) \rangle \right) + \exp \left( \langle \frac{\tau_q + \tilde{\tau}}{2}, T(x) \rangle \right) \right] T(x) T(x)^\top \\
&= \int_x \left[ \exp \left( \langle \frac{\tau_q + \tau_*}{2} - \frac{c}{2} \boldsymbol{u}, T(x) \rangle \right) + \exp \left( \langle \frac{\tau_q + \tau_*}{2} + \frac{c}{2} \boldsymbol{u}, T(x) \rangle \right) \right] T(x) T(x)^\top \\
&= \int_x \left[ \exp \left( \langle -\frac{c}{2} \boldsymbol{u}, T(x) \rangle \right) + \exp \left( \langle \frac{c}{2} \boldsymbol{u}, T(x) \rangle \right) \right] \exp \left( \langle \frac{\tau_q + \tau_*}{2}, T(x) \rangle \right) T(x) T(x)^\top \\
&= \underbrace{\frac{Z(\frac{\theta_* + \theta_q}{2})}{\sqrt{Z(\theta_q) Z(\theta_*)}}}_{B(P_*, Q)} \int_x \left[ \exp \left( \langle -\frac{c}{2} \boldsymbol{u}, T(x) \rangle \right) + \exp \left( \langle \frac{c}{2} \boldsymbol{u}, T(x) \rangle \right) \right] \exp \left( \langle \tau(\frac{\theta_q + \theta_*}{2}), T(x) \rangle \right) T(x) T(x)^\top.
\end{aligned}
$$
(B.2)

Note that without the term in the square brackets, the integration is exactly the same as the one for $\boldsymbol{H}_*$.

We would like to bound the ratio $\frac{\boldsymbol{v}^\top \nabla^2 L_{\exp}(\tilde{\tau}) \boldsymbol{v}}{\boldsymbol{v}^\top \boldsymbol{H}_* \boldsymbol{v}}$ for any unit vector $\boldsymbol{v}$. Denote $\bar{\delta} := \frac{c\boldsymbol{u}}{2}$, $\bar{\tau} := \tau \left( \frac{\theta_q + \theta_*}{2} \right)$ for notation convenience, and denote $\mathcal{S}_1 := \{x : \bar{\delta}^\top T(x) > 0\}$, $\mathcal{S}_{-1} := \{x : \bar{\delta}^\top T(x) \leq 0\}$. We have:

$$
\begin{aligned}
\frac{\boldsymbol{v}^\top \nabla^2 L_{\exp}(\tilde{\tau}) \boldsymbol{v}}{\boldsymbol{v}^\top \boldsymbol{H}_* \boldsymbol{v}} &\simeq \frac{\int_{x \in \mathcal{S}_1} \exp \left( \bar{\delta}^\top T(x) \right) \exp \left( \bar{\tau}^\top T(x) \right) (\boldsymbol{v}^\top T(x))^2}{\int_x \exp \left( \bar{\tau}^\top T(x) \right) (\boldsymbol{v}^\top T(x))^2} \\
&\quad + \frac{\int_{x \in \mathcal{S}_{-1}} \exp \left( \bar{\delta}^\top T(x) \right) \exp \left( \bar{\tau}^\top T(x) \right) (\boldsymbol{v}^\top T(x))^2}{\int_x \exp \left( \bar{\tau}^\top T(x) \right) (\boldsymbol{v}^\top T(x))^2} := T_1 + T_{-1}.
\end{aligned}
$$
(B.3)

Recall that $f \simeq g$ means functions $f, g$ differ only by a constant factor. This equation will be used to calculate both the upper and the lower bound.

For the upper bound, let $\chi \in \{\pm 1\}$, we have

$$
\begin{aligned}
T_\chi &= \frac{\int_{x : \chi \bar{\delta}^\top T(x) > 0} \exp \left( \chi \bar{\delta}^\top T(x) \right) \exp \left( \bar{\tau}^\top T(x) \right) (\boldsymbol{v}^\top T(x))^2}{\int_x \exp \left( \bar{\tau}^\top T(x) \right) (\boldsymbol{v}^\top T(x))^2} \\
&= \frac{Z(\chi \bar{\theta} + \frac{\theta_q + \theta_*}{2})}{Z(\frac{\theta_q + \theta_*}{2}) \cdot \exp(\chi \bar{\alpha})} \cdot \frac{\int_{x : \bar{\delta}^\top T(x) > 0} p_{\chi \bar{\theta} + \frac{\theta_* + \theta_q}{2}}(x) (\boldsymbol{v}^\top T(x))^2}{\int_x p_{\frac{\theta_* + \theta_q}{2}} (\boldsymbol{v}^\top T(x))^2} \\
&\leq \frac{Z(\chi \bar{\theta} + \frac{\theta_q + \theta_*}{2})}{Z(\frac{\theta_q + \theta_*}{2}) \cdot \exp(\chi \bar{\alpha})} \cdot \frac{\mathbb{E}_{\chi \bar{\theta} + \frac{\theta_* + \theta_q}{2}} [(\boldsymbol{v}^\top T(x))^2]}{\mathbb{E}_{\frac{\theta_* + \theta_q}{2}} [(\boldsymbol{v}^\top T(x))^2]} \\
&\overset{(i)}{\leq} \exp \left( \beta_Z \| \bar{\theta} \| - \chi \bar{\alpha} \right) \frac{\mathbb{E}_{\chi \bar{\theta} + \frac{\theta_* + \theta_q}{2}} [(\boldsymbol{v}^\top T(x))^2]}{\mathbb{E}_{\frac{\theta_* + \theta_q}{2}} [(\boldsymbol{v}^\top T(x))^2]}
\end{aligned}
$$
(B.4)

where step $(i)$ uses the Lipschitz property of the log partition function in assumption 2.2.

For the lower bound, let $\chi^* := \arg\max_{\chi \in \{\pm 1\}} T_\chi$. Write $\bar{\delta} = [\bar{\theta}, \bar{\alpha}]$ (i.e. separating out $\bar{\alpha}$ which is the normalizing constant), let $\mathcal{S}_{\frac{1}{2}}(\boldsymbol{v}) \subset \mathcal{S}_{\chi^*}$ denote a set s.t.

$$
\int_{x \in \mathcal{S}_{\frac{1}{2}}(\boldsymbol{v})} p_{\chi^* \bar{\theta} + \frac{\theta_* + \theta_q}{2}}(x) (\boldsymbol{v}^\top T(x))^2 \geq \frac{1}{2} \int_x p_{\chi^* \bar{\theta} + \frac{\theta_* + \theta_q}{2}}(x) (\boldsymbol{v}^\top T(x))^2.
$$

Then $T_\chi$ for $\chi \in \{\pm 1\}$ can be lower bounded as:

$$
\begin{aligned}
T_{\chi^*} &\geq \frac{Z(\chi\bar{\theta} + \frac{\theta_q+\theta_*}{2})}{Z(\frac{\theta_q+\theta_*}{2}) \cdot \exp(\bar{\alpha})} \cdot \frac{\int_{x \in \mathcal{S}_{\frac{1}{2}}(\boldsymbol{v})} p_{\chi^*\bar{\theta}+\frac{\theta_*+\theta_q}{2}}(x)(\boldsymbol{v}^\top T(x))^2}{\int_x p_{\frac{\theta_*+\theta_q}{2}}(\boldsymbol{v}^\top T(x))^2} \\
&\geq \frac{1}{2} \frac{Z(\chi^*\bar{\theta} + \frac{\theta_q+\theta_*}{2})}{Z(\frac{\theta_q+\theta_*}{2}) \cdot \exp(\bar{\alpha})} \cdot \frac{\mathbb{E}_{\chi^*\bar{\theta}+\frac{\theta_*+\theta_q}{2}}[(\boldsymbol{v}^\top T(x))^2]}{\mathbb{E}_{\frac{\theta_*+\theta_q}{2}}[(\boldsymbol{v}^\top T(x))^2]} \\
&\overset{(i)}{\geq} \frac{1}{2} \exp\left(-\beta_Z\|\bar{\theta}\| - \chi^*\bar{\alpha}\right) \cdot \frac{\mathbb{E}_{\chi^*\bar{\theta}+\frac{\theta_*+\theta_q}{2}}[(\boldsymbol{v}^\top T(x))^2]}{\mathbb{E}_{\frac{\theta_*+\theta_q}{2}}[(\boldsymbol{v}^\top T(x))^2]}
\end{aligned}
\tag{B.5}
$$

$$
T_{-\chi^*} \geq 0
$$

where step $(i)$ uses the Lipschitz property of the log partition function in assumption 2.2.

This means for any unit vector $\boldsymbol{v}$, we have

$$
\begin{aligned}
\frac{\boldsymbol{v}^\top \nabla^2 L_{\exp}(\tilde{\tau})\boldsymbol{v}}{\boldsymbol{v}^\top \boldsymbol{H}_*\boldsymbol{v}} &= T_1 + T_{-1} \\
&\leq \exp\left(\beta_Z\|\bar{\theta}\| + |\bar{\alpha}|\right) \cdot \left[\frac{\mathbb{E}_{\bar{\theta}+\frac{\theta_*+\theta_q}{2}}[(\boldsymbol{v}^\top T(x))^2]}{\mathbb{E}_{\frac{\theta_*+\theta_q}{2}}[(\boldsymbol{v}^\top T(x))^2]} + \frac{\mathbb{E}_{-\bar{\theta}+\frac{\theta_*+\theta_q}{2}}[(\boldsymbol{v}^\top T(x))^2]}{\mathbb{E}_{\frac{\theta_*+\theta_q}{2}}[(\boldsymbol{v}^\top T(x))^2]}\right] \\
&\overset{(i)}{\leq} 2 \exp\left(\beta_Z\|\bar{\theta}\| + |\bar{\alpha}|\right) \cdot \frac{\lambda_{\max}}{\lambda_{\min}} \leq 2\exp\left(\frac{c}{2}(1 + \beta_Z)\right) \cdot \frac{\lambda_{\max}}{\lambda_{\min}} \leq 2\exp(1) \cdot \frac{\lambda_{\max}}{\lambda_{\min}} \\
\frac{\boldsymbol{v}^\top \nabla^2 L_{\exp}(\tilde{\tau})\boldsymbol{v}}{\boldsymbol{v}^\top \boldsymbol{H}_*\boldsymbol{v}} &= T_1 + T_{-1} \overset{(ii)}{\geq} \frac{1}{2}\exp\left(-\beta_Z\|\bar{\theta}\| - |\bar{\alpha}|\right) \cdot \frac{\lambda_{\min}}{\lambda_{\max}} \geq \frac{1}{2\exp(1)} \cdot \frac{\lambda_{\min}}{\lambda_{\max}}
\end{aligned}
\tag{B.6}
$$

where step $(i), (ii)$ follow from assumption 2.3.

Hence the eNCE loss satisfies assumption 5.1 with constants $\beta_u = 2\exp(1) \cdot \frac{\lambda_{\max}}{\lambda_{\min}}, \beta_l = \frac{1}{2\exp(1)} \cdot \frac{\lambda_{\min}}{\lambda_{\max}}$.

# C PROOFS FOR SECTION 5 (NGD AND CONDITION NUMBER AT THE OPTIMUM)

This section provides proofs for results in section 5. Results for NGD convergence rate (Theorem 5.1 and Lemma 5.1) are proved in section C.1 and C.2. Section C.3 proves the convergence rate stated in terms of the Bhattacharyya coefficient (Theorem 5.2), and the bound on Bhattacharyya coefficient (Lemma 5.2) is proved in section C.4.

## C.1 PROOF OF THEOREM 5.1 (NGD CONVERGENCE RATE)

**Theorem C.1** (Theorem 5.1 restated). *Let $L$ be any loss function that is convex in the exponential family parameter and satisfies Assumptions 5.1 and 2.1 - 2.4. Furthermore, let $P_*, Q$ be exponential family distributions with parameters $\tau_*, \tau_q$ and let $\kappa_*$ be the condition number of the Hessian at $P = P_*$. Then, for any $0 < \delta \leq \frac{1}{\beta_Z}$ and parameter initialization $\tau_0$, let the step size be $\eta \leq \sqrt{\frac{\beta_l}{\beta_u\kappa_*}}\delta$, performing NGD on the population objective $L$ guarantees that after $T \leq \frac{\beta_u\kappa_*}{\beta_l} \cdot \frac{\|\tau_0-\tau_*\|^2}{\delta^2}$ steps, there exists an iterate $t \leq T$ such that $\|\tau_t - \tau_*\|_2 \leq \delta$.*

*Proof.* Denote $g_t := \nabla L(\tau_t)$ and $R := \|\tau_* - \tau_q\|_2$ for notation convenience. Recall that the NGD update with step size $\eta$ is $\tau_{t+1} = \tau_t - \eta \cdot \frac{g_t}{\|g_t\|_2}$. Then, $\|\tau_t - \tau_*\|^2$ can be rewritten as:

$$
\|\tau_{t+1} - \tau_*\|^2 = \|\tau_t - \tau_*\|^2 - 2\gamma\eta + \eta^2 + 2\eta\frac{g_t^\top}{\|g_t\|}\left(\tau_* + \gamma\frac{g_t}{\|g_t\|} - \tau_t\right)
\tag{C.1}
$$

If we set $\gamma$ s.t. the last term is smaller than 0 for all $\tau$ that are not within distance $\delta$ to $\tau_*$, setting $\eta = \gamma$ gives:

$$\|\tau_{t+1} - \tau_*\|^2 \leq \|\tau_t - \tau_*\|^2 - 2\gamma\eta + \eta^2 = \|\tau_t - \tau_*\|^2 - \gamma^2 \tag{C.2}$$

Hence the number of steps required to find a $\tau$ s.t. $\|\tau - \tau_*\|_2 \leq \delta$ is at most $T \leq \frac{\|\tau_0 - \tau_*\|^2}{\gamma^2}$.

By Lemma 5.1, setting $\gamma = \sqrt{\frac{\beta_l}{\beta_u \kappa_*}}\delta$ ensures $L(\tau_* + \gamma\frac{g_t}{\|g_t\|}) \leq L(\tau_t)$ for any $\tau_t$ that is at least $\delta$ away from $\tau_*$. It then follows from the convexity of $L$ that

$$g_t^\top \left(\tau_* + \gamma\frac{g_t}{\|g_t\|} - \tau_t\right) \leq L\left(\tau_* + \gamma\frac{g_t}{\|g_t\|}\right) - L(\tau_t) \leq 0. \tag{C.3}$$

Substituting this choice of $\gamma$ back to the bound for $T$ gives $T \leq \frac{\beta_u \kappa_*}{\beta_l} \cdot \frac{\|\tau_0 - \tau_*\|^2}{\delta^2}$. $\qquad\square$

## C.2 PROOF OF LEMMA 5.1

**Lemma C.1** (Lemma 5.1 restated). *Suppose Assumptions 2.2 and 5.1 hold with constant $\beta_Z$, $\beta_u$ and $\beta_l$. Let $L$ be a convex function with minimizer $\tau_*$, and let $g := \nabla L(\tau)$. For any $\delta \leq \frac{1}{\beta_Z}$, let $\gamma = \sqrt{\frac{\beta_l}{\beta_u \kappa_*}}\delta$, then for all $\tau$ s.t. $\|\tau - \tau_*\|_2 \geq \delta$, we have $L(\tau_* + \gamma\frac{g}{\|g\|}) \leq L(\tau)$.*

*Proof.* The proof follows from the Taylor expansion around $\tau_*$: for any unit vector $\boldsymbol{v}$ and any constant $c \leq \gamma$, the Taylor remainder theorem states that there exists some constant $c' < c$ and unit vector $\boldsymbol{v}'$ such that $L(\tau_* + c\boldsymbol{v}) - L(\tau_*) = \frac{c^2}{2}\boldsymbol{v}^\top \boldsymbol{H}(\tau_* + c'\boldsymbol{v}')\boldsymbol{v}$.

For any unit vector $\boldsymbol{v}_1, \boldsymbol{v}_2$ and constants $c_1, c_2 \leq \delta$ such that $L(\tau_* + c_1\boldsymbol{v}_1) = L(\tau_* + c_2\boldsymbol{v}_2)$, we have

$$L(\tau_* + c_1\boldsymbol{v}_1) - L(\tau_*) = \frac{c_1^2}{2}\boldsymbol{v}_1^\top \boldsymbol{H}(\tau_* + c_1'\boldsymbol{v}_1')\boldsymbol{v}_1 = \frac{c_2^2}{2}\boldsymbol{v}_2^\top \boldsymbol{H}(\tau_* + c_2'\boldsymbol{v}_2')\boldsymbol{v}_2 = L(\tau_* + c_2\boldsymbol{v}_2) - L(\tau_*)$$

$$\Rightarrow \frac{c_1}{c_2} \leq \sqrt{\frac{\sigma_{\max}(\boldsymbol{H}(\tau_* + c_1'\boldsymbol{v}_1'))}{\sigma_{\min}(\boldsymbol{H}(\tau_* + c_2'\boldsymbol{v}_2'))}} \leq \sqrt{\frac{\beta_u}{\beta_l}}\kappa_*. \tag{C.4}$$

This means for any two points with the same loss, the ratio between their distances to $\tau_*$ will be at most $\sqrt{\frac{\beta_u}{\beta_l}}\kappa_*$. Therefore setting $\gamma = \sqrt{\frac{\beta_l}{\beta_u \kappa_*}}\delta$ guarantees that for any $\tau$ that is at least $\delta$ away from $\tau_*$, $\tau$ will have a larger loss than any point that is $\gamma$ away from $\tau_*$. In other words, $L(\tau_1) \leq L(\tau_2)$ holds for any $\tau_1 \in \mathcal{B}(\tau_*, \gamma)$, $\tau_2 \notin \mathcal{B}(\tau_*, \delta)$. $\qquad\square$

## C.3 PROOF OF THEOREM 5.2 (CONVERGENCE RATE IN TERMS OF BHATTACHARYYA COEFFICIENT)

Recall that the *Bhattacharyya coefficient* of $P_*, Q$ is defined as $\mathrm{BC}(P_*, Q) := \int_x \sqrt{p_*(x)q(x)}dx$.

**Theorem C.2** (Theorem 5.2, restated). *Suppose Assumptions 2.1, 2.3 hold with constants $\omega$, $\lambda_{\max}$, and $\lambda_{\min}$. Consider a NCE task with data distribution $P_*$ and noise distribution $Q$, parameterized by $\theta_*, \theta_q \in \Theta$ respectively. Then for any given $\delta \leq \frac{1}{R}$ and initial estimate $\tau_0 = \tau_q$, NGD finds an estimate $\tau$ such that $\|\tau - \tau_*\|_2 \leq \delta$ within $T \leq C \cdot \frac{1}{\mathrm{BC}(P_*, Q)^3}\frac{\|\tau_0 - \tau_*\|^2}{\delta^2}$ steps, where $C := 18\exp\left(\frac{2}{\beta_Z}\right) \cdot \left(\frac{\lambda_{\max}}{\lambda_{\min}}\right)^3 \cdot \min\left\{\frac{2\lambda_{\max}^2}{\lambda_{\min}^2}, \frac{2\lambda_{\min} + \gamma_{\max}\|\bar{\delta}\|}{\lambda_{\min} - \gamma_{\min}\|\bar{\delta}\|}\right\}$.*

*Proof.* Proving Theorem 5.2 requires bounding the condition number $\kappa_*$ and the Hessian-related constants $\beta_u, \beta_l$.

We first show that $\kappa_*$ is inversely related to $\mathrm{BC}(P_*, Q)$:

**Lemma C.2.** *Let $\Theta$ be the set of parameters for an exponential family satisfying assumption 2.1-2.2. Then, for any pair of $P_*, Q$ parameterized by $\theta_*, \theta_q \in \Theta$, the NCE problem defined with $P_*, Q$ has $\kappa_* \leq \frac{\lambda_{\max}}{2\lambda_{\min}}\frac{1}{\mathrm{BC}(P_*, Q)}$.*

The next lemma provides the Hessian-related constants in Assumption 5.1:

**Lemma C.3.** *Let $\bar{\delta} := \tau - \tau_*$. Let $BC(P_*, Q)$ denote the Bhattacharyya coefficient between $P_*$ and $Q$, then for any $\tau$ such that $\|\bar{\delta}\| \leq \frac{1}{\beta_Z}$, we have:*

$$\frac{\sigma_{\max}(\nabla^2 L(\tau))}{\sigma_{\max}(\nabla^2 L(\tau_*))} \leq \frac{1}{BC(P_*, Q)} \cdot 8 \exp\left(\frac{3}{2} + \frac{1}{\beta_Z}\right) \cdot \frac{\lambda_{\max}}{\lambda_{\min}} \cdot \min\left\{\frac{2\lambda_{\max}}{\lambda_{\min}}, 2 + \frac{\gamma_{\max}\|\bar{\delta}\|}{\lambda_{\min}}\right\}$$

$$\frac{\sigma_{\min}(\nabla^2 L(\tau))}{\sigma_{\min}(\nabla^2 L(\tau_*))} \geq BC(P_*, Q) \cdot 16 \exp\left(-2 - \frac{1}{\beta_Z}\right) \cdot \frac{\lambda_{\min}}{\lambda_{\max}} \cdot \max\left\{\frac{\lambda_{\min}}{\lambda_{\max}}, 1 - \frac{\gamma_{\min}\|\bar{\delta}\|}{\lambda_{\min}}\right\}.$$

*Hence Assumption 5.1 is satisfied with constants $\beta_u, \beta_l$ equal to the respective right hand sides.*

The factor $C$ in the theorem statement is then chosen such that $\frac{C}{BC(P_*, Q)^3} \geq \frac{\beta_u}{\beta_l}$, and the proof of Theorem 5.2 is completed by applying Theorem 5.1 and the above lemmas. $\qquad\square$

### C.3.1 PROOF OF LEMMA C.2

For exponential family with pdf $p(x) = h(x) \exp\left(\theta^\top x - \log Z(\theta)\right)$, the Hessian at the optimum is:

$$\boldsymbol{H}_* = \int_x \frac{p_* q}{p_* + q} T(x) T(x)^\top dx \preceq \int_x \min\{p_*, q\} T(x) T(x)^\top dx := \boldsymbol{M}. \tag{C.5}$$

We also have $\boldsymbol{H}_* \succeq \frac{1}{2}\boldsymbol{M}$ by noting that $p_* + q \leq 2\max\{p_*, q\}$. Therefore in order to bound $\kappa_*$, it suffices to analyze the condition number of $\boldsymbol{M}$.

For any pair of distributions parameterized by $\theta, \theta_q \in \Theta$ with PDFs $p, q$, and for any unit vector $\boldsymbol{v}$, we have

$$\left(\int_x \sqrt{p}\sqrt{q}(\boldsymbol{v}^\top T(x))^2\right)^2 = \left(\int_x \min\{\sqrt{p}, \sqrt{q}\} \max\{\sqrt{p}, \sqrt{q}\}(\boldsymbol{v}^\top T(x))^2\right)^2$$

$$\overset{(i)}{\leq} \left(\int_x (\min\{\sqrt{p}, \sqrt{q}\})^2(\boldsymbol{v}^\top T(x))^2\right) \cdot \left(\int_x (\max\{\sqrt{p}, \sqrt{q}\})^2(\boldsymbol{v}^\top T(x))^2\right)$$

$$\leq \left(\int_x \min\{p, q\}(\boldsymbol{v}^\top T(x))^2\right) \cdot \left(\int_x (p + q)(\boldsymbol{v}^\top T(x))^2\right) \overset{(ii)}{\leq} 2\lambda_{\max} \int_x \min\{p, q\}(\boldsymbol{v}^\top T(x))^2 \tag{C.6}$$

where $(i)$ uses Cauchy-Schwarz, and $(ii)$ uses assumption 2.3.

Denote $B := \frac{\sqrt{Z(\theta)Z(\theta_q)}}{Z\left(\frac{\theta + \theta_q}{2}\right)}$. We have:

$$\left(\int_x \sqrt{p}\sqrt{q}(\boldsymbol{v}^\top T(x))^2\right)^2 = \frac{Z\left(\frac{\theta + \theta_q}{2}\right)^2}{Z(\theta)Z(\theta_q)}\left(\int_x p_{\frac{\theta + \theta_q}{2}}(x)(\boldsymbol{v}^\top T(x))^2\right)^2 = \frac{1}{B^2}\left(\mathbb{E}_{\frac{\theta + \theta_q}{2}}(\boldsymbol{v}^\top T(x))^2\right)^2. \tag{C.7}$$

Combining equation C.6, C.7 gives a lower bound of $\int_x \min\{p, q\}(\boldsymbol{v}^\top T(x))^2$:

$$\int_x \min\{p, q\}(\boldsymbol{v}^\top T(x))^2 \geq \frac{1}{2\lambda_{\max}}\frac{1}{B^2}\left(\mathbb{E}_{\frac{\theta + \theta_q}{2}}(\boldsymbol{v}^\top T(x))^2\right)^2. \tag{C.8}$$

On the other hand, $\int_x \min\{p, q\}(\boldsymbol{v}^\top T(x))^2$ can also be upper bounded as:

$$\int_x \min\{p, q\}(\boldsymbol{v}^\top T(x))^2 \leq \int_x \sqrt{p}\sqrt{q}(\boldsymbol{v}^\top T(x))^2 \leq \frac{1}{B}\mathbb{E}_{\frac{\theta + \theta_q}{2}}\left[(\boldsymbol{v}^\top T(x))^2\right]. \tag{C.9}$$

Hence the condition number of $\boldsymbol{M}$ is bounded as:

$$\kappa(\boldsymbol{M}) := \frac{\max_v \int_x \min\{p, q\}(\boldsymbol{v}^\top T(x))^2}{\min_v \int_x \min\{p, q\}(\boldsymbol{v}^\top T(x))^2} \leq \frac{\lambda_{\max}B}{2\min_{\boldsymbol{v}} \mathbb{E}_{\frac{\theta + \theta_q}{2}}\left[(\boldsymbol{v}^\top T(x))^2\right]} \leq \frac{\lambda_{\max}}{2\lambda_{\min}} \cdot B. \tag{C.10}$$

It is left to determine the value of $B$. We claim that $B = \frac{1}{\text{BC}(P,Q)}$, where $\text{BC}(P,Q)$ is the Bhattacharyya coefficient of $P$ and $Q$ defined as $\text{BC}(P,Q) := \int_x \sqrt{p(x)q(x)}dx$. To see this, note that it holds for any $x$ that $\log Z_\theta = \theta^\top x + \log h(x) - \log p_\theta(x)$. Hence for any $x$,

$$B^{-1} = \exp\left(\log Z_{\frac{\theta+\theta_q}{2}} - \frac{1}{2}\log Z_\theta - \frac{1}{2}\log Z_{\theta_q}\right) = \frac{\sqrt{p_\theta(x)p_{\theta_q}(x)}}{p_{\frac{\theta+\theta_q}{2}}(x)}. \tag{C.11}$$

Therefore $B^{-1} = \left(\int_x p_{\frac{\theta+\theta_q}{2}}(x)\right) \cdot B^{-1} = \int_x \sqrt{p_\theta(x)p_{\theta_q}(x)} = \text{BC}(P,Q)$.

### C.3.2 Proof for Lemma C.3 (Bound on $\text{BC}(P_*, Q)$)

*Proof.* For notational convenience, write $\bar\delta = [\bar\theta, \bar\alpha]$, where $\bar\alpha = \log Z(\theta_*) - \log Z(\theta)$ is the difference in the coordinate for the log partition function.

**Upper bounding** $\frac{\sigma_{\max}(\nabla^2 L(\tau))}{\sigma_{\max}(\nabla^2 L(\tau_*))}$**:** We proceed by splitting $\boldsymbol{v}^\top \nabla^2 L(\tau)\boldsymbol{v}$ into two terms:

$$\boldsymbol{v}^\top \nabla^2 L(\tau)\boldsymbol{v}$$
$$= \int_{\bar\delta^\top T(x)<0} (p_* + q)\frac{pq}{(p+q)^2}(\boldsymbol{v}^\top T(x))^2 dx + \int_{\bar\delta^\top T(x)>0} (p_* + q)\frac{pq}{(p+q)^2}(\boldsymbol{v}^\top T(x))^2 dx. \tag{C.12}$$

The first term is bounded as:

$$\int_{\bar\delta^\top T(x)<0} (p_* + q)\frac{pq}{(p+q)^2}(\boldsymbol{v}^\top T(x))^2 dx = \int_{\bar\delta^\top T(x)<0} (p_* + q)\frac{1}{\frac{p}{q} + \frac{q}{p} + 2}(\boldsymbol{v}^\top T(x))^2 dx$$

$$\leq \int_{\bar\delta^\top T(x)<0} (p_* + q)\frac{1}{\frac{p}{q} + \frac{q}{p}}(\boldsymbol{v}^\top T(x))^2 dx \leq \int_{\bar\delta^\top T(x)<0} (p_* + q)\cdot\min\left\{\frac{q}{p},\frac{p}{q}\right\}(\boldsymbol{v}^\top T(x))^2 dx$$

$$= \int_{\bar\delta^\top T(x)<0} (p_* + q)\cdot\min\left\{\frac{q}{p_* \exp\left(\bar\delta^\top T(x)\right)}, \frac{p_* \exp(\bar\delta^\top T(x))}{q}\right\}(\boldsymbol{v}^\top T(x))^2 dx$$

$$= \int_{\bar\delta^\top T(x)<0} \exp\left(-\bar\delta^\top T(x)\right)(p_* + q)\min\left\{\frac{q}{p_*}, \frac{p_* \exp(2\bar\delta^\top T(x))}{q}\right\}(\boldsymbol{v}^\top T(x))^2 dx$$

$$\overset{(i)}{\leq} \int_{\bar\delta^\top T(x)<0} \exp\left(-\bar\delta^\top T(x)\right)(p_* + q)\min\left\{\frac{q}{p_*}, \frac{p_*}{q}\right\}(\boldsymbol{v}^\top T(x))^2 dx$$

$$\leq 2\int_{\bar\delta^\top T(x)<0} \exp\left(-\bar\delta^\top T(x)\right)\min\{q, p_*\}(\boldsymbol{v}^\top T(x))^2 dx$$

$$\overset{(ii)}{\leq} 2\int_x \exp\left(-\bar\delta^\top T(x)\right)\min\{q, p_*\}(\boldsymbol{v}^\top T(x))^2 dx \leq 2\int_x \exp\left(-\bar\delta^\top T(x)\right)\sqrt{p_* q}(\boldsymbol{v}^\top T(x))^2 dx$$

$$= 2\frac{Z(\frac{\theta_*+\theta_q}{2} - \bar\theta)\exp(-\bar\alpha)}{\sqrt{Z(\theta_*)Z(\theta_q)}}\int_x p_{\frac{\theta_*+\theta_q}{2}-\bar\theta}\cdot(\boldsymbol{v}^\top T(x))^2 dx$$

$$\leq 2\frac{Z(\frac{\theta_*+\theta_q}{2} - \bar\theta)\exp(-\bar\alpha)}{\sqrt{Z(\theta_*)Z(\theta_q)}}\mathbb{E}_{\frac{\theta_*+\theta_q}{2}-\bar\theta}(\boldsymbol{v}^\top T(x))^2$$

$$\overset{(iii)}{\leq} 2\underbrace{\frac{Z(\frac{\theta_*+\theta_q}{2})}{\sqrt{Z(\theta_*)Z(\theta_q)}}}_{:=1/B}\cdot\exp\left(\beta_Z\bar\theta - \bar\alpha\right)\cdot\mathbb{E}_{\frac{\theta_*+\theta_q}{2}-\bar\theta}(\boldsymbol{v}^\top T(x))^2$$

$$\overset{(iv)}{\leq} \frac{2}{B}\cdot\exp\left(1 + \frac{1}{\beta_Z}\right)\cdot\mathbb{E}_{\frac{\theta_*+\theta_q}{2}-\bar\theta}(\boldsymbol{v}^\top T(x))^2 \tag{C.13}$$

where step $(i)$ is because $\bar\delta^\top T(x) < 0$; step $(ii)$ increases the value by integrating over all $x$; step $(iii)$ uses Assumption 2.2 on Lipschitz log partition function; and step $(iv)$ follows from the choice of $\bar\delta = [\bar\theta, \bar\alpha]$ that $\|\bar\delta\| \leq \frac{1}{\beta_Z}$.

The second term can be bounded as:

$$\int_{\bar{\delta}^\top T(x)>0} (p_* + q)\frac{pq}{(p+q)^2}(\boldsymbol{v}^\top T(x))^2 dx$$

$$\leq \int_{\bar{\delta}^\top T(x)>0} \frac{p_* + q}{p+q}\min\{p,q\}(\boldsymbol{v}^\top T(x))^2 dx \leq \int_{\bar{\delta}^\top T(x)>0}\min\{p,q\}(\boldsymbol{v}^\top T(x))^2 dx$$

$$\leq \int_x \sqrt{pq}(\boldsymbol{v}^\top T(x))^2 dx = \frac{Z(\frac{\theta_*+\bar{\theta}+\theta_q}{2})\exp(-\bar{\alpha})}{\sqrt{Z(\theta_*+\bar{\theta})Z(\theta_q)}}\mathbb{E}_{\frac{\theta_*+\bar{\theta}+\theta_q}{2}}(\boldsymbol{v}^\top T(x))^2 \qquad \text{(C.14)}$$

$$\overset{(i)}{\leq} \frac{Z(\frac{\theta_*+\theta_q}{2})}{\sqrt{Z(\theta_*)Z(\theta_q)}}\mathbb{E}_{\frac{\theta_*+\bar{\theta}+\theta_q}{2}}(\boldsymbol{v}^\top T(x))^2 \cdot \exp\left(\frac{3}{2}\beta_Z\|\bar{\theta}\|_2 - \bar{\alpha}\right)$$

$$\leq \frac{1}{B}\exp\left(\frac{3}{2} + \frac{1}{\beta_Z}\right)\cdot \mathbb{E}_{\frac{\theta_*+\bar{\theta}+\theta_q}{2}}(\boldsymbol{v}^\top T(x))^2$$

where step $(i)$ uses Assumption 2.2 about Lipschitzness of the log partition function, and step $(ii)$ is because we have chosen that $\|\bar{\delta}\|_2 \leq \frac{1}{\beta_Z}$.

Substituting back to equation C.12 gives:

$$\boldsymbol{v}^\top\nabla^2 L(\tau)\boldsymbol{v} \leq \frac{1}{B}\left[2\exp\left(1 + \frac{1}{\beta_Z}\right)\cdot \mathbb{E}_{\frac{\theta_*+\theta_q}{2}-\bar{\theta}}(\boldsymbol{v}^\top T(x))^2 + \exp\left(\frac{3}{2} + \frac{1}{\beta_Z}\right)\cdot \mathbb{E}_{\frac{\theta_*+\theta_q+\bar{\theta}}{2}}(\boldsymbol{v}^\top T(x))^2\right]$$

$$\leq \frac{2\exp(\frac{3}{2} + \frac{1}{\beta_Z})}{B}\cdot\min\left\{\lambda_{\max},\ \sigma_{\max}(\mathbb{E}_{\frac{\theta_*+\theta_q}{2}}[T(x)T(x)^\top]) + \gamma_{\max}\|\bar{\delta}\|\right\} \qquad \text{(C.15)}$$

where the second inequality uses Assumption 2.3 and Assumption 2.4 for the first and second term respectively.

Recall that $\boldsymbol{v}^\top\nabla^2 L(\tau_*)\boldsymbol{v} \geq \frac{1}{4B^2}\frac{1}{\lambda_{\max}}\left(\mathbb{E}_{\frac{\theta_*+\theta_q}{2}}(\boldsymbol{v}^\top T(x))^2\right)^2$. Hence:

$$\frac{\sigma_{\max}(\nabla^2 L(\tau))}{\sigma_{\max}(\nabla^2 L(\tau_*))} = \frac{\max_{\boldsymbol{v}}\boldsymbol{v}^\top\nabla^2 L(\tau)\boldsymbol{v}}{\max_{\tilde{\boldsymbol{v}}'}\tilde{\boldsymbol{v}}^\top\nabla^2 L(\tau_*)\tilde{\boldsymbol{v}}}$$

$$\leq 8\lambda_{\max}B\exp\left(\frac{3}{2} + \frac{1}{\beta_Z}\right)\frac{\mathbb{E}_{\frac{\theta_*+\theta_q}{2}-\bar{\theta}}(\boldsymbol{v}^\top T(x))^2 + \mathbb{E}_{\frac{\theta_*+\theta_q+\bar{\theta}}{2}}(\boldsymbol{v}^\top T(x))^2}{\max_{\tilde{\boldsymbol{v}}}\left(\mathbb{E}_{\frac{\theta_*+\theta_q}{2}}(\tilde{\boldsymbol{v}}^\top T(x))^2\right)^2} \qquad \text{(C.16)}$$

$$\leq 8\frac{\lambda_{\max}}{\lambda_{\min}}B\exp\left(\frac{3}{2} + \frac{1}{\beta_Z}\right)\cdot\min\left\{\frac{2\lambda_{\max}}{\lambda_{\min}}, 2 + \frac{\gamma_{\max}\|\bar{\delta}\|}{\sigma_{\max}(\mathbb{E}_{\frac{\theta_*+\theta_q}{2}}[T(x)T(x)^\top])}\right\}$$

$$\leq 8\frac{\lambda_{\max}}{\lambda_{\min}}B\exp\left(\frac{3}{2} + \frac{1}{\beta_Z}\right)\cdot\min\left\{\frac{2\lambda_{\max}}{\lambda_{\min}}, 2 + \frac{\gamma_{\max}\|\bar{\delta}\|}{\lambda_{\min}}\right\}.$$

**Lower bounding** $\frac{\sigma_{\min}(\nabla^2 L(\tau))}{\sigma_{\min}(\nabla^2 L(\tau_*))}$: Let us denote $\mathcal{S}_1 := \{x : \bar{\delta}^\top T(x) > 0\}$ and $\mathcal{S}_{-1} := \{x : \bar{\delta}^\top T(x) \leq 0\}$. The goal is to lower bound:

$$\boldsymbol{v}^\top\nabla^2 L(\tau)\boldsymbol{v} = \int_{x\in\mathcal{S}_1}(p_* + q)\frac{pq}{(p+q)^2}(\boldsymbol{v}^\top T(x))^2 dx + \int_{x\in\mathcal{S}_{-1}}(p_* + q)\frac{pq}{(p+q)^2}(\boldsymbol{v}^\top T(x))^2 dx$$

$$:= T_1 + T_{-1} \qquad \text{(C.17)}$$

Let's lower bound $T_1, T_{-1}$ in each of the following two cases.

The first case is when $T_{-1} \geq T_1$. Let $\mathcal{S}_{\frac{1}{2}}(\boldsymbol{v}) \subset \mathcal{S}_{-1}$ denote a set s.t.

$$\int_{x\in\mathcal{S}_{\frac{1}{2}}(\boldsymbol{v})}\min\{p,q\}(\boldsymbol{v}^\top T(x))^2 \geq \frac{1}{2}\int_x\min\{p,q\}(\boldsymbol{v}^\top T(x))^2.$$

Write $\bar{\delta} = [\bar{\theta}, \bar{\alpha}]$ as before, then

$$T_1 \geq 0$$

$$T_{-1} = \int_{\bar{\delta}^\top T(x) < 0} (p_* + q) \frac{pq}{(p+q)^2} (\boldsymbol{v}^\top T(x))^2 dx \overset{(i)}{\geq} \int_{\bar{\delta}^\top T(x) < 0} \frac{pq}{p+q} (\boldsymbol{v}^\top T(x))^2 dx$$

$$\geq \frac{1}{2} \int_{\bar{\delta}^\top T(x) < 0} \min\{p, q\}(\boldsymbol{v}^\top T(x))^2 dx \overset{(ii)}{\geq} \frac{1}{2} \int_{\mathcal{S}_{\frac{1}{2}}(\boldsymbol{v})} \min\{p, q\}(\boldsymbol{v}^\top T(x))^2 dx \qquad \text{(C.18)}$$

$$\overset{(iii)}{\geq} \frac{1}{4} \int \min\{p, q\}(\boldsymbol{v}^\top T(x))^2 dx \overset{(iv)}{\geq} \frac{\exp(-\bar{\alpha})}{8\lambda_{\max}} \cdot \frac{Z(\frac{\theta + \theta_q}{2})^2}{Z(\theta)Z(\theta_q)} \left( \mathbb{E}_{\frac{\theta + \theta_q}{2}}(\boldsymbol{v}^\top T(x))^2 \right)^2$$

$$\overset{(v)}{\geq} \frac{\exp(-\bar{\alpha})}{8\lambda_{\max}} \cdot \frac{1}{B^2} \exp(-2\beta_Z \|\bar{\delta}\|) \cdot \left( \mathbb{E}_{\frac{\theta_* + \theta_q + \bar{\theta}}{2}}(\boldsymbol{v}^\top T(x))^2 \right)^2$$

where step $(i)$ uses $\frac{p_* + q}{p + q} < 1$ since $\bar{\delta}^\top T(x) < 0$; step $(ii), (iii)$ follows from the definition of $\mathcal{S}_{-1}$; step $(iv)$ uses equation C.8; and step $(v)$ uses Assumption 2.2 that the log partition function is Lipschitz.

The second case is when $T_1 \geq T_{-1}$. Let $\mathcal{S}_{\frac{1}{2}}(\boldsymbol{v}) \subset \mathcal{S}_1$ denote a set s.t.

$$\int_{x \in \mathcal{S}_{\frac{1}{2}}(\boldsymbol{v})} \min\{p, q\}(\boldsymbol{v}^\top T(x))^2 \geq \frac{1}{2} \int_x \min\{p, q\}(\boldsymbol{v}^\top T(x))^2.$$

Then $T_{-1}, T_1$ can be lower bounded as:

$$T_{-1} \geq 0$$

$$T_1 = \int_{x \in \mathcal{S}_1} (p_* + q) \frac{pq}{(p+q)^2} (\boldsymbol{v}^\top T(x))^2 dx$$

$$\geq \frac{1}{2} \int_{x \in \mathcal{S}_1} \frac{p_* + q}{p + q} \cdot \min\{p, q\}(\boldsymbol{v}^\top T(x))^2 dx \geq \frac{1}{2} \int_{x \in \mathcal{S}_1} \frac{p_*}{p} \cdot \min\{p, q\}(\boldsymbol{v}^\top T(x))^2 dx$$

$$= \frac{\exp(\bar{\alpha})}{2} \int_{x \in \mathcal{S}_1} \min\{p_{\theta_*}, p_{\theta_q - \bar{\theta}}\}(\boldsymbol{v}^\top T(x))^2 dx$$

$$\geq \frac{\exp(\bar{\alpha})}{2} \int_{x \in \mathcal{S}_{\frac{1}{2}}(\boldsymbol{v})} \min\{p_{\theta_*}, p_{\theta_q - \bar{\theta}}\}(\boldsymbol{v}^\top T(x))^2 dx \geq \frac{\exp(\bar{\alpha})}{4} \int_x \min\{p_{\theta_*}, p_{\theta_q - \bar{\theta}}\}(\boldsymbol{v}^\top T(x))^2 dx$$

$$\geq \frac{\exp(\bar{\alpha})}{8\lambda_{\max}} \cdot \frac{Z(\frac{\theta_* + \theta_q - \bar{\theta}}{2})^2}{Z(\theta_*)Z(\theta_q - \bar{\theta})} \left( \mathbb{E}_{\frac{\theta_* + \theta_q - \bar{\theta}}{2}}(\boldsymbol{v}^\top T(x))^2 \right)^2$$

$$\geq \frac{\exp(\bar{\alpha})}{8\lambda_{\max}} \cdot \frac{1}{B^2} \exp(-2\beta_Z \|\bar{\delta}\|) \cdot \left( \mathbb{E}_{\frac{\theta_* + \theta_q - \bar{\theta}}{2}}(\boldsymbol{v}^\top T(x))^2 \right)^2.$$

$$\text{(C.19)}$$

Combining both cases and using $\|\bar{\delta}\| \leq \frac{1}{\beta_Z}$, we get:

$$\boldsymbol{v}^\top \nabla^2 L(\tau)\boldsymbol{v} = T_1 + T_{-1}$$

$$\geq \frac{\exp(-2 - \frac{1}{\beta_Z})}{8\lambda_{\max}} \cdot \frac{1}{B^2} \cdot \min \left\{ \left( \mathbb{E}_{\frac{\theta_* + \theta_q + \bar{\theta}}{2}}(\boldsymbol{v}^\top T(x))^2 \right)^2, \left( \mathbb{E}_{\frac{\theta_* + \theta_q - \bar{\theta}}{2}}(\boldsymbol{v}^\top T(x))^2 \right)^2 \right\}. \qquad \text{(C.20)}$$

Recall that $\boldsymbol{v}^\top \nabla^2 L(\tau_*)\boldsymbol{v} \leq \frac{2}{B} \mathbb{E}_{\frac{\theta_* + \theta_q}{2}}[(\boldsymbol{v}^\top T(x))^2]$. Hence

$$\frac{\sigma_{\min}(\nabla^2 L(\tau))}{\sigma_{\min}(\nabla^2 L(\tau_*))} = \frac{\min_{\boldsymbol{v}} \boldsymbol{v}^\top \nabla^2 L(\tau)\boldsymbol{v}}{\min_{\tilde{\boldsymbol{v}}'} \tilde{\boldsymbol{v}}^\top \nabla^2 L(\tau_*)\tilde{\boldsymbol{v}}}$$

$$\geq \frac{16 \exp(-2 - \frac{1}{\beta_Z})}{B} \frac{\lambda_{\min}}{\lambda_{\max}} \cdot \max \left\{ \frac{\lambda_{\min}}{\lambda_{\max}}, \min \left\{ \frac{\sigma_{\min}(\mathbb{E}_{\frac{\theta_* + \theta_q + \bar{\theta}}{2}} TT^\top)}{\sigma_{\min}(\mathbb{E}_{\frac{\theta_* + \theta_q}{2}}[TT^\top])}, \frac{\sigma_{\min}(\mathbb{E}_{\frac{\theta_* + \theta_q - \bar{\theta}}{2}} TT^\top)}{\sigma_{\min}(\mathbb{E}_{\frac{\theta_* + \theta_q}{2}}[TT^\top])} \right\} \right\}$$

$$\geq \frac{16 \exp(-2 - \frac{1}{\beta_Z})}{B} \frac{\lambda_{\min}}{\lambda_{\max}} \cdot \max \left\{ \frac{\lambda_{\min}}{\lambda_{\max}}, 1 - \frac{\gamma_{\min} \|\bar{\delta}\|}{\lambda_{\min}} \right\}.$$

$$\text{(C.21)}$$

$\square$

## C.4 Proof of Lemma 5.2 (Bound on the Bhattacharyya coefficient)

**Lemma C.4** (Lemma 5.2 restated). *For $P_1, P_2$ parameterized by $\theta_1, \theta_2 \in \Theta$, if $\|\theta_1 - \theta_2\|_2^2 \leq \frac{4}{\lambda_{\max}}$, then $BC(P_1, P_2) \geq \frac{1}{2}$.*

*Proof.* Given $\theta_1, \theta_2 \in \Theta$, define a map $\phi$ from $[0, 1]$ to a function $\sqrt{p}$, where $p$ is the PDF for a distribution parameterized by some $\theta \in \Theta$: let $Z(\theta)$ denote the partition function for parameter $\theta \in \Theta$, and let $\delta := \theta_2 - \theta_1$, then $\phi(t)$ is a function of $x$ defined as:

$$\phi(t)(x) = \sqrt{h(x) \exp\left((\theta_1 + t\delta)^\top x - \log Z(\theta_1 + t\delta)\right)}. \tag{C.22}$$

Denote $\phi_t(x) := \phi(t)(x)$ and $\theta_t := \theta_1 + t\delta$ for notation convenience. Then

$$\frac{\partial \phi_t(x)}{\partial t} = \frac{\partial}{\partial t}\left(\frac{\sqrt{h}\exp\left(\frac{1}{2}\theta_t^\top x\right)}{\sqrt{Z(\theta_t)}}\right) = \frac{\sqrt{h}}{2}\exp\left(\frac{1}{2}\theta_t^\top x\right)\frac{\delta^\top x \cdot \sqrt{Z(\theta_t)} - \frac{1}{\sqrt{Z(\theta_t)}}\frac{\partial Z(\theta_t)}{\partial t}}{Z(\theta_t)}$$

$$\stackrel{(*)}{=} \frac{\sqrt{h}}{2}\exp\left(\frac{1}{2}\theta_t^\top x\right)\frac{\delta^\top x - \mathbb{E}_{\theta_t}[\delta^\top x]}{\sqrt{Z(\theta_t)}} = \frac{1}{2}\sqrt{p_{\theta_t}(x)}(\delta^\top x - \mathbb{E}_{\theta_t}[\delta^\top x]) \tag{C.23}$$

where step $(*)$ used

$$\frac{\partial Z(\theta_t)}{\partial t} = \frac{\partial}{\partial t}\int_x h(x)\exp\left(\theta_t^\top x\right) = \int_x h(x)\exp\left(\theta_t^\top x\right)\delta^\top x = Z(\theta_t)\mathbb{E}_{\theta_t}[\delta^\top x]. \tag{C.24}$$

Hence

$$\left\|\frac{\partial \phi_t}{\partial t}\right\|_{L_2} := \int_x \left(\frac{\partial \phi_t(x)}{\partial t}\right)^2 = \frac{\int_x p_{\theta_t}(x)\left(\delta^\top x - \mathbb{E}_{\theta_t}[\delta^\top x]\right)^2}{4}$$

$$= \frac{\text{Var}_{\theta_t}(\delta^\top x)}{4} = \frac{\delta^\top \mathbb{E}_{\theta_t}[xx^\top]\delta^\top}{4} \leq \frac{\lambda_{\max}}{4}\|\delta\|_2^2. \tag{C.25}$$

Using the fundamental theorem of calculus, we get

$$\|\sqrt{p_{\theta_1}} - \sqrt{p_{\theta_2}}\|_{L_2} = \|\phi(1) - \phi(0)\|_{L_2} = \int_{t=0}^1 \frac{\partial \phi_t(x)}{\partial t}dt \leq \int_{t=0}^1 \left\|\frac{\partial \phi_t(x)}{\partial t}\right\|dt \leq \frac{\lambda_{\max}}{4}\|\delta\|_2^2. \tag{C.26}$$

Hence $\int_x \sqrt{p_{\theta_1}}\sqrt{p_{\theta_2}} \geq 1 - \frac{\lambda_{\max}}{8}\|\delta\|^2$, or $\frac{1}{\int_x \sqrt{p_{\theta_1}p_{\theta_2}}} \leq \frac{1}{1 - \frac{\lambda_{\max}}{8}\|\delta\|^2}$ for $\|\delta\|^2 < \frac{8}{\lambda_{\max}}$. In particular, for any $\theta_1, \theta_2$ satisfying $\|\delta\|^2 := \|\theta_1 - \theta_2\|^2 \leq \frac{4}{\lambda_{\max}}$, $\frac{1}{\int_x \sqrt{p_{\theta_1}p_{\theta_2}}} = \frac{1}{BC(P,Q)} \leq 2$, i.e. $BC(P,Q) \geq \frac{1}{2}$. $\square$

As a side note, another bound we can get is from Lipschitzness of the log partition function:

$$B := \frac{Z(\frac{\theta_1 + \theta_2}{2})}{\sqrt{Z(\theta_1)Z(\theta_2)}} = \frac{1}{BC(P,Q)}$$

$$\leq \frac{Z(\theta_*)\exp\left(\beta_Z \cdot \frac{\|\theta_* - \theta_q\|_2}{2}\right)}{\sqrt{Z(\theta_*) \cdot Z(\theta_*)\exp\left(-\beta_Z\|\theta_* - \theta_q\|\right)}} = \exp\left(\frac{3}{2}\beta_Z \cdot \|\theta_* - \theta_q\|_2\right) \tag{C.27}$$

which is tighter than $\frac{1}{1 - \frac{\lambda_{\max}}{8}\|\delta\|^2}$ if $\sqrt{\lambda_{\max}} \gg \beta_Z$.

## D Proofs for Section 4 (negative results of NCE)

This section provides proofs for the negative results in section 4, that is, the NCE landscape is ill-behaved with exponentially flat loss, gradient, and curvature. We will first prove Theorem 4.1 and properties regarding losses and gradients, then prove results related to second-order properties (Lemma 4.1, 4.3, 4.2, Theorem 4.2).

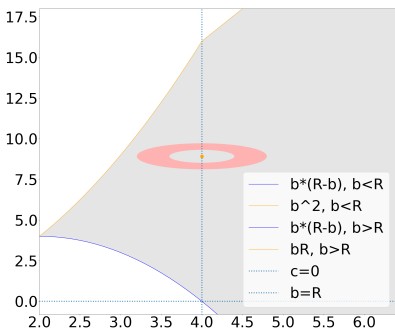

Figure 3: The gray-shaded area is the region where equation D.7 is satisfied. The orange dot marks $\tau_*$, which is enclosed in the green-shaded area. Moreover, the red-shaded area centered at $\tau_*$ corresponds the width-$0.1R$ annulus $\mathcal{A}$, within which the gradient is exponentially small.

### D.1  PROOF OF THEOREM 4.1 (LOWER BOUND FOR GRADIENT-BASED METHODS)

**Theorem D.1** (Theorem 4.1 restated). *Let $P_*, Q, P$ be 1d Gaussian with variance 1. Assume $\theta_q = 0, \theta_* > 0$ without loss of generality, and assume $R := \theta_* - \theta_q \gg 1$. Then, gradient descent with any step size $\eta = o(1)$ from an initialization $\tau = \tau_q$ will need an exponential number of steps to reach some $\tau'$ that is $O(1)$ close to $\tau_*$.*

*Proof.* The key lemma to prove Theorem 4.1 is as follows, which upper bounds the decrease in parameter distance from each gradient step:

**Lemma D.1.** *Consider the annulus $\mathcal{A} := \{(b, c) : (c - \frac{R^2}{2})^2 + (b - R)^2 \in [(0.1R)^2, (0.2R)^2]\}$. Then, for any $(b, c) \in \mathcal{A}$, it satisfies that*

$$\left| \langle \nabla L(\tau), \frac{\tau_* - \tau}{\|\tau_* - \tau\|} \rangle \right| = O(1) \cdot \exp\left( -\frac{\kappa(b, c) \cdot R^2}{8} \right) \tag{D.1}$$

*where $\kappa(b, c) \in [\frac{3}{4}, \frac{5}{4}]$ is a small constant.*

Lemma D.1 is proved in section D.2.

To prove Theorem 4.1, we will first show that Lemma D.1 serves as an upper bound for the decrease in parameter distance, that is, showing $\eta \left| \langle \nabla L(\tau), \frac{\tau_* - \tau}{\|\tau_* - \tau\|} \rangle \right| \geq \|\tau_t - \tau_*\| - \|\tau_{t+1} - \tau_*\|$. Towards this claim, we write $\tau_{t+1}$ as:

$$\tau_{t+1} = \tau_t - \eta \nabla L(\tau_t) = \tau_t - \eta \left\langle \nabla L(\tau_t), \frac{\tau_* - \tau_t}{\|\tau_* - \tau_t\|} \right\rangle \cdot \frac{\tau_* - \tau_t}{\|\tau_* - \tau_t\|} - \eta v \tag{D.2}$$

where $v := \nabla L(\tau_t) - \langle \nabla L(\tau_t), \frac{\tau_* - \tau_t}{\|\tau_* - \tau_t\|} \rangle \cdot \frac{\tau_* - \tau_t}{\|\tau_* - \tau_t\|}$ is orthogonal to $\tau_* - \tau_t$. Hence

$$\|\tau_{t+1} - \tau_*\| = \left( 1 - \frac{\eta}{\|\tau_* - \tau_t\|} \left\langle \nabla L(\tau_t), \frac{\tau_* - \tau_t}{\|\tau_* - \tau_t\|} \right\rangle \right) \cdot \|\tau_t - \tau_*\| + \eta \|v\|. \tag{D.3}$$

From this, we can conclude

$$\|\tau_t - \tau_*\| - \|\tau_{t+1} - \tau_*\| = \eta \left\langle \nabla L(\tau_t), \frac{\tau_* - \tau_t}{\|\tau_* - \tau_t\|} \right\rangle - \eta \|v\| \leq \eta \left| \left\langle \nabla L(\tau_t), \frac{\tau_* - \tau_t}{\|\tau_* - \tau_t\|} \right\rangle \right|. \tag{D.4}$$

The next step is to show that there is a path lying in $\mathcal{A}$ of length at $0.01R$ that gradient descent has to go through. We have the following lemma (proof in appendix D.3):

**Lemma D.2.** *Let $\eta = o(1)$. For any $\tau$ s.t. $\|\tau - \tau_*\| \geq 0.2R$, let $\tau'$ denote the point after one step of gradient descent from $\tau$, then $\|\tau' - \tau_*\| > 0.15R$.*

From any such $\tau'$, the shortest way to exit the annulus $\mathcal{A}$ is to project onto the inner circle defining $\mathcal{A}$, i.e. the circle centered at $\tau_*$ with radius $0.1R$ which is a convex set. Denote this inner circle as $\mathcal{B}(\tau_*, 0.1R)$ whose projection is $\Pi_{\mathcal{B}(\tau_*, 0.1R)}$, then the shortest path is the line segment $\tau' - \Pi_{\mathcal{B}(\tau_*, 0.1R)}(\tau')$. Further, this line segment is of length $0.05R$ since $\|\tau' - \tau_*\| > 0.15R$ by Lemma D.2, while the decrease of the parameter distance (i.e. $\|\tau - \tau_*\|$) is exponentially small at any point in $\mathcal{A}$ by Lemma D.1 and equation D.4. Hence the number of steps to exit $\mathcal{A}$ is lower bounded by

$$\frac{0.05R}{\eta \cdot O(1) \cdot \exp\left(-\frac{\kappa R^2}{8}\right)} = \omega(R)\exp\left(\frac{\kappa R^2}{8}\right). \qquad \square$$

### D.2 Proof of Lemma D.1

Recall that for 1d Gaussian with a known unit covariance, we can use parameter $\tau := [b, c]$ and sufficient statistics $T(x) := [x, -1]$, with pdf $p(x) = \exp\left(-\frac{x^2}{2}\right) \cdot \exp\left(\langle \tau, T(x)\rangle\right)$.

For any $\tau$ such that $\|\tau_* - \tau\| \geq 1$, $\left|\langle \nabla L(\tau), \frac{\tau_* - \tau}{\|\tau_* - \tau\|}\rangle\right|$ can be upper bounded as:

$$2\left|\langle \nabla L(\tau), \frac{\tau_* - \tau}{\|\tau_* - \tau\|}\rangle\right| \leq 2\left|\langle \nabla L(\tau), \tau_* - \tau\rangle\right| = \left|\int_x \frac{p - p_*}{\frac{p}{q}+1}\langle T(x), \tau_* - \tau\rangle\right|$$

$$= \left|\int_x \frac{p - p_*}{\frac{p}{q}+1}\left[(R-b)x - \frac{R^2}{2} - \log\sqrt{2\pi} + c\right]\right| \qquad (D.5)$$

$$\leq (R-b)\left|\int_x \frac{p - p_*}{\frac{p}{q}+1}x\right| + \left|\frac{R^2}{2} + \log\sqrt{2\pi} - c\right| \cdot \left|\int_x \frac{p - p_*}{\frac{p}{q}+1}\right|.$$

Let $a \simeq b$ denote $a = kb$ for a constant $k = \Theta(1)$. We first show the calculations with $b > 0$ for cleaner presentation; the $b < 0$ case is analogous and deferred to D.2.2.

**Bounding** $\left|\int_x \frac{p - p_*}{\frac{p}{q}+1}\right|$**:**

$$\left|\int_x \frac{p - p_*}{\frac{p}{q}+1}\right| = \left|\int_x \frac{\exp\left(-\frac{x^2}{2} + bx - c\right) - \exp\left(-\frac{(x-R)^2}{2} - \log\sqrt{2\pi}\right)}{\exp\left(bx - c + \log\sqrt{2\pi}\right) + 1}\right|$$

$$\leq \underbrace{\int_{x < \frac{c - \log\sqrt{2\pi}}{b}} \exp\left(-\frac{x^2}{2} + bx - c\right)}_{T_1^{(0)}} + \underbrace{\int_{x \geq \frac{c - \log\sqrt{2\pi}}{b}} \exp\left(-\frac{x^2}{2} - \log\sqrt{2\pi}\right)}_{T_2^{(0)}}$$

$$+ \underbrace{\int_{x < \frac{c - \log\sqrt{2\pi}}{b}} \exp\left(-\frac{(x-R)^2}{2} - \log\sqrt{2\pi}\right)}_{T_3^{(0)}} + \underbrace{\int_{x \geq \frac{c - \log\sqrt{2\pi}}{b}} \exp\left(-\frac{x^2}{2} + (R-b)x + c - \frac{R^2}{2} - 2\log\sqrt{2\pi}\right)}_{T_4^{(0)}}$$

$$\overset{(i)}{\simeq} \frac{1}{\sqrt{2\pi}}\frac{1}{b - \frac{c - \log\sqrt{2\pi}}{b}} \cdot \exp\left(-\frac{(c - \log\sqrt{2\pi})^2}{2b^2}\right) + \frac{1}{\sqrt{2\pi}}\frac{1}{\frac{c - \log\sqrt{2\pi}}{b}}\exp\left(-\frac{(c - \log\sqrt{2\pi})^2}{2b^2}\right)$$

$$+ \frac{1}{\sqrt{2\pi}}\frac{1}{R - \frac{c - \log\sqrt{2\pi}}{b}}\exp\left(-\frac{(\frac{c - \log\sqrt{2\pi}}{b} - R)^2}{2}\right) + \frac{1}{\sqrt{2\pi}}\frac{1}{\frac{c - \log\sqrt{2\pi}}{b} - (R-b)}\exp\left(-\frac{\left(\frac{c - \log\sqrt{2\pi}}{b} - R\right)^2}{2}\right)$$

$$\simeq \frac{b}{b^2 - c} \cdot \exp\left(-\frac{c^2}{2b^2}\right) + \frac{b}{c}\exp\left(-\frac{c^2}{2b^2}\right) + \frac{b}{bR - c}\exp\left(-\frac{(c - bR)^2}{2b^2}\right) + \frac{b}{c - b(R-b)}\exp\left(-\frac{(c - bR)^2}{2b^2}\right)$$

$$\overset{(ii)}{=} O(R^{-1}) \cdot \exp\left(-\frac{\kappa(b,c) \cdot R^2}{8}\right) \qquad (D.6)$$

where $\kappa(b,c) \in [\frac{3}{4}, \frac{5}{4}]$. Step $(i)$ uses calculations in equation D.12-D.15 (deferred to subsection D.2.1 for cleaner presentation), and assumes $(b, c)$ belongs to the set $\mathcal{V} := \{(b, c) : c \in [b(R-b), b\cdot$

$\min\{b, R\}]\}$. In particular, the annulus $\mathcal{A} := \{(b, c) : (c - \frac{R^2}{2})^2 + (b - R)^2 \in [(0.1R)^2, (0.2R)^2]\}$ is a subset of $\mathcal{V}$ when $R \gg 1$. Step $(ii)$ considers $(b, c) \in \mathcal{A}$.

We can choose $b, c$ s.t. $b \geq \frac{R}{2}, c \in [b(R - b), b \cdot \min\{b, R\}]$, so that we pick up the tails in $T_1^{(0)}$ to $T_4^{(0)}$. This means:

$$\begin{cases} c \in [b(R - b), b^2], & b \in [\frac{R}{2}, R], \\ c \in [-b(b - R), bR], & b \in [R, \infty]. \end{cases} \tag{D.7}$$

**Bounding** $\left| \int_x \frac{p - p_*}{\frac{p}{q} + 1} x \right|$: Using similar calculations as before, we have that when $c - \log \sqrt{2\pi} > 0$ (which is the case for $\tau = [b, c] \in \mathcal{A}$),

$$\left| \int_x \frac{p - p_*}{\frac{p}{q} + 1} x \right| \leq \left| \int_x \frac{p}{\frac{p}{q} + 1} x \right| + \left| \int_x \frac{p_*}{\frac{p}{q} + 1} x \right|$$

$$\leq \max \left\{ \int_{x>0} \frac{p}{\frac{p}{q} + 1} x, \ - \int_{x<0} \frac{p}{\frac{p}{q} + 1} x \right\} + \max \left\{ \int_{x>0} \frac{p_*}{\frac{p}{q} + 1} x, \ - \int_{x<0} \frac{p_*}{\frac{p}{q} + 1} x \right\}. \tag{D.8}$$

Below we bound the case where $x > 0$; the other case (i.e. $x < 0$) has an upper bound of the same order following similar calculations and is hence omitted.

$$\int_{x>0} \frac{p}{\frac{p}{q} + 1} x + \int_{x>0} \frac{p_*}{\frac{p}{q} + 1} x$$

$$\leq \underbrace{\int_{x \in [0, \frac{c - \log \sqrt{2\pi}}{b}]} \exp \left( -\frac{x^2}{2} + bx - c \right) x}_{T_1^{(1)}} + \underbrace{\int_{x \geq \frac{c - \log \sqrt{2\pi}}{b}} \exp \left( -\frac{x^2}{2} - \log \sqrt{2\pi} \right) x}_{T_2^{(1)}}$$

$$+ \underbrace{\int_{x \in [0, \frac{c - \log \sqrt{2\pi}}{b}]} \exp \left( -\frac{(x - R)^2}{2} - \log \sqrt{2\pi} \right) x}_{T_3^{(1)}}$$

$$+ \underbrace{\int_{x \geq \frac{c - \log \sqrt{2\pi}}{b}} \exp \left( -\frac{x^2}{2} + (R - b)x + c - \frac{R^2}{2} - 2 \log \sqrt{2\pi} \right) x}_{T_4^{(1)}}$$

$$\overset{(i)}{\simeq} \exp(-c) - \frac{1}{\sqrt{2\pi}} \exp \left( -\frac{(c - \log \sqrt{2\pi})^2}{2b^2} \right) + bT_1^{(0)} + \frac{1}{\sqrt{2\pi}} \frac{1}{\frac{c - \log \sqrt{2\pi}}{b}} \exp \left( -\frac{(c - \log \sqrt{2\pi})^2}{2b^2} \right)$$

$$+ \frac{1}{\sqrt{2\pi}} \exp \left( -\frac{R^2}{2} \right) - \frac{1}{\sqrt{2\pi}} \exp \left( -\frac{(c - \log \sqrt{2\pi} - bR)^2}{2b^2} \right) + RT_3^{(0)}$$

$$+ \frac{1}{\sqrt{2\pi}} \exp \left( -\frac{(c - \log \sqrt{2\pi} - bR)^2}{2b^2} \right) + (R - b)T_4^{(0)} \tag{D.9}$$

where step $(i)$ uses calculations in equation D.16-D.19. Ignoring small constants $\log \sqrt{2\pi}$ in $c - \log \sqrt{2\pi}$, and denoting $E_1 := \exp \left( -\frac{c^2}{2b^2} \right)$, $E_2 := \exp \left( -\frac{(c - bR)^2}{2b^2} \right)$ for notation convenience, we

can substitute equation D.6 and D.8 into equation D.5 as:

$$
\begin{aligned}
(R-b) & \left| \int_x \frac{p-p_*}{\frac{p}{q}+1} x \right| + \left| \frac{R^2}{2} + \log\sqrt{2} - c \right| \cdot \left| \int_x \frac{p-p_*}{\frac{p}{q}+1} \right| \\
\leq & (R-b) \cdot (T_1^{(1)} + T_2^{(1)} + T_3^{(1)} + T_4^{(1)}) + \left| \frac{R^2}{2} + \log\sqrt{2} - c \right| \cdot (T_1^{(0)} + T_2^{()} + T_3^{(0)} + T_4^{(0)}) \\
= & O(R) \left[ \exp(-c) - E_1 + bT_1^{(0)} + E_1 + \exp\left(-\frac{R^2}{2}\right) - E_2 + RT_3^{(0)} + E_2 + (R-b)T_4^{(0)} \right] \\
& + \Theta(R^2) \cdot (T_1^{(0)} + T_2^{(0)} + T_3^{(0)} + T_4^{(0)}) \\
= & O(R) \left[ \exp(-c) + \exp\left(-\frac{R^2}{2}\right) \right] + \Theta(R^2) \cdot O(R^{-1}) \exp\left(-\frac{\kappa(b,c)R^2}{8}\right) \\
= & O(R) \exp\left(-\frac{\kappa(b,c) \cdot R^2}{8}\right)
\end{aligned}
\tag{D.10}
$$

where $\kappa(b,c) \in [\frac{3}{4}, \frac{5}{4}]$ is the constant defined in equation D.6.

Since $\tau \in \mathcal{R}$, $\|\tau_* - \tau\| = \Theta(R)$, and the proof is completed by:

$$
\left| \langle \nabla L(\tau), \frac{\tau_* - \tau}{\|\tau_* - \tau\|} \rangle \right| = \frac{O(R) \exp\left(-\frac{\kappa(b,c) \cdot R^2}{8}\right)}{\Theta(R)} = O(1) \exp\left(-\frac{\kappa(b,c) \cdot R^2}{8}\right).
\tag{D.11}
$$

### D.2.1 CALCULATION DETAILS FOR EQUATION D.6 AND D.8

We now calculate term $T_i^{(0}$ and $T_i^{(1)}$ used in equation D.6 and D.8.

$$
\begin{aligned}
T_1^{(0)} &= \int_{x < \frac{c - \log\sqrt{2\pi}}{b}} \exp\left(-\frac{x^2}{2} + bx - c\right) = \exp\left(\frac{b^2}{2} - c\right) \int_{x < \frac{c - \log\sqrt{2\pi}}{b}} \exp\left(-\frac{(x-b)^2}{2}\right) \\
&= \exp\left(\frac{b^2}{2} - c\right) \int_{x < \frac{c - \log\sqrt{2\pi}}{b} - b} \exp\left(-\frac{x^2}{2}\right) \\
&\simeq \begin{cases} \exp\left(\frac{b^2}{2} - c\right) \cdot \frac{1}{b - \frac{c - \log\sqrt{2\pi}}{b}} \cdot \exp\left(-\frac{1}{2}\left(\frac{c - \log\sqrt{2\pi}}{b} - b\right)^2\right), & c - \log\sqrt{2\pi} < b^2 \\ \exp\left(\frac{b^2}{2} - c\right) \cdot \left[1 - \frac{1}{\frac{c - \log\sqrt{2\pi}}{b} - b} \cdot \exp\left(-\frac{1}{2}\left(\frac{c - \log\sqrt{2\pi}}{b} - b\right)^2\right)\right], & c - \log\sqrt{2\pi} \geq b^2 \end{cases} \\
&= \begin{cases} \frac{1}{\sqrt{2\pi}} \frac{1}{b - \frac{c - \log\sqrt{2\pi}}{b}} \cdot \exp\left(-\frac{(c - \log\sqrt{2\pi})^2}{2b^2}\right), & c - \log\sqrt{2\pi} < b^2 \\ \exp\left(\frac{b^2}{2} - c\right) - \frac{1}{\sqrt{2\pi}} \frac{1}{\frac{c - \log\sqrt{2\pi}}{b} - b} \cdot \exp\left(-\frac{(c - \log\sqrt{2\pi})^2}{2b^2}\right), & c - \log\sqrt{2\pi} \geq b^2 \end{cases}
\end{aligned}
\tag{D.12}
$$

$$
\begin{aligned}
T_2^{(0)} &= \int_{x \geq \frac{c - \log\sqrt{2\pi}}{b}} \frac{1}{\sqrt{2\pi}} \exp\left(-\frac{x^2}{2}\right) \\
&\simeq \begin{cases} \frac{1}{\sqrt{2\pi}} \frac{1}{\frac{c - \log\sqrt{2\pi}}{b}} \exp\left(-\frac{(c - \log\sqrt{2\pi})^2}{2b^2}\right), & c - \log\sqrt{2\pi} > 0 \\ 1 - \frac{1}{\sqrt{2\pi}} \frac{1}{\frac{|c - \log\sqrt{2\pi}|}{b}} \exp\left(-\frac{(c - \log\sqrt{2\pi})^2}{2b^2}\right), & c - \log\sqrt{2\pi} < 0 \end{cases}
\end{aligned}
\tag{D.13}
$$

$$
\begin{aligned}
T_3^{(0)} &= \int_{x < \frac{c - \log\sqrt{2\pi}}{b}} \frac{1}{\sqrt{2\pi}} \exp\left(-\frac{(x-R)^2}{2}\right) = \int_{x < \frac{c - \log\sqrt{2\pi}}{b} - R} \frac{1}{\sqrt{2\pi}} \exp\left(-\frac{x^2}{2}\right) \\
&= \begin{cases} \frac{1}{\sqrt{2\pi}} \frac{1}{R - \frac{c - \log\sqrt{2\pi}}{b}} \exp\left(-\frac{\left(\frac{c - \log\sqrt{2\pi}}{b} - R\right)^2}{2}\right), & c - \log\sqrt{2\pi} < bR \\ 1 - \frac{1}{\sqrt{2\pi}} \frac{1}{\frac{c - \log\sqrt{2\pi}}{b} - R} \exp\left(-\frac{\left(\frac{c - \log\sqrt{2\pi}}{b} - R\right)^2}{2}\right), & c - \log\sqrt{2\pi} \geq bR \end{cases}
\end{aligned}
\tag{D.14}
$$

$$T_4^{(0)} = \exp\left(\frac{(R-b)^2}{2} + c - \frac{R^2}{2} - 2\log\sqrt{2\pi}\right) \int_{x \geq \frac{c-\log\sqrt{2\pi}}{b}} \exp\left(-\frac{(x-(R-b))^2}{2}\right)$$

$$= \exp\left(\frac{(R-b)^2}{2} + c - \frac{R^2}{2} - 2\log\sqrt{2\pi}\right) \int_{x \geq \frac{c-\log\sqrt{2\pi}}{b}-(R-b)} \exp\left(-\frac{x^2}{2}\right)$$

$$= \begin{cases} \exp\left(\frac{(R-b)^2}{2} + c - \frac{R^2}{2} - 2\log\sqrt{2\pi}\right)\left[1 - \frac{1}{R-b-\frac{c-\log\sqrt{2\pi}}{b}}\exp\left(-\frac{(R-b-\frac{c-\log\sqrt{2\pi}}{b})^2}{2}\right)\right], & c - \log\sqrt{2\pi} < b(R-b) \\ \exp\left(\frac{(R-b)^2}{2} + c - \frac{R^2}{2} - 2\log\sqrt{2\pi}\right)\frac{1}{\frac{c-\log\sqrt{2\pi}}{b}-(R-b)}\exp\left(-\frac{(R-b-\frac{c-\log\sqrt{2\pi}}{b})^2}{2}\right), & c - \log\sqrt{2\pi} \geq b(R-b) \end{cases}$$

$$= \begin{cases} \frac{1}{2\pi}\exp\left(\frac{(R-b)^2}{2} + c - \frac{R^2}{2}\right) - \frac{1}{\sqrt{2\pi}}\frac{1}{R-b-\frac{c-\log\sqrt{2\pi}}{b}}\exp\left(-\frac{\left(\frac{c-\log\sqrt{2\pi}}{b}-R\right)^2}{2}\right) & c - \log\sqrt{2\pi} < b(R-b) \\ \frac{1}{\sqrt{2\pi}}\frac{1}{\frac{c-\log\sqrt{2\pi}}{b}-(R-b)}\exp\left(-\frac{\left(\frac{c-\log\sqrt{2\pi}}{b}-R\right)^2}{2}\right), & c - \log\sqrt{2\pi} \geq b(R-b) \end{cases}$$

$$(D.15)$$

$$T_1^{(1)} = \int_{x \in [0, \frac{c-\log\sqrt{2\pi}}{b}]} \exp\left(-\frac{x^2}{2} + bx - c\right) x$$

$$= \exp\left(\frac{b^2}{2} - c\right) \int_{x \in [0, \frac{c-\log\sqrt{2\pi}}{b}]} \exp\left(-\frac{(x-b)^2}{2}\right)(x-b) + b \cdot \int_{x \in [0, \frac{c-\log\sqrt{2\pi}}{b}]} \exp\left(-\frac{(x-b)^2}{2}\right)$$

$$\leq \exp\left(\frac{b^2}{2} - c\right) \int_{x \in [-b, \frac{c-\log\sqrt{2\pi}}{b}-b]} \exp\left(-\frac{x^2}{2}\right) x + bT_1^{(0)}$$

$$= \exp\left(\frac{b^2}{2} - c\right)\left[-\exp\left(-\frac{x^2}{2}\right)\right]_{-b}^{\frac{c-\log\sqrt{2\pi}}{b}-b} + bT_1^{(0)}$$

$$= \exp\left(\frac{b^2}{2} - c\right)\left(\exp\left(-\frac{b^2}{2}\right) - \exp\left(-\frac{(c-\log\sqrt{2\pi}-b^2)^2}{2b^2}\right)\right) + bT_1^{(0)}$$

$$= \exp(-c) - \frac{1}{\sqrt{2\pi}}\exp\left(-\frac{(c-\log\sqrt{2\pi})^2}{2b^2}\right) + bT_1^{(0)}$$

$$(D.16)$$

$$T_2^{(1)} = \int_{x \geq \frac{c-\log\sqrt{2\pi}}{b}} \frac{1}{\sqrt{2\pi}}\exp\left(-\frac{x^2}{2}\right) x = \frac{1}{\sqrt{2\pi}}\left[-\exp\left(-\frac{x^2}{2}\right)\right]_{\frac{c-\log\sqrt{2\pi}}{b}}^{\infty}$$

$$= \frac{1}{\sqrt{2\pi}}\exp\left(-\frac{(c-\log\sqrt{2\pi})^2}{2b^2}\right)$$

$$(D.17)$$

$$T_3^{(1)} = \int_{x \in [0, \frac{c-\log\sqrt{2\pi}}{b}]} \frac{1}{\sqrt{2\pi}}\exp\left(-\frac{(x-R)^2}{2}\right) x$$

$$\leq \int_{x \in [0, \frac{c-\log\sqrt{2\pi}}{b}]} \frac{1}{\sqrt{2\pi}}\exp\left(-\frac{(x-R)^2}{2}\right)(x-R) + RT_3^{(0)}$$

$$= \int_{x \in [-R, \frac{c-\log\sqrt{2\pi}}{b}-R]} \frac{1}{\sqrt{2\pi}}\exp\left(-\frac{x^2}{2}\right) x + RT_3^{(0)} = \frac{1}{\sqrt{2\pi}}\left[-\exp\left(-\frac{x^2}{2}\right)\right]_{-R}^{\frac{c-\log\sqrt{2\pi}}{b}-R} + RT_3^{(0)}$$

$$= \frac{1}{\sqrt{2\pi}}\exp\left(-\frac{R^2}{2}\right) - \frac{1}{\sqrt{2\pi}}\exp\left(-\frac{(c-\log\sqrt{2\pi}-bR)^2}{2b^2}\right) + RT_3^{(0)}$$

$$(D.18)$$

$T_4^{(1)}$

$$= \exp\left(\frac{(R-b)^2}{2} + c - \frac{R^2}{2} - 2\log\sqrt{2\pi}\right) \int_{x \geq \frac{c - \log\sqrt{2\pi}}{b}} \exp\left(-\frac{(x-(R-b))^2}{2}\right)(x-(R-b)) + (R-b)T_4^{(0)}$$

$$= \exp\left(\frac{(R-b)^2}{2} + c - \frac{R^2}{2} - 2\log\sqrt{2\pi}\right) \int_{x \geq \frac{c - \log\sqrt{2\pi}}{b} - (R-b)} \exp\left(-\frac{x^2}{2}\right)x + (R-b)T_4^{(0)}$$

$$= \exp\left(\frac{(R-b)^2}{2} + c - \frac{R^2}{2} - 2\log\sqrt{2\pi}\right) \left[-\exp\left(-\frac{x^2}{2}\right)\right]_{\frac{c - \log\sqrt{2\pi}}{b} - (R-b)}^{\infty} + (R-b)T_4^{(0)}$$

$$= \frac{1}{\sqrt{2\pi}} \exp\left(-\frac{(c - \log\sqrt{2\pi} - bR)^2}{2b^2}\right) + (R-b)T_4^{(0)}$$

$$\text{(D.19)}$$

### D.2.2 CALCULATIONS FOR $b < 0$

We now calculate the gradient norm bound for the case where $b < 0$. Recall that:

$$\|\nabla L(\tau)\|_2 \leq \|\nabla L(\tau)\|_1 = \left|\int_x \frac{p - p_*}{\frac{p}{q} + 1}x\right| + \left|\int_x \frac{p - p_*}{\frac{p}{q} + 1}\right|. \tag{D.20}$$

**Bounding** $\left|\int_x \frac{p - p_*}{\frac{p}{q} + 1}\right|$:

$$\left|\int_x \frac{p - p_*}{\frac{p}{q} + 1}\right| = \left|\int_x \frac{\exp\left(-\frac{x^2}{2} + bx - c\right) - \exp\left(-\frac{(x-R)^2}{2} - \log\sqrt{2\pi}\right)}{\exp\left(bx - c + \log\sqrt{2\pi}\right) + 1}\right|$$

$$\leq \underbrace{\int_{x < \frac{c - \log\sqrt{2\pi}}{b}} \exp\left(-\frac{x^2}{2} - \log\sqrt{2\pi}\right)}_{T_{1,-}^{(0)}} + \underbrace{\int_{x \geq \frac{c - \log\sqrt{2\pi}}{b}} \exp\left(-\frac{x^2}{2} + bx - c\right)}_{T_{2,-}^{(0)}}$$

$$+ \underbrace{\int_{x < \frac{c - \log\sqrt{2\pi}}{b}} \exp\left(-\frac{x^2}{2} + (R-b)x + c - \frac{R^2}{2} - 2\log\sqrt{2\pi}\right)}_{T_{3,-}^{(0)}} + \underbrace{\int_{x \geq \frac{c - \log\sqrt{2\pi}}{b}} \exp\left(-\frac{(x-R)^2}{2} - \log\sqrt{2\pi}\right)}_{T_{4,-}^{(0)}}$$

$$= O(1)$$

$$\text{(D.21)}$$

where $T_{i,-}^{(0)}$ terms are calculated as:

$$T_{1,-}^{(0)} = \int_{x < \frac{c - \log\sqrt{2\pi}}{b}} \exp\left(-\frac{x^2}{2} - \log\sqrt{2\pi}\right)$$

$$\simeq \begin{cases} \frac{1}{\sqrt{2\pi}} \cdot \frac{1}{-\frac{c - \log\sqrt{2\pi}}{b}} \exp\left(-\frac{(c - \log\sqrt{2\pi})^2}{2b^2}\right), & \frac{c - \log\sqrt{2\pi}}{b} < 0 \\ 1 - \frac{1}{\sqrt{2\pi}} \cdot \frac{1}{\frac{c - \log\sqrt{2\pi}}{b}} \exp\left(-\frac{(c - \log\sqrt{2\pi})^2}{2b^2}\right), & \frac{c - \log\sqrt{2\pi}}{b} > 0 \end{cases} \tag{D.22}$$

$$T_{2,-}^{(0)} = \int_{x \geq \frac{c - \log \sqrt{2\pi}}{b}} \exp\left(-\frac{x^2}{2} + bx - c\right) = \exp\left(\frac{b^2}{2} - c\right) \int_{x \geq \frac{c - \log \sqrt{2\pi}}{b}} \exp\left(-\frac{(x-b)^2}{2}\right)$$

$$= \exp\left(\frac{b^2}{2} - c\right) \int_{x \geq \frac{c - \log \sqrt{2\pi}}{b} - b} \exp\left(-\frac{x^2}{2}\right)$$

$$\simeq \begin{cases} \exp\left(\frac{b^2}{2} - c\right) \cdot \left[1 - \frac{1}{b - \frac{c - \log \sqrt{2\pi}}{b}} \cdot \exp\left(-\frac{1}{2}\left(\frac{c - \log \sqrt{2\pi}}{b} - b\right)^2\right)\right], & \frac{c - \log \sqrt{2\pi}}{b} - b < 0 \\[3mm] \exp\left(\frac{b^2}{2} - c\right) \cdot \frac{1}{\frac{c - \log \sqrt{2\pi}}{b} - b} \cdot \exp\left(-\frac{1}{2}\left(\frac{c - \log \sqrt{2\pi}}{b} - b\right)^2\right), & \frac{c - \log \sqrt{2\pi}}{b} - b > 0 \end{cases}$$

$$= \begin{cases} \exp\left(\frac{b^2}{2} - c\right) - \frac{1}{\sqrt{2\pi}} \frac{1}{b - \frac{c - \log \sqrt{2\pi}}{b}} \cdot \exp\left(-\frac{(c - \log \sqrt{2\pi})^2}{2b^2}\right), & \frac{c - \log \sqrt{2\pi}}{b} - b < 0 \\[3mm] \frac{1}{\sqrt{2\pi}} \frac{1}{\frac{c - \log \sqrt{2\pi}}{b} - b} \cdot \exp\left(-\frac{(c - \log \sqrt{2\pi})^2}{2b^2}\right), & \frac{c - \log \sqrt{2\pi}}{b} - b > 0 \end{cases}$$

$$\text{(D.23)}$$

$$T_{3,-}^{(0)} = \exp\left(\frac{(R-b)^2}{2} + c - \frac{R^2}{2} - 2\log\sqrt{2\pi}\right) \int_{x < \frac{c - \log \sqrt{2\pi}}{b}} \exp\left(-\frac{(x - (R-b))^2}{2}\right)$$

$$= \exp\left(\frac{(R-b)^2}{2} + c - \frac{R^2}{2} - 2\log\sqrt{2\pi}\right) \int_{x < \frac{c - \log \sqrt{2\pi}}{b} - (R-b)} \exp\left(-\frac{x^2}{2}\right)$$

$$= \begin{cases} \exp\left(\frac{(R-b)^2}{2} + c - \frac{R^2}{2} - 2\log\sqrt{2\pi}\right) \frac{1}{R - b - \frac{c - \log \sqrt{2\pi}}{b}} \exp\left(-\frac{(R - b - \frac{c - \log \sqrt{2\pi}}{b})^2}{2}\right), & \frac{c - \log \sqrt{2\pi}}{b} - (R-b) < 0 \\[3mm] \exp\left(\frac{(R-b)^2}{2} + c - \frac{R^2}{2} - 2\log\sqrt{2\pi}\right)\left[1 - \frac{1}{\frac{c - \log \sqrt{2\pi}}{b} - (R-b)} \exp\left(-\frac{(R - b - \frac{c - \log \sqrt{2\pi}}{b})^2}{2}\right)\right], & \frac{c - \log \sqrt{2\pi}}{b} - (R-b) > 0 \end{cases}$$

$$= \begin{cases} \frac{1}{2\pi} \exp\left(\frac{(R-b)^2}{2} + c - \frac{R^2}{2}\right) - \frac{1}{\sqrt{2\pi}} \frac{1}{\frac{c - \log \sqrt{2\pi}}{b} - (R-b)} \exp\left(-\frac{\left(\frac{c - \log \sqrt{2\pi}}{b} - R\right)^2}{2}\right) & \frac{c - \log \sqrt{2\pi}}{b} - (R-b) > 0 \\[5mm] \frac{1}{\sqrt{2\pi}} \frac{1}{R - b - \frac{c - \log \sqrt{2\pi}}{b}} \exp\left(-\frac{\left(\frac{c - \log \sqrt{2\pi}}{b} - R\right)^2}{2}\right), & \frac{c - \log \sqrt{2\pi}}{b} - (R-b) < 0 \end{cases}$$

$$\text{(D.24)}$$

$$T_{4,-}^{(0)} = \int_{x \geq \frac{c - \log \sqrt{2\pi}}{b}} \frac{1}{\sqrt{2\pi}} \exp\left(-\frac{(x - R)^2}{2}\right) = \int_{x \geq \frac{c - \log \sqrt{2\pi}}{b} - R} \frac{1}{\sqrt{2\pi}} \exp\left(-\frac{x^2}{2}\right)$$

$$= \begin{cases} \frac{1}{\sqrt{2\pi}} \frac{1}{\frac{c - \log \sqrt{2\pi}}{b} - R} \exp\left(-\frac{(\frac{c - \log \sqrt{2\pi}}{b} - R)^2}{2}\right), & \frac{c - \log \sqrt{2\pi}}{b} - R > 0 \\[3mm] 1 - \frac{1}{\sqrt{2\pi}} \frac{1}{R - \frac{c - \log \sqrt{2\pi}}{b}} \exp\left(-\frac{(\frac{c - \log \sqrt{2\pi}}{b} - R)^2}{2}\right), & \frac{c - \log \sqrt{2\pi}}{b} - R < 0 \end{cases}$$

$$\text{(D.25)}$$

**Bounding** $\left|\int_x \frac{p - p_*}{\frac{p}{q} + 1} x\right|$**:**

$$\left|\int_x \frac{p - p_*}{\frac{p}{q} + 1} x\right| \leq \left|\int_x \frac{p}{\frac{p}{q} + 1} x\right| + \left|\int_x \frac{p_*}{\frac{p}{q} + 1} x\right|$$

$$\leq \max\left\{\int_{x > 0} \frac{p}{\frac{p}{q} + 1} x, \; -\int_{x < 0} \frac{p}{\frac{p}{q} + 1} x\right\} + \max\left\{\int_{x > 0} \frac{p_*}{\frac{p}{q} + 1} x, \; -\int_{x < 0} \frac{p_*}{\frac{p}{q} + 1} x\right\}.$$

$$\text{(D.26)}$$

As before, we will show the bound for the case where $x > 0$; the other case (i.e. $x < 0$) follows a similar calculation and has an upper bound on the same order.

First consider $b < 0$, $c - \log\sqrt{2\pi} > 0$:

$$\int_{x > 0} \frac{p}{\frac{p}{q} + 1} x + \int_{x > 0} \frac{p_*}{\frac{p}{q} + 1} x$$

$$\leq \int_{x > 0} px + \int_{x > 0} p_* x \overset{(i)}{\simeq} \frac{1}{b^2 + 1} \exp(-c) + 1 + \frac{1}{1 + R^2} \exp\left(-\frac{R^2}{2}\right) = O(1),$$

$$\text{(D.27)}$$

where step $(i)$ uses the following:

$$\int_{x>0} \exp\left(-\frac{x^2}{2} + bx - c\right) x = \exp\left(\frac{b^2}{2} - c\right) \int_{x>0} \exp\left(-\frac{(x-b)^2}{2}\right) (x - b + b)$$

$$= \exp\left(\frac{b^2}{2} - c\right) \left[\int_{x>-b} \exp\left(-\frac{x^2}{2}\right) x + b \int_{x>-b} \exp\left(-\frac{x^2}{2}\right)\right]$$

$$= \exp\left(\frac{b^2}{2} - c\right) \left[\exp\left(-\frac{b^2}{2}\right) - \frac{b^2}{b^2+1} \exp\left(-\frac{b^2}{2}\right)\right] = \frac{1}{b^2+1} \exp(-c) \tag{D.28}$$

$$\int_{x>0} \exp\left(-\frac{(x-R)^2}{2}\right) x \leq \exp\left(-\frac{R^2}{2}\right) + 1 - \frac{R^2}{1+R^2} \exp\left(-\frac{R^2}{2}\right)$$

$$= 1 + \frac{1}{1+R^2} \exp\left(-\frac{R^2}{2}\right).$$

When $b < 0, c - \log\sqrt{2\pi} < 0$,

$$\int_{x>0} \frac{p}{\frac{p}{q}+1} x + \int_{x>0} \frac{p_*}{\frac{p}{q}+1} x$$

$$= \underbrace{\int_{x\in[0,\frac{c-\log\sqrt{2\pi}}{b}]} qx}_{T_{1,-}^{(1)}} + \underbrace{\int_{x\geq\frac{c-\log\sqrt{2\pi}}{b}} px}_{T_{2,-}^{(1)}} + \underbrace{\int_{x\in[0,\frac{c-\log\sqrt{2\pi}}{b}]} \frac{p_*q}{p}}_{T_{3,-}^{(1)}} + \underbrace{\int_{x\geq\frac{c-\log\sqrt{2\pi}}{b}} p_*}_{T_{4,-}^{(1)}} \tag{D.29}$$

$$\leq 16 \max\{R, |b|\},$$

where $T_{i,-}^{(1)}$ terms are calculated as:

$$T_{1,-}^{(1)} = \int_{x\in[0,\frac{c-\log\sqrt{2\pi}}{b}]} qx = \left[-\exp\left(-\frac{x^2}{2}\right)\right]_0^{\frac{c-\log\sqrt{2\pi}}{b}} = 1 - \exp\left(-\frac{(c-\log\sqrt{2\pi})^2}{2b^2}\right) \tag{D.30}$$

$$T_{2,-}^{(1)} = \int_{x\geq\frac{c-\log\sqrt{2\pi}}{b}} px = \exp\left(\frac{b^2}{2} - c\right) \int_{x\geq\frac{c-\log\sqrt{2\pi}}{b}} \exp\left(-\frac{(x-b)^2}{2}\right) x$$

$$= \exp\left(\frac{b^2}{2} - c\right) \left[\int_{x\geq\frac{c-\log\sqrt{2\pi}}{b}-b} \exp\left(-\frac{x^2}{2}\right) x + b \int_{x\geq\frac{c-\log\sqrt{2\pi}}{b}-b} \exp\left(-\frac{x^2}{2}\right)\right] \tag{D.31}$$

$$\simeq \left(1 - \frac{1}{1 - \frac{c-\log\sqrt{2\pi}}{b^2}}\right) \cdot \exp\left(-\frac{(c-\log\sqrt{2\pi})^2}{2b^2}\right)$$

$$T_{3,-}^{(1)} = \int_{x\in[0,\frac{c-\log\sqrt{2\pi}}{b}]} \frac{p_*q}{p} \simeq \int_{x\in[0,\frac{c-\log\sqrt{2\pi}}{b}]} \exp\left(-\frac{(x-R)^2}{2} - bx + c - \log\sqrt{2\pi}\right)$$

$$= \exp\left(\frac{(R-b)^2}{2} - \frac{R^2}{2} + c - \log\sqrt{2\pi}\right) \int_{x\in[0,\frac{c-\log\sqrt{2\pi}}{b}]} \exp\left(-\frac{(x-(R-b))^2}{2}\right)$$

$$= \exp\left(\frac{(R-b)^2}{2} - \frac{R^2}{2} + c - \log\sqrt{2\pi}\right) \int_{x\in[-(R-b),\frac{c-\log\sqrt{2\pi}}{b}-(R-b)]} \exp\left(-\frac{x^2}{2}\right) (x + R - b)$$

$$= \exp\left(-\frac{R^2}{2} + c - \log\sqrt{2\pi}\right) - \exp\left(-\frac{(\frac{c-\log\sqrt{2\pi}}{b} - R)^2}{2}\right) + (R - b) \cdot \beta_{3,-}^{(1)} \tag{D.32}$$

where $\beta_{3,-}^{(1)} = O(1)$ is:

$$\beta_{3,-}^{(1)} = \begin{cases} 2 - \frac{1}{R-b} \exp\left(-\frac{R^2}{2} + c - \log\sqrt{2\pi}\right) - \frac{1}{\frac{c-\log\sqrt{2\pi}}{b}-(R-b)} \exp\left(-\frac{(\frac{c-\log\sqrt{2\pi}}{b}-R)^2}{2}\right), & \frac{c-\log\sqrt{2\pi}}{b} \geq R - b \\ \frac{1}{\frac{c-\log\sqrt{2\pi}}{b}-(R-b)} \exp\left(-\frac{(\frac{c-\log\sqrt{2\pi}}{b}-R)^2}{2}\right) - \frac{1}{R-b} \exp\left(-\frac{R^2}{2} + c - \log\sqrt{2\pi}\right), & \frac{c-\log\sqrt{2\pi}}{b} < R - b \end{cases} \tag{D.33}$$

$$T_{4,-}^{(1)} = \int_{x \geq \frac{c-\log\sqrt{2\pi}}{b}} p_* = \int_{x \geq \frac{c-\log\sqrt{2\pi}}{b}} \exp\left(-\frac{(x-R)^2}{2}\right)(x - R + R)$$

$$= \int_{x \geq \frac{c-\log\sqrt{2\pi}}{b}-R} \exp\left(-\frac{x^2}{2}\right)x + R\int_{x \geq \frac{c-\log\sqrt{2\pi}}{b}-R} \exp\left(-\frac{x^2}{2}\right) \qquad (D.34)$$

$$= \exp\left(-\frac{(\frac{c-\log\sqrt{2\pi}}{b}-R)^2}{2}\right) + R \cdot \beta_{4,-}^{(1)}$$

where $\beta_{4,-}^{(1)} = O(1)$ is:

$$\beta_{4,-}^{(1)} = \begin{cases} 1 - \frac{1}{R-\frac{c-\log\sqrt{2\pi}}{b}}\exp\left(-\frac{(\frac{c-\log\sqrt{2\pi}}{b}-R)^2}{2}\right), & \frac{c-\log\sqrt{2\pi}}{b} < R \\ \frac{1}{\frac{c-\log\sqrt{2\pi}}{b}-R}\exp\left(-\frac{(\frac{c-\log\sqrt{2\pi}}{b}-R)^2}{2}\right), & \frac{c-\log\sqrt{2\pi}}{b} > R \end{cases} \qquad (D.35)$$

Combining equation D.21, D.27, and D.26 we have that $\|\nabla L([b,c])\|_2 \leq 32\max\{R,|b|\}$ for $b < 0$.

### D.3 PROOF OF LEMMA D.2

We first show the following claim, and prove Lemma D.2 at the end of this subsection:

**Claim D.1.** *For any $\tau = [b,c] \in \mathbb{R}^2$, the gradient norm at $\tau$ is $\|\nabla L(\tau)\|_2 \leq 32\max\{R,|b|\}$.*

*Proof.* For parameter $\tau = [b,c]$ where $b > 0$, $c - \log\sqrt{2\pi} > 0$,

$$\|\nabla L(\tau)\|_2 \leq \|\nabla L(\tau)\|_1 = \left|\int_x \frac{p-p_*}{\frac{p}{q}+1}x\right| + \left|\int_x \frac{p-p_*}{\frac{p}{q}+1}\right|$$

$$\stackrel{(i)}{\leq} \exp(-c) - \exp\left(-\frac{c^2}{2b^2}\right) + bT_1^{(0)} + \exp\left(-\frac{c^2}{2b^2}\right) + \exp\left(-\frac{R^2}{2}\right) - \exp\left(-\frac{(c-bR)^2}{2b^2}\right)$$

$$+ RT_3^{(0)} + \exp\left(-\frac{(c-bR)^2}{2b^2}\right) + (R-b)T_4^{(0)} + T_1^{(0)} + T_2^{(0)} + T_3^{(0)} + T_4^{(0)}$$

$$\stackrel{(ii)}{\simeq} (b+1)T_1^{(0)} + T_2^{(0)} + (R+1)T_3^{(0)} + (R-b+1)T_4^{(0)}$$

$$\leq 4 + b + R + \max\{R-b,0\} \lesssim 2\max\{R,b\}. \qquad (D.36)$$

where step $(i)$ and $(ii)$ use equation D.16-D.19 and equation D.12-D.15. Moreover, step $(ii)$ increases the value by at most 16. Hence overall we have $\|\nabla_\tau L\|_2 \leq 32\max\{R,b\}$.

When $b > 0$, $c - \log\sqrt{2\pi} < 0$:

$$\left|\int_x \frac{p-p_*}{\frac{p}{q}+1}x\right| \leq \left|\int_x \frac{p}{\frac{p}{q}+1}x\right| + \left|\int_x \frac{p_*}{\frac{p}{q}+1}x\right|$$

$$\leq \max\left\{\int_{x>0}\frac{p}{\frac{p}{q}+1}x + \int_{x>0}\frac{p_*}{\frac{p}{q}+1}x, -\int_{x<0}\frac{p}{\frac{p}{q}+1}x - \int_{x<0}\frac{p_*}{\frac{p}{q}+1}x\right\}. \qquad (D.37)$$

Let's bound the first term (i.e. $x > 0$); the bound for the second term (i.e. $x < 0$) follows from similar calculations and is on the same order.

$$\int_{x>0} \frac{p}{\frac{p}{q}+1} x + \int_{x>0} \frac{p_*}{\frac{p}{q}+1} x \le \int_{x>0} qx + \int_{x>0} \frac{p_* q}{p} x$$

$$= \frac{1}{\sqrt{2\pi}} \int_{x>0} \exp\left(-\frac{x^2}{2}\right) x + \frac{1}{\sqrt{2\pi}} \int_{x>0} \exp\left(-\frac{x^2}{2} + (R-b)x + c - \frac{R^2}{2} - \log\sqrt{2\pi}\right) x$$

$$\overset{(i)}{=} \begin{cases} 1 + (R-b)\exp\left(\frac{b^2}{2} - Rb + c - \log\sqrt{2\pi}\right) + \frac{1}{(b-R)^2+1}\exp\left(-\frac{R^2}{2} + c - \log\sqrt{2\pi}\right), & R-b > 0 \\ 1 + \frac{1}{(b-R)^2+1}\exp\left(-\frac{R^2}{2} + c - \log\sqrt{2\pi}\right), & R-b < 0 \end{cases}$$

$$= O(1).$$

(D.38)

Step $(i)$ omits a factor of $\frac{1}{\sqrt{2\pi}}$ and uses:

$$\int_{x>0} \exp\left(-\frac{x^2}{2} + (R-b)x + c - \frac{R^2}{2} - 2\log\sqrt{2\pi}\right) x$$

$$= \exp\left(\frac{(R-b)^2}{2} - \frac{R^2}{2} + c - \log\sqrt{2\pi}\right) \int_{x>0} \exp\left(-\frac{(x-(R-b))^2}{2}\right) x$$

$$= \exp\left(\frac{(R-b)^2}{2} - \frac{R^2}{2} + c - \log\sqrt{2\pi}\right) \left[\int_{x>-(R-b)} \exp\left(-\frac{x^2}{2}\right) x + (R-b)\int_{x>-(R-b)} \exp\left(-\frac{x^2}{2}\right)\right]$$

$$\simeq \begin{cases} (R-b)\exp\left(\frac{b^2}{2} - Rb + c - \log\sqrt{2\pi}\right) + \frac{1}{(b-R)^2+1}\exp\left(-\frac{R^2}{2} + c - \log\sqrt{2\pi}\right) & R-b > 0 \\ \frac{1}{(b-R)^2+1}\exp\left(-\frac{R^2}{2} + c - \log\sqrt{2\pi}\right), & R-b < 0 \end{cases}$$

(D.39)

For $b < 0$, we similarly have $\|\nabla L(\tau)\|_2 = O(\max\{R, -b\})$. The calculations are similar to the $b > 0$ case and hence omitted.

□

We are now ready to prove Lemma D.2, which we restate below.

**Lemma D.3** (Lemma D.2, restated). *Let $\eta = o(1)$. For any $\tau$ s.t. $\|\tau - \tau_*\| \ge 0.2R$, let $\tau'$ denote the point after one step of gradient descent from $\tau$, then $\|\tau' - \tau_*\| > 0.15R$.*

*Proof of Lemma D.3.* We will prove by contradiction. First assume that we can go from $\tau$ where $\|\tau - \tau_*\|_2 \ge 0.2R$ to some $\tau'$ where $\|\tau' - \tau_*\|_2 \le 0.15R$. Then $\|\tau' - \tau_*\|$ is lower bounded as:

$$\|\tau' - \tau_*\| \ge \|\tau - \tau_*\| - \eta\|\nabla L(\tau)\| \overset{(i)}{\ge} |b - R| - \eta\|\nabla L(\tau)\|$$
$$\overset{(ii)}{\ge} |b - R| - 32\eta b = \left(\left|1 - \frac{R}{b}\right| - 32\eta\right) b$$

(D.40)

where step $(i)$ uses $\|\tau - \tau^*\| \ge |\tau[1] - \tau^*[1]| \ge \|\tau_1| - |\tau_1^*\|| = |b - R|$, and step $(ii)$ is by Claim D.1.

On the other hand, we have $\|\tau' - \tau_*\| \le 0.15R$ by assumption, which when combined with equation D.40 gives $b \le \frac{0.15R}{\left|1 - \frac{R}{b}\right| - 32\eta}$, or $b = O(R)$. This means $\|\tau - \tau_*\| - \|\tau' - \tau_*\| \le \eta\|\nabla L(\tau)\| = o(1) \cdot O(\max\{R, |b|\}) = o(R)$. However, we also have $\|\tau - \tau_*\| - \|\tau' - \tau_*\| \ge 0.05R = \Theta(R)$ by assumption. This is a contradiction, which means the assumption must be false, i.e. $\tau'$ cannot satisfy $\|\tau' - \tau_*\|_2 \le 0.15R$.

□

## D.4 PROOF OF LEMMA 4.1 AND LEMMA 4.2

We prove Lemmas 4.1 and 4.2 in this section. First recall the lemma statements:

**Lemma D.4** (Smoothness at $P = P^*$, Lemma 4.1 restated). *Consider the 1d Gaussian mean estimation task with $R := |\theta_* - \theta_q| \gg 1$. Then the smoothness at $P = P_*$ is upper bounded as:*

$$\sigma_{\max}^* := \sigma_{\max}(\nabla^2 L(\tau_*)) \leq \frac{R}{\sqrt{2\pi}} \exp(-R^2/8). \tag{D.41}$$

We will also need a bound on the strong convexity constant (i.e. smallest singular value) at $P = P^*$:

**Lemma D.5** (Strong convexity at $P = P^*$, Lemma 4.2 restated). *Under the same setup as lemma 4.1, the minimum singular value at $P = P_*$ is $\sigma_{\min}^*(\nabla^2 L(\tau_*)) = \Theta\left(\frac{1}{R} \exp\left(-\frac{R^2}{8}\right)\right)$.*

### D.4.1 PROOF OF LEMMA 4.1 (SMOOTHNESS AT $P = P_*$)

We will show the smoothness constant (i.e. $\sigma_{\max}(\nabla^2 L)$) is exponentially small at the optimum, i.e. when $P = P_*$. The Hessian at the optimum is:

$$\nabla^2 L(\tau) = \frac{1}{2} \int_x \frac{p_* q}{p_* + q} T(x) T(x)^\top dx = \frac{1}{2} \int_x \frac{p_* q}{p_* + q} [x, -1]^\top [x, -1] dx. \tag{D.42}$$

Recall that $\theta_q = 0$ w.l.o.g, and assume $\theta_* = R \gg 1$. Then

$$\begin{aligned}
\nabla^2 L(\tau) =& \frac{1}{2} \int_{x \leq R/2} \frac{p_* q}{p_* + q} T(x) T(x)^\top dx + \frac{1}{2} \int_{x > R/2} \frac{p_* q}{p_* + q} T(x) T(x)^\top dx \\
\lesssim& \frac{1}{2} \int_{x \leq R/2} p_* T(x) T(x)^\top dx + \frac{1}{2} \int_{x > R/2} q T(x) T(x)^\top dx.
\end{aligned} \tag{D.43}$$

Let $\mathcal{S}^1 \subset \mathbb{R}^2$ denote the circle centered as the origin with radius 1. The maximum singular value is upper bounded by

$$\begin{aligned}
\sigma_{\max}^* :=& \max_{[a_1, a_2] \in \mathcal{S}^1} \frac{1}{2} \int_x \frac{p_* q}{p_* + q} (a_1 x - a_2)^2 dx \\
\leq& \frac{1}{2} \max_{[a_1, a_2] \in \mathcal{S}^1} \left[ \int_{x \leq R/2} p_* (a_1 x - a_2)^2 dx + \int_{x > R/2} q (a_1 x - a_2)^2 dx \right] \\
=& \frac{1}{2} \max_{[a_1, a_2] \in \mathcal{S}^1} \left[ \int_{x \leq R/2} p^* (a_1^2 x^2 - 2 a_1 a_2 x + a_2^2) dx + \int_{x > R/2} q (a_1^2 x^2 - 2 a_1 a_2 x + a_2^2) dx \right] \\
=& \frac{1}{2} \max_{[a_1, a_2] \in \mathcal{S}^1} \left[ \underbrace{\left( \int_{x \leq \frac{R}{2}} p_* x^2 + \int_{x > \frac{R}{2}} q x^2 \right)}_{T_2} \cdot a_1^2 - \underbrace{\left( \int_{x \leq \frac{R}{2}} p_* x + \int_{x > \frac{R}{2}} q x \right)}_{T_1} 2 a_1 a_2 \right. \\
& \left. + \underbrace{\left( \int_{x \leq \frac{R}{2}} p_* + \int_{x > \frac{R}{2}} q \right)}_{T_0} a_3^2 \right] \\
\overset{(i)}{\leq}& \frac{1}{2} T_2 + T_1 + \frac{T_0}{2} \overset{(ii)}{\leq} \left( \frac{R}{2} + \frac{1}{R} + 2 + \frac{2}{R} \right) \cdot \frac{1}{\sqrt{2\pi}} \exp\left( -\frac{R^2}{8} \right) \\
=& \left( \frac{R}{2} + 2 + \frac{3}{R} \right) \cdot \frac{1}{\sqrt{2\pi}} \exp\left( -\frac{R^2}{8} \right) \leq \frac{R}{\sqrt{2\pi}} \exp\left( -\frac{R^2}{8} \right),
\end{aligned} \tag{D.44}$$

where $(i)$ substitutes in $1$ or $-1$ for $a_1, a_2$ and uses the fact that the upper bounds for $T_0, T_1, T_2$ are positive. $(ii)$ uses the calculations on $T_0$ to $T_2$ shown below. We note that these calculations rely on properties of Gaussian and do not extend to general exponential families.

$$
\begin{aligned}
T_0 =& 2\int_{x>R/2} q \leq \frac{4}{R} \cdot \frac{1}{\sqrt{2\pi}} \exp\left(-\frac{R^2}{8}\right) \\
T_1 =& \int_{x\leq R/2} \frac{1}{\sqrt{2\pi}} \exp\left(-\frac{(x-\theta)^2}{2}\right) x dx + \int_{x>R/2} \frac{1}{\sqrt{2\pi}} \exp\left(-\frac{x^2}{2}\right) x dx \\
=& \int_{x'\geq R/2} \frac{1}{\sqrt{2\pi}} \exp\left(-\frac{(x')^2}{2}\right)(R-x')dx + \int_{x>R/2} \frac{1}{\sqrt{2\pi}} \exp\left(-\frac{x^2}{2}\right) x dx \\
=& R\int_{x\geq R/2} q dx \leq \frac{2}{\sqrt{2\pi}} \exp\left(-\frac{R^2}{8}\right)
\end{aligned}
\tag{D.45}
$$

For $T_2$, denote $P_{R/2} := P_*\left(\left\{x : x \leq \frac{R}{2}\right\}\right) = P_Q\left(\left\{x : x \geq \frac{R}{2}\right\}\right)$; Gaussian tail bound gives $P_{R/2} \leq \frac{2}{\sqrt{2\pi}} \frac{1}{R} \exp\left(-\frac{R^2}{8}\right)$. Then we can calculate each term in $T_2$ as:

$$
\begin{aligned}
\int_{x\leq\frac{R}{2}} p_*(x)x^2 dx =& \int_{x\leq\frac{R}{2}} \frac{1}{\sqrt{2\pi}} \exp\left(-\frac{(x-R)^2}{2}\right) x^2 dx \\
=& \int_{x\leq\frac{R}{2}} \frac{1}{\sqrt{2\pi}} \exp\left(-\frac{(x-R)^2}{2}\right)(x-R)\cdot x dx + R\int_{x\leq\frac{R}{2}} \frac{1}{\sqrt{2\pi}} \exp\left(-\frac{(x-R)^2}{2}\right) x dx \\
=& \left[-\frac{\exp\left(-\frac{(x-R)^2}{2}\right)x}{\sqrt{2\pi}}\right]_{-\infty}^{\frac{R}{2}} + \int_{x\leq\frac{R}{2}} \frac{\exp\left(-\frac{(x-R)^2}{2}\right)}{\sqrt{2\pi}}(x-R)dx \\
& + R\left(R\cdot P_{R/2} - \frac{1}{\sqrt{2\pi}} \exp\left(-\frac{R^2}{8}\right)\right) \\
=& -\left(\frac{R}{2}+1\right)\cdot\frac{1}{\sqrt{2\pi}} \exp\left(-\frac{R^2}{8}\right) + R\left(R\cdot P_{R/2} - \frac{1}{\sqrt{2\pi}} \exp\left(-\frac{R^2}{8}\right)\right) \\
=& -\left(\frac{3R}{2}+1\right)\cdot\frac{1}{\sqrt{2\pi}} \exp\left(-\frac{R^2}{8}\right) + R^2\cdot P_{R/2} \\
\leq& -\frac{3R}{2}\cdot\frac{1}{\sqrt{2\pi}} \exp\left(-\frac{R^2}{8}\right) + R^2\cdot P_{R/2}
\end{aligned}
\tag{D.46}
$$

$$
\begin{aligned}
\int_{x\geq\frac{R}{2}} q(x)x^2 dx =& \int_{x\geq\frac{R}{2}} \frac{1}{\sqrt{2\pi}} \exp\left(-\frac{x^2}{2}\right) x^2 dx = \left[-\frac{\exp(-\frac{x^2}{2})x}{\sqrt{2\pi}}\right]_{\frac{R}{2}}^{\infty} + P_{R/2} \\
=& \frac{1}{\sqrt{2\pi}}\frac{R}{2} \exp\left(-\frac{R^2}{8}\right) + P_{R/2}
\end{aligned}
\tag{D.47}
$$

Hence $T_2 \leq -R\cdot\frac{1}{\sqrt{2\pi}} \exp\left(-\frac{R^2}{8}\right) + (R^2+1)P_{R/2} \leq \left(R+\frac{2}{R}\right)\frac{1}{\sqrt{2\pi}} \exp\left(-\frac{R^2}{8}\right)$

### D.4.2 PROOF OF LEMMA 4.2 (STRONG CONVEXITY AT $P = P_*$)

Lower bounding $\sigma_{\min}^*$ follows a similar calculation as for upper bounding $\sigma_{\max}^*$:

$$
\begin{aligned}
\sigma_{\min}^* := &\min_{[a_1,a_2]\in\mathcal{S}^1} \frac{1}{2} \int_x \frac{p_* q}{p_* + q} \left(a_1 x - a_2\right)^2 dx \gtrsim \min_{[a_1,a_2]\in\mathcal{S}^1} \frac{1}{4} \int_x \frac{p_* q}{\max\{p_*, q\}} \left(a_1 x - a_2\right)^2 dx \\
= &\frac{1}{4} \min_{[a_1,a_2]\in\mathcal{S}^1} \left[ \int_{x\le R/2} p_* \left(a_1 x - a_2\right)^2 dx + \int_{x > R/2} q \left(a_1 x - a_2\right)^2 dx \right] \\
= &\frac{1}{4} \min_{[a_1,a_2]\in\mathcal{S}^1} \left[ \int_{x\le R/2} p^* \left(a_1 x^2 - 2 a_1 a_2 x + a_2^2\right) dx + \int_{x > R/2} q \left(a_1 x^2 - 2 a_1 a_2 x + a_2^2\right) dx \right] \\
= &\frac{1}{4} \min_{[a_1,a_2]\in\mathcal{S}^1} \left[ \underbrace{\left( \int_{x\le \frac{R}{2}} p_* x^2 + \int_{x > \frac{R}{2}} q x^2 \right)}_{T_2} \cdot a_1^2 - \underbrace{\left( \int_{x\le \frac{R}{2}} p_* x + \int_{x > \frac{R}{2}} q x \right)}_{T_1} 2 a_1 a_2 \right. \\
&\left. + \underbrace{\left( \int_{x\le \frac{R}{2}} p_* + \int_{x > \frac{R}{2}} q \right)}_{T_0} a_2^2 \right] \\
\overset{(i)}{\ge} &\frac{1}{4} \frac{1}{\sqrt{2\pi}} \exp\left(-\frac{R^2}{8}\right) \min_{[a_1,a_2]\in\mathcal{S}^1} \left[ \left(\frac{R}{2} + \frac{1}{R}\right) a_1^2 - 4 a_1 a_2 + \frac{1}{R} a_2^2 \right] \\
= &\frac{1}{4} \frac{1}{\sqrt{2\pi}} \exp\left(-\frac{R^2}{8}\right) \min_{a\in[0,1]} \left[ \left(\frac{R}{2} + \frac{1}{R}\right) a^2 - 4 a\sqrt{1 - a^2} + \frac{1}{R}(1 - a^2) \right] \\
= &\frac{1}{4} \frac{1}{\sqrt{2\pi}} \exp\left(-\frac{R^2}{8}\right) \min_{a\in[0,1]} \left[ \frac{R}{2} a^2 - 4 a\sqrt{1 - a^2} + \frac{1}{R} \right] \\
= &\frac{1}{4} \frac{1}{\sqrt{2\pi}} \exp\left(-\frac{R^2}{8}\right) \min_{a\in[0,1]} \left[ a\left(\frac{R}{2} a - 4\sqrt{1 - a^2}\right) + \frac{1}{R} \right] \\
\overset{(ii)}{\ge} &\frac{1}{4R} \frac{1}{\sqrt{2\pi}} \exp\left(-\frac{R^2}{8}\right)
\end{aligned}
$$

(D.48)

where $(i)$ uses the calculations on $T_0$ to $T_2$ stated in equation D.45 and D.46. Step $(ii)$ replaces $a = 0$ to remove the $O(R)$ term.

### D.5 PROOF OF LEMMA 4.3 (CURVATURE AT $P = Q$)

**Lemma D.6** (Smoothness at $P = Q$, Lemma 4.3 restated). *Under the same setup as Lemma 4.1, the smoothness at $P = Q$ is lower bounded as $\sigma_{\max}(\nabla^2 L(\tau_q)) \ge \frac{R^2}{2}$.*

*Proof.* The result follows from direct calculation of the Hessian at $P = Q$:

$$
\begin{aligned}
\nabla^2 L(\tau) = &\frac{1}{2} \int_x \frac{p_* + q}{4} T(x) T(x)^\top dx = \frac{1}{8} \left( \mathbb{E}_*(T(x) T(x)^\top) + \mathbb{E}_Q(T(x) T(x)^\top) \right) \\
= &\frac{1}{8} \left( \mathbb{E}_* \left( \begin{bmatrix} x \\ -1 \end{bmatrix} [x, -1] \right) + \mathbb{E}_Q \left( \begin{bmatrix} x \\ -1 \end{bmatrix} [x, -1] \right) \right) = \frac{1}{8} \begin{bmatrix} \mathbb{E}_* x^2 + \mathbb{E}_Q x^2 & 0 \\ 0 & 2 \end{bmatrix} \\
= &\frac{1}{8} \begin{bmatrix} R^2 + 2 & 0 \\ 0 & 2 \end{bmatrix}.
\end{aligned}
$$

(D.49)

Hence $\sigma_{\max}(\nabla_\tau^2 L) \ge \boldsymbol{e}_1^\top \nabla^2 L(\tau) \boldsymbol{e}_1 \ge \frac{R^2}{2}$. $\qquad\square$

### D.6 PROOF OF THEOREM 4.2 (LOWER BOUND FOR SECOND-ORDER METHODS)

The proof of Theorem 4.2 is similar to that of Theorem 4.1, where we show that there is a ring of width $\Theta(R)$ in which the amount of progress at each step is exponentially small, hence the number of steps required to cross this ring is exponential.

We show that starting from $\tau_0 = \tau_q$, the optimization path will necessarily steps into $\mathcal{A}$:

**Lemma D.7.** *Let* $\eta := O(\frac{\lambda_\rho}{\lambda_M})$, *where* $\lambda_\rho := \min_{\theta \in \Theta} \sigma_{\min}(\nabla^2 L(\tau_\theta))$, $\lambda_M := \max_{\theta \in \Theta} \sigma_{\max}(\nabla^2 L(\tau_\theta))$ *as defined in Section 4. For any* $\tau$ *s.t.* $\|\tau - \tau_*\| \geq 0.2R$, *let* $\tau'$ *denote the point after one step of gradient descent from* $\tau$, *then* $\|\tau' - \tau_*\|_2 > 0.15R$.

*Proof.* First note that $\forall \tau$, the next point after one step of Newton update is:

$$\tau_{t'} = \tau - \eta(\nabla^2 L(\tau))^{-1} \nabla L(\tau) = \tau - \eta \left[ \left\langle (\nabla^2 L(\tau))^{-1} \nabla L(\tau), \frac{\tau - \tau_*}{\|\tau - \tau_*\|_2} \right\rangle \cdot \frac{\tau - \tau_*}{\|\tau - \tau_*\|_2} + \boldsymbol{v} \right] \tag{D.50}$$

where $\boldsymbol{v} := (\nabla^2 L(\tau))^{-1} \nabla L(\tau) - \langle (\nabla^2 L(\tau))^{-1} \nabla L(\tau), \frac{\tau - \tau_*}{\|\tau - \tau_*\|_2} \rangle \cdot \frac{\tau - \tau_*}{\|\tau - \tau_*\|_2}$ is orthogonal to $\tau - \tau_*$. Hence

$$\|\tau - \tau_*\| - \|\tau' - \tau_*\| = \eta \left\langle (\nabla^2 L(\tau))^{-1} \nabla L(\tau), \frac{\tau - \tau_*}{\|\tau - \tau_*\|_2} \right\rangle - \eta \|\boldsymbol{v}\|$$

$$\leq \frac{\eta}{\sigma_{\min}(\nabla^2 L(\tau))} \cdot \left| \left\langle \nabla L(\tau), \frac{\tau - \tau_*}{\|\tau - \tau_*\|_2} \right\rangle \right| \leq \frac{\eta \|\nabla L(\tau)\|_2}{\sigma_{\min}(\nabla^2 L(\tau))} \overset{(i)}{\leq} \frac{32\eta \max\{R, |b|\}}{\sigma_{\min}(\nabla^2 L(\tau))} \tag{D.51}$$

$$\overset{(ii)}{\leq} 32 \frac{\lambda_\rho}{\sigma_{\min}(\nabla^2 L(\tau))} \frac{\max\{R, |b|\}}{\lambda_M} \leq \frac{32 \max\{R, |b|\}}{\lambda_M} \leq \frac{64 \max\{R, |b|\}}{R^2}$$

where step $(i)$ uses Claim D.1, and step $(ii)$ follows from the choice of $\eta$.

Suppose $\|\tau' - \tau_*\|_2 < 0.15R$, then

$$0.05R \leq \|\tau - \tau_*\| - \|\tau' - \tau_*\| \leq \frac{64 \max\{R, |b|\}}{R^2} \Rightarrow b = \Omega(R^3). \tag{D.52}$$

However, $\|\tau' - \tau_*\|_2$ entails $b = \Theta(R)$, which is a contradiction. Hence it must be that $\|\tau' - \tau_*\|_2 > 0.15R$. $\qquad\square$

*Proof of Theorem 4.2.* By Lemma D.7, the optimization path will go to a point $\tau' \in \mathcal{A}$ s.t. $\|\tau' - \tau_*\|_2 > 0.15R$. From any such $\tau'$, the shortest way to exit the annulus $\mathcal{A}$ is to project onto the inner circle defining $\mathcal{A}$, i.e. the circle centered at $\tau_*$ with radius $0.1R$ which is a convex set. Denote this inner circle as $\mathcal{B}(\tau_*, 0.1R)$ whose projection is $\Pi_{\mathcal{B}(\tau_*, 0.1R)}$, then the shortest path is the line segment $\tau' - \Pi_{\mathcal{B}(\tau_*, 0.1R)}(\tau')$. Further, this line segment is of length $0.05R$ since $\|\tau' - \tau_*\| > 0.15R$ by Lemma D.7.

However, the decrease of the parameter distance (i.e. $\|\tau - \tau_*\|$) is exponentially small at any point in $\mathcal{A}$:

$$\|\tau_t - \tau_*\| - \|\tau_{t+1} - \tau_*\| \overset{(i)}{\leq} \frac{\eta}{\sigma_{\min}(\nabla^2 L(\tau_t))} \left| \left\langle \nabla L(\tau_t), \frac{\tau_t - \tau_*}{\|\tau_t - \tau_*\|_2} \right\rangle \right|$$

$$\overset{(ii)}{\leq} \frac{\left| \left\langle \nabla L(\tau_t), \frac{\tau_t - \tau_*}{\|\tau_t - \tau_*\|_2} \right\rangle \right|}{\lambda_M} \overset{(iii)}{\leq} O\left( \frac{\exp\left(-\frac{\kappa(b,c)R^2}{8}\right)}{R^3} \right) \tag{D.53}$$

where step $(i)$ uses the calculations in equation D.51; step $(ii)$ use the choice of $\eta$; and step $(iii)$ uses Lemma D.1.

Hence the number of steps to exit $\mathcal{A}$ is lower bounded by $\frac{0.05R}{O(\frac{2}{R^2} \exp(-\frac{R^2}{8}))} = \Omega\left( R^3 \exp\left(\frac{R^2}{8}\right) \right)$.

$\qquad\square$

# E    ADDITIONAL NOTES ON EXPERIMENTS

## E.1    IMPLEMENTATION DETAILS

**Parameterization**: For the 1-dimensional Gaussian, we take $P_*, Q$ to have mean $\mu_* = 16, \mu_q = 0$, and unit variance $\sigma_*^2 = \sigma_q^2 = 1$; see Figure 4a for an illustration of the flat loss landscape. We

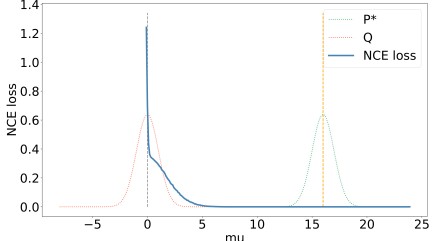
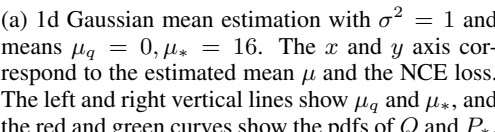
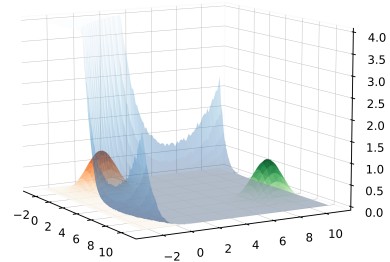

(a) 1d Gaussian mean estimation with $\sigma^2 = 1$ and means $\mu_q = 0, \mu_* = 16$. The $x$ and $y$ axis correspond to the estimated mean $\mu$ and the NCE loss. The left and right vertical lines show $\mu_q$ and $\mu_*$, and the red and green curves show the pdfs of $Q$ and $P_*$.

(b) 2d Gaussian mean estimation with $\sigma^2 = 1$ and means $\mu_q = 0, \mu_* = 8$. The blue surface shows the NCE loss surface, and the orange and green surfaces show the pdfs of $Q$ and $P_*$.

Figure 4: An illustration of the flat landscape caused by the "density chasm" NCE loss quickly flattens out for 1d and 2d Gaussian mean estimation.

use $h(x) := \exp(-\frac{x^2}{2})$, $T(x) := [x, -1]$ to be consistent with the notation in Section 4. For the 16-dimensional Gaussian, $P_*, Q$ share the same mean $\mu_* = \mu_q = 0$ but have different covariance with $\mathrm{Cov}_q = \boldsymbol{I}_d$ and $\mathrm{Cov}_p = \mathrm{diag}([s_1, ..., s_d])$, where $s_i = \mathrm{Uniform}[8 \times 0.75, 8 \times 1.5]$. [8]

For MNIST, we adapt the TRE implementation by Rhodes et al. (2020). We model the log density ratio $\log(p/q)$ by a quadratic of the form $g(x) := -f(x)^\top \boldsymbol{W} f(x) - \boldsymbol{b}^\top f(x) - c$, where $f$ is ResNet-18, and $\boldsymbol{W}, \boldsymbol{b}, c$ are trainable parameters with $\boldsymbol{W}$ constrained to be positive definite.

**Implementation notes**: We include some tricks we found useful for implementation:

- Calculation in log space: instead of dividing two pdfs, we found it more numerically stable to use subtraction between the log pdfs and then exponentiate.

- Removing common additive factors: the empirical loss is the average loss over a batch of samples where overflow can happen. [9] We found it more stable to calculate the mean by first subtract the largest value of the batch, calculate the mean of the remaining values, then add back the large value—akin to the usual log-sum-exp trick. For example, $\mathrm{mean}([a, b]) = \max(a, b) + \mathrm{mean}([a - \max(a, b), b - \max(a, b)])$.

- Per-sample gradient clipping: it is sometimes helpful to limit the amount of gradient contributed by any data point in a batch. We ensure this by limiting the norm of the gradient, that is, the gradient from a sample $x$ is now $\min\{1, \frac{K}{\|\nabla \ell(x)\|}\} \nabla \ell(x)$ for some prespecified constant $K$ (Tsai et al., 2021).

- Per-sample log ratio clipping: an alternative to per-sample gradient clipping is to upper threshold the absolute value of the log density ratio on each sample, before passing it to the loss function. Setting a proper threshold prevents the loss from growing too large, and consequently prevents a large gradient update.

### E.2 ADDITIONAL RESULTS

**Results for training with a larger computation budget**: We provide additional results on Gaussian mean estimation and MNIST, both trained with a larger computation budget.

Figure 5 shows results similar to those of Figure 1, except that we now run the optimization process for 5 times longer than in Figure 1, and additionally show results on eNCE optimized with gradient descent (GD). The conclusion is the same as that of Figure 1: for both NCE and eNCE , normalized

---

[8]Generally, for $d$-dimensional Gaussian with mean $\mu$ and a diagonal covariance matrix $\Sigma := \mathrm{diag}([\sigma_1^2, ..., \sigma_d^2])$, the exponential parametrization is $\tau = [\frac{1}{\sigma_1^2}, ..., \frac{1}{\sigma_d^2}, \frac{\mu_1}{\sigma_1^2}, ... \frac{\mu_d}{\sigma_d^2}, \frac{\mu^\top \Sigma^{-1} \mu}{2} + \frac{1}{2}\log((2\pi)^d \det(\Sigma))]$.

[9]This is because the mean function is internally implemented as the sum of all entries divided by the batch size, and the sum of a large batch size where each value is also large can lead to overflow.

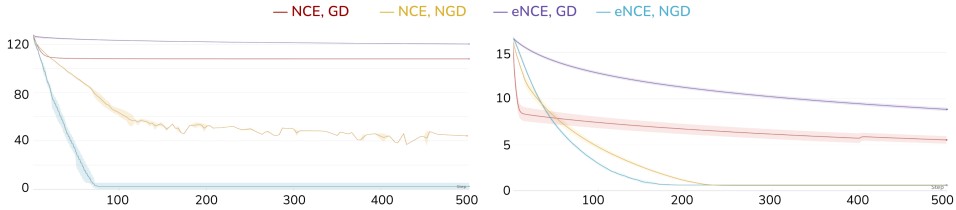

Figure 5: Results for estimating 1d (left) and 16d (right) Gaussians, plotting $\min_{t \in [T]} \|\tau_* - \tau_t\|_2$ ($y$-axis) against the number of updates $T$ ($x$-axis). Normalized gradient descent (NGD) significantly outperforms vanilla gradient descent (GD) for both NCE and eNCE . In addition, eNCE decays faster than NCE when optimized with NGD. The results are averaged over 5 runs, with shaded areas showing the standard deviation.

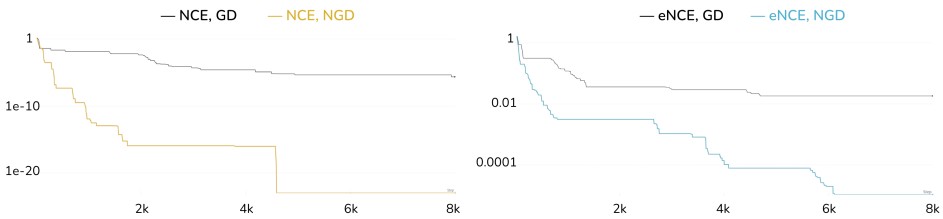

Figure 6: Results on MNIST, plotting loss value ($y$-axis, log scale) against the number of update steps ($x$-axis). The left plot shows NCE optimized by GD (black) and NGD (yellow), and the right shows eNCE optimized by GD (black) and NGD (blue). The setup is the same as that for Figure 2 except that we now let training run 4 times longer. NGD outperforms GD in both cases, consistent with the results in Figure 2.

gradient descent (NGD) significantly outperforms GD. Moreover, eNCE performs competitively compared to NCE when optimized with NGD.

Similarly, we train with a large computation budget on MNIST, whose results are shown in Figure 6. The results are again consistent with those in Figure 2.

**MNIST samples** We run annealed importance sampling (AIS) Neal (2001) following Rhodes et al. (2020) on the models trained on NCE and eNCE , optimized with GD or NGD. Figure 7, 8 show samples generated with 4k or 10k sampling steps, from different random initialization. We can see that eNCE gives much sharper results than NCE, and eNCE with NGD results in more diverse samples. A downside though is that eNCE samples seem to show signs of mode collapse. However, it is unclear whether this is a problem with the model or due to the sampling procedure.

**Results for training with other normalized optimization method**: One interesting question to ask is, whether the results of NGD generalize to other optimizers that perform some form of normalization. One example that is commonly used in practice is the RMSprop, which performs per-coordinate normalization on the gradient. Specifically, at the $t_{th}$ step with gradient $g_t$, RMSprop first updates a cumulative term $\boldsymbol{v}_t := \alpha \boldsymbol{v}_{t-1} + (1 - \alpha)g_t^2$, where $\alpha \in [0, 1]$ is a hyperparameter

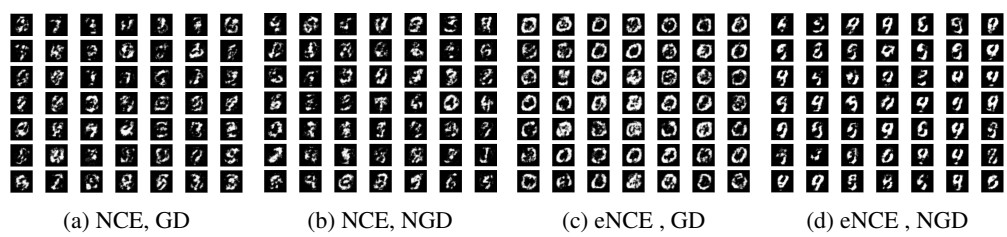

    (a) NCE, GD          (b) NCE, NGD         (c) eNCE , GD        (d) eNCE , NGD

Figure 7: MNIST samples from 4000 sampling steps

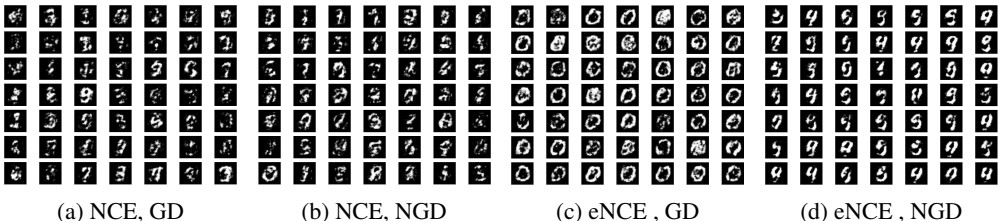

    (a) NCE, GD           (b) NCE, NGD           (c) eNCE , GD        (d) eNCE , NGD

Figure 8: MNIST samples from 10000 sampling steps

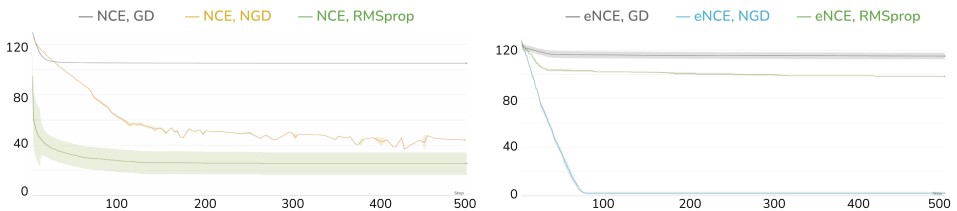

Figure 9: Results for estimating 1d Gaussian with NCE (left) or eNCE (right), using GD (gray), NGD (yellow for NCE, blue for eNCE ), or RMSprop (green). The effectiveness of RMSprop seems to depend on the task: RMSprop performs the best for NCE, but falls short than NGD for eNCE.

controlling how much to "damp" the current gradient, and the square on $g_t$ is applied entrywise. It then normalizes the gradient as $\tilde{g}_t := g_t / v_t$, with division applied entry-wise.

Figure 9 compares RMSprop with GD or NGD on 1d Gaussian estimation task. The results suggest that how well RMSprop performs may be task-specific: RMSprop performs the best when optimizing for NCE, but only slightly better than GD when optimizing for eNCE . Moreover, NCE favors a higher value of $\alpha$ where the best performance is achieved by $\alpha = 0.99$, whereas eNCE prefers $\alpha$ to be small with $\alpha = 0.01$ performing the best.

It is not yet clear what the theoretical answer should be for normalized methods in general: though methods like RMSprop perform certain form of normalization (e.g. per-coordinate normalization), so does Newton's method, and Theorem 4.2 has shown that it still suffers from an exponentially bad convergence rate (at least with standard choices of step size). It is unclear what quantity replaces the condition number of the Hessian for RMSProp, which governs the convergence of NGD. Theoretical guarantees for these optimization methods is an interesting open question.

