# OpenReview forum: "Analyzing and Improving the Optimization Landscape of Noise-Contrastive Estimation"
_ICLR.cc/2022/Conference — ICLR 2022 Spotlight_

### Official Review · Reviewer_6syu · 2021-10-27

**Correctness:** 3
**Technical Novelty And Significance:** 2
**Empirical Novelty And Significance:** 2
**Recommendation:** 6
**Confidence:** 3

**Main Review:**

Pros:
The author constructs a bad case, and improves the NCE by fixing problems in the bad case. The improvement is theoretically guaranteed.

Questions:
1. The Proposition 4.2 is confusing. It states that $||\tau - \tau^*|| \leq \delta \Rightarrow L(\tau) - L(\tau^*) = R \exp(-R^2 / 8) \delta^2$. However, according to it, we can also derive $||\tau - \tau^*|| \leq \delta \Rightarrow  ||\tau - \tau^*|| \leq (\delta+1) / 2 \Rightarrow L(\tau) - L(\tau^*) = R \exp(-R^2 / 8) ( (\delta+1) / 2)^2$. Thereby, we have $||\tau - \tau^*|| \leq \delta \Rightarrow R \exp(-R^2 / 8) \delta^2 = R \exp(-R^2 / 8) ( (\delta+1) / 2)^2 \Rightarrow \delta = (\delta+1) / 2 \Rightarrow \delta = 1$, which is a contradiction. The author can use quantifiers like $\exists$, $\forall$ to make Proposition 4.2 more clear.

2. In Section 4, the  $R$ appears before it is first defined. Besides, above Section 4.1, the author defines $\tau(\theta) = [\theta, \frac{\theta^2}{2}+\log \sqrt{2\pi}]$, but then says "In particlular, $\tau(\theta_*) = [R, -\frac{R^2}{2}-\log \sqrt{2\pi}]$". It seems that there is a contradiction between $\tau(\theta)$ and $\tau(\theta_*)$.

3. Although the improvement can solve problems in the Gaussian case, it is not guaranteed that the improvement generalizes well on more complex data sets. I notice that the author provides results on MNIST and the loss curve becomes better. It would be better if more experimental results on more complex data sets are provides. For example, the author can show some generated samples of MNIST or try on datasets like CIFAR10 and provides FID[1*] results.

4. The author can plot a figure of the landscape of loss in the 1-d Gaussian example. It helps readers understand the illness of the landscape better.

5. The uniqueness in Lemma 2.1 needs $p_*(x) \neq 0 \rightarrow q(x) \neq 0$ to hold.

[1*] https://github.com/bioinf-jku/TTUR

**Summary Of The Paper:**

This paper constructs a Gaussian case to illustrate why NCE doesn't work. The failure in this case is due to the flat loss landscape. The traditional GD or Newton's methods don't work on such a flat landscape. To overcome this problem, the author proposes a normalized GD, which improves the convergence rate. The author also introduces a modification of the NCE loss.

**Summary Of The Review:**

This paper is generally solid, although there is some confusion.

---

> ### Author Response · Authors · 2021-11-13
> **Thank you for your feedback!**
>
> We thank the reviewer for their thoughtful questions.
>
> **Question 1 and 2**: We thank the reviewers for pointing out the typos in Proposition 4.2 and Section 4. We just want to point out the changes are minor and we have updated them in the submission.
> To clarify, we made the following changes:
>   - For Proposition 4.2, the statement should be $L(\tau) - L(\tau_*) \leq R\exp(-R^2/8)\delta^2$, i.e. the equality should be changed to inequality, where the inequality can be saturated by taking $\tau - \tau_*$ to be aligned with the direction of the eigenvector for the max eigenvalue.
>   - For the definition of $\tau_*$, it should be $\tau_* = [R, \frac{R^2}{2} + \log\sqrt{2\pi}]$, i.e. in the previous version we had flipped the sign of the second coordinate. We have also rearranged the text to have $R$ explained before defining $\tau_*$.
>
> **Question 3**: We thank the reviewer for the constructive feedback. While we agree that showing results on real-world datasets would make the paper stronger, we would like to emphasize that the main contribution of our work is a theoretical understanding of the NCE objective.
> We did not include samples for MNIST/CIFAR because of two reasons: 1) generating reasonable samples even for MNIST generally takes large EBMs (e.g. MNIST already requires ResNet18) which is computationally expensive,
> and 2) the samples themselves may not always truthfully reflect the quality of the learned model, since the sample quality will be largely affected by the sampling algorithm, which is not the focus of our work.
>
> **Question 4**: Thank you for the suggestion, we have added a plot (Figure 4) in Section E of the appendix to illustrate the flatness of the landscape for 1d Gaussian mean estimation.
>
> **Question 5**: We thank the reviewer for pointing this out and have clarified this in the updated version.
>
> We hope our response addresses your concerns, and would appreciate it if you could consider raising the score. Thank you!

---

> > ### Comment · Reviewer_6syu · 2021-11-17
> > **Thanks for your reply**
> >
> > I'm satisfied with the landscape plot and I suggest the author to put it to the main paper if there is enough space, since it directly visualize the illness. Besides, I still suggest the authors to put some samples in the paper, since it is interesting to see samples learned via NCE. On MNIST, based on my experiences, it takes about several minutes to generate 100 samples, on a single GPU.
> >
> > Overall, I'm satisfied with the reply and I've increased my score to 6.

---

> > > ### Author Response · Authors · 2021-11-17
> > > **Thank you for your response!**
> > >
> > > We are glad that the reviewer found our reply satisfying, and thank you very much for raising the score!
> > > Thank you also for the suggestion of adding samples; we will work on this and add the results to the draft when they are ready.

---

> > > > ### Author Response · Authors · 2021-11-22
> > > > **MNIST samples added**
> > > >
> > > > Dear reviewer,
> > > >
> > > > Please note that we added some MNIST samples generated from NCE and eNCE (optimized with GD and NGD) in Appendix E.2 of the revised draft. It can be seen that eNCE gives sharper results, and NGD helps encourage diversity in samples. Thank you very much again for the suggestion!

---

### Official Review · Reviewer_VmJ3 · 2021-10-30

**Correctness:** 4
**Technical Novelty And Significance:** 4
**Empirical Novelty And Significance:** Not applicable
**Recommendation:** 8
**Confidence:** 3

**Main Review:**

**Caveats:**
- I am not so familiar with the literature in this area, so cannot determine the novelty of the results or whether there are missing citations.
- I did not read the proofs in detail, so cannot vouch for the correctness of the mathematical results.


**Summary:**

I think this is a good paper and I am in favour of its acceptance. I do have a few points of concern / thoughts on how the paper can be improved, but I hope the authors will take these into account.

I think it is great when theoreticians analyse settings that are motivated by empiricists, as I believe that this synergy can lead to improvements in understanding of things that matter. This is an interesting and seemingly poorly-studied area. NCE and the related GANs are very widely used, and so improving understanding of them is important.

The paper is well written and explains the problem and their solution clearly. I didn't spot any technical issues, and I believe they make a solid contribution. By exploring the simplest possible empirical setting (1D Gaussians) they clearly highlight the issues with existing methods, but prove results that are much more general.

Therefore, although there are some areas where things could be improved -- a sober explanation of the downsides of their assumptions, and where the connection to the empirical world might break down, as well as the experimental validation -- I am overall happy with the paper.

Please see the detailed comments below for all questions / wishes for the authors. Here I would like to highlight the main points of concern:

- Discussion of the assumptions made -- under what conditions they hold and do not hold, perhaps with examples to give an intuition as to whether they are strong or weak assumptions.
- Discussion of the gap between the theoretical setting they study, and the ones encountered in the 'real world'. For example: dimension, families of distributions, mini-batching vs full gradient descent.
- Experiments: please provide code to make things easier for future researchers to build on your work.


I am giving the paper a "6: marginally above the acceptance threshold", but would be willing to raise my score if the authors can address my concerns.




**Detailed comments:**


Page 3:
- Exponential family: could you define the partition function and add a short comment on why this (or the log-partition-fn) is so important?
- Assumptions 2.1 -> 2.4. Could you give examples of distributions that satisfy and violate these assumptions, to give an intuition of how strong/weak the assumptions are?


Page 7:
- Definition of eNCE. This is perhaps more of a philosophical question, and is also a stretch outside of my area of expertise, so it is possible that I am confused and talking rubbish, if so please ignore me (though I would appreciate an explanation of where I am wrong!)

  As I understand it, the NCE loss can be viewed as an upper bound on the Jensen-Shannon divergence between P* and Q, i.e. if you minimise the loss wrt \theta, the resulting minimum is exactly JS(P*, Q) [possibly with some sign flips and min <-> max].

  Jensen-Shannon is an instance of an f-divergence. It looks to me like the same as the above could be said of the eNCE loss, but just with a different choice of f-divergence. Do you think that could be the case?

  If so, that would be interesting connection to draw because there has been work in the generative modelling (GANs, VAEs etc) community on using different choices of f-divergences other than JS to overcome some limitations of JS. Of the top of my head I remember this 'canonical' paper, but I'm sure there are many other works, also probably some more theory oriented ones: https://arxiv.org/abs/1606.00709 (f-GAN, by Nowozin et al 2016)


- Lemmas 6.1-2 and Theorem 6.1. There are no free lunches, so I guess there must be some downside of using the eNCE objective. I would suspect that the exponential (rather than log) loss would lead to numerical instability, or very large gradient variances when using mini-batch optimisation methods. Could you comment on this? Also, what is the impact of dimension on your results? It doesn't appear explicitly, but perhaps comes into play via \lambda_min ?


Page 9:
Experiments:
- Figure 1: what is meant by "best parameter distance"? I didn't quite understand the explanation in the Results paragraph.
- Figure 1: Perhaps you could consider extending the plots beyond 100 steps, to see how the worse methods perform after a larger number of steps? (You could consider making the x-axis be log-scale.)

Figure 2:
- Qn1: Are you doing full gradient descent on MNIST? (i.e. without minibatching?)
- Qn2: Rather than comparing to the baseline of a fixed 2k budget, why not let all the methods run for a long time and see where they converge, and use that as the baseline? [It doesn't say how long a run of 2k steps takes, but I imagine not so long?]

- Could you comment on the full gradient descent assumed in the paper/experiments, vs the mini-batch based methods commonly used by practitioners?
- I'd also be interested to see how the experiments behave if you repeat them just naively using mini-batches of different sizes.
- Please make code for the experiments/methods available -- this is the best way for readers to check if they are not sure that they understand something with the implementation.


Minor:
- Page 3: the spacing seems to be a little messed up here, e.g. in lines 2 and 3 of assumption 2.3 the equations intersect a little bit.






**Summary Of The Paper:**

TLDR: The authors theoretically study convergence properties of the Noise Contrastive Estimation (NCE) loss, identify issues that make its optimisation difficult in some practical scenarios, propose a method to resolve this and provide theoretical guarantees that their proposal works.

Longer version:

The authors point out that although NCE in theory should work regardless of the choice of noise distribution Q (assuming population distributions and ignoring optimisation etc), practitioners have empirically found that the choice of Q is crucial to methods working well.

There is not a great deal of work theoretically analysing the NCE objective, and so the reasons for why choice of Q is so important is not rigorously understood.

In this paper they study the setting in which all distributions are in the exponential family of distributions.

- They first show that in the very simple case of 1d Gaussian distributions, if the means are far apart then the number of gradient updates required to make the distributions close grows exponentially in the initial distance (because the norm of the update steps decay exponentially in this distance)
- They next show that, provided the distributions are 'sufficiently close', using *normalised* gradient descent results in requiring only polynomially many steps.
- Next, they show that modifying the NCE loss (exponential-NCE) results in such polynomial guarantees even without the distributions being 'sufficiently close'
- Finally they validate their theory with basic experiments.



**Summary Of The Review:**


I think this is a good paper and I am in favour of its acceptance. I do have a few points of concern / thoughts on how the paper can be improved, but I hope the authors will take these into account.

As such I have set my recommendation to "6: marginally above the acceptance threshold", but will increase my rating if the authors are able to satisfactorily address my comments.

[Edit: updated my score to "8: accept, good paper" after author response]

---

> ### Author Response · Authors · 2021-11-13
> **Thank you for your feedback! [re: main concerns]**
>
> We thank the reviewer for their very detailed and constructive feedback.
>
> We would first like to address the main concerns.
>
> **Discussion of the assumptions made**: Thank you for the question. We will comment on the assumption that $p_{\theta}$ is an exponential family in the following section ("Discussing the gap between the theoretical setting and practice".) Assumptions 2.1-2.4 can in fact be viewed as introducing structural parameters of the distributions.
> For example, distributions with flatter tails will have a larger $\lambda_{\max}$, which then translates to a slower rate in the results; distributions that are closer to being singular with have a smaller $\lambda_{\min}$, etc.
> To get a sense of the scaling of the constants, consider a $k$-dimensional Gaussian with unknown mean and variance. Then $\beta_Z = O(\omega)$ (where $\omega$ is the norm bound on the parameters introduced in Assumption 2.1), $\lambda_{\max} = O(\omega^2)$, and $\gamma_{\max} = O(\omega)$.
> $\lambda_{\min}$ and $\gamma_{\min}$ correspond to how close to singular the covariance matrix of the Gaussian is allowed to be.
> Note, the norm bound in Assumption 2.1 can be viewed as bounding the optimization domain, and is commonly required in order for the performance on the training set to generalize.
> We have updated the paper to include some comments on these assumptions, thank you for the suggestion!
>
> **Discussion of the gap between the theoretical setting and the practice**:
> This is also a great question. We will speak to all the aspects you mentioned.
>
> *Dimension dependency*: The dimension dependence is implicitly absorbed in the class-dependent constants introduced in Assumptions 2.1 - 2.4. For instance, for a Gaussian with mean $\mu \in \mathbb{R}^d$ and variance $\sigma^2\mathbb{I}_d$, the log partition function is $\log Z([\mu, \sigma^2]) = \frac{\|\mu\|^2}{2\sigma^2} + \frac{d}{2}\log{2\pi\sigma^2}$, which grows linearly in $d$.
> In general, we expect the logarithm of the partition function to scale linearly with the dimension $d$ (while the partition function, without the log, can scale exponentially with the dimension as it's a $d$-dimensional integral).
>
> *Family of distributions*: Theoretically, exponential families, for a sufficiently rich family of sufficient statistics $T$ (e.g. polynomials of a sufficiently large degree), are universal distribution approximators. For high-dimensional probability distributions, having a rich class $T$ can result in a very high-dimensional vector of parameters $\theta$ (e.g. the number of polynomials of degree $m$ over $d$ variables is roughly $d^m$). As a result, alternate parametrizations (e.g. via a judiciously engineered neural network) can perform better. This mirrors the difference between using kernels and neural networks for standard supervised learning.
>
> *Mini-batch vs full gradient descent*: We used mini-batches of size 250 rather than full gradient descent, so there is indeed a gap between our theoretical results and the experiments.
> The reason that we are studying full gradient descent is to decouple the statistical aspect and the algorithmic aspect of the optimization problem: with access to the full gradient, we can be sure that any challenge to optimization is caused by the loss landscape itself, rather than noises from using finite samples.
> It would be interesting to study the finite sample behavior of normalized gradient descent using NCE or eNCE. Though we currently do not have anything formal in the paper on this, we agree with the reviewer that eNCE would likely to require a larger batch than NCE since eNCE gradients should have a larger variance.
>
> **Experiments**: We plan to release the code, thank you for the suggestion!

---

> > ### Author Response · Authors · 2021-11-13
> > **[re: detailed comments]**
> >
> > We next respond to the detailed comments below.
> >
> > **Partition function of exponential family**: The normalization constant of an unnormalized density is often referred to as its partition function; we have added a footnote (footnote 1 on page 1) to explicitly define this terminology.
> >
> > **Relation to $f$-divergence**: You are exactly right about this: there is a notion of "generalized NCE" proposed in prior work (Pihlaja et al., 2012; Gutmann and Hirayama, 2012; Uehara et al., 2020), which relates the NCE objective to minimizing the Bregman divergence. More specifically, generalized NCE says that we can design a family of training objectives by using different convex functions to define the Bregman divergence, and the proposed eNCE is an instance of this generalized NCE objective.
> > The difference between these prior work and ours is again the different focuses on asymptotic behavior versus finite step convergence.
> > We have updated the related work section to include this discussion.
> >
> > **Results on eNCE (Lemma 6.1-2, Theorem 6.1)**: As the reviewer correctly pointed out, eNCE tends to be more numerically unstable, since the gradients tend to be larger in magnitude. We mentioned this at the end of the experiment section, and provided notes on implementation in Section E in the appendix.
> >
> > **Figure 1**: The "best parameter distance" at step $T$ is defined as $\min_{t \leq T}||\tau_t - \tau_\star||_2$. The reason for using this measure is that for normalized gradient descent, the parameter distance $||\tau_t - \tau_\star||_2$ is not guaranteed to decrease monotonically. (Such behavior also happens in practice, and is not just an artifact of the analysis.)
> >
> > **Running the experiments for longer**:
> > We train on the Gaussian mean estimation task for 5 times longer than Figure 1 in the paper, and additionally include the results for eNCE trained with GD. The results are shown in Figure 5 in Appendix E.
> > Consistent with Figure 1, we see that NGD significantly outperforms GD on both NCE and eNCE loss, and eNCE loss outperforms NCE when using NGD. The gap is especially clear for 1d Gaussian with well-separated means: even when using NGD, NCE struggles to get close to the ground truth parameter, because the landscape has become so flat that it may run into numerical underflow.
> > We similarly run experiments on MNIST for longer (Figure 6 in Appendix E), whose results again show that NGD outperforms GD, consistent with Figure 2.
> >
> > Please let us know if you have additional questions or concerns, thank you!
> >
> > *References*:
> >   - Miika Pihlaja, Michael Gutmann, and Aapo Hyvärinen.  A family of computationally efficient andsimple estimators for unnormalized statistical models (UAI 2010).
> >   - Michael Gutmann and Jun-ichiro Hirayama. Bregman divergence as general framework to estimate unnormalized statistical models (UAI 2011).
> >   - Masatoshi Uehara, Takafumi Kanamori, Takashi Takenouchi, and Takeru Matsuda. A unified statis-tically efficient estimation framework for unnormalized models (AISTATS 2020).

---

> > > ### Comment · Reviewer_VmJ3 · 2021-11-22
> > > **I'm happy**
> > >
> > > Thanks very much for the detailed response! I'm satisfied that the authors have addressed the points that I raised, and so I will increase my score as promised.

---

### Official Review · Reviewer_rZGo · 2021-11-02

**Correctness:** 4
**Technical Novelty And Significance:** 3
**Empirical Novelty And Significance:** 3
**Recommendation:** 6
**Confidence:** 4

**Main Review:**

Strength
-	Theoretically understanding of NCE method is an important direction and I believe this paper gives some interesting results in this direction.
-	This paper gives a concrete reason and example to explain why NCE can fail in practice in certain situation. Authors further give several ways to provably overcome the issue, which is quite interesting.
-	The paper is overall clearly written and easy to follow.

Weakness:
-	The paper only studies the NCE method for the exponential family of probability distribution, which does not include the more complicated distributions that might be used in practice. However, I would not view this as a major weakness, as the current results already seems to be interesting to me.

Some minor comments:
-	At the beginning of page 4 (Overcoming flatness using normalized gradient descent): use delta without explanation (which I believe is the distance to ground truth according to Theorem 5.1)
-	Proposition 4.2: missing an O(1) factor in front of R exp(R^2/8)\delta^2 ?


**Summary Of The Paper:**

This paper studies the noise-contrastive estimation (NCE) method for the exponential family of probability distribution, which is usually used to learn probabilistic models. The authors first give a negative result in the setting of Gaussian mean estimation, which shows that NCE suffers from the ill-behaved population landscape when the proposed distribution is far away from the target distribution. As a consequence, standard gradient descent and Newton’s method need to run exponential time for convergence. To overcome this issue, authors first show that normalize gradient descent (NGD) can have polynomial convergence guarantee under certain condition. Furthermore, a variant of NCE called eNCE was proposed so that NGD can converge to optima within polynomial time. Experiments are also provided to show that using eNCE with NGD can have comparable performance with NCE.

**Summary Of The Review:**

Based on the above comment, I believe this paper gives interesting theoretical results on NCE and would vote for accept.

---

> ### Author Response · Authors · 2021-11-13
> **Thank you for your feedback!**
>
> We thank the reviewer for their constructive feedback and for voting in favor of acceptance.
>
> **Limitation of exponential family**: We would like to point out that at least from the point of view of expressivity, exponential families do not pose a restriction. Namely, with a rich enough family of sufficient statistics (e.g. polynomials of a sufficiently large degree), exponential families are universal distribution approximators.
> Having a more expressive family may require fitting a larger number of parameters (e.g. the number of polynomials of degree $m$ over $d$ variables is roughly $d^m$).
> However, working with the exponential family is still desirable since it enjoys the benefit of a convex objective. This mirrors the trade-offs between using kernels and neural networks for standard supervised learning.
>
> We would also like to thank the reviewer for pointing out the typos, which we have fixed in the updated version. Since the reviewer seems to have appreciated the paper's insights into the NCE loss landscape, we hope the reviewer would consider raising their score. Thank you!

---

### Official Review · Reviewer_yXSX · 2021-11-09

**Correctness:** 4
**Technical Novelty And Significance:** 4
**Empirical Novelty And Significance:** 3
**Recommendation:** 8
**Confidence:** 2

**Main Review:**

The paper has many strengths. It takes a step back to ask a fundamental question in the Contrastive Learning (what happens when the discrimination task is too easy) and Generative Modelling literature (what happens when the data and noise are too far from each other - this is a motivation behind current advances in diffusion models).

It deals with it effectively while proving an analytical characterisation on a simple yet generic use-case (exponential family distributions). For another, related Contrastive Loss (InfoNCE), intuitions that the gradient norm is related to the hardness of the task are explored in Khosla et al 2021, yet a complete formalization is not provided. Furthermore, Pihlaja et al 2012 consider modifications around the NCE loss and comment on the effect (e.g. stable gradients) on the corresponding estimation tasks, yet a conclusive recommendation is not reached. This paper makes a convincing case on how to modify the loss with algorithmic considerations in mind.

Out of curiosity:

1. Is it possible to empirically compare NGD with other normalized-gradient methods?
Because a central contribution is "normalizing the gradient", would it be worth comparing the authors' Normalized Gradient Descent (NGD) with other normalized-gradient methods (Adagrad, Rmsprop, Adam) from recent years in the Stochastic Optimization literature? Or with other such methods?

2. What is the impact of the proportion of noise samples hyperparameter on this paper's analysis?
The NCE task in this paper has fixed the proportion of noise samples at 50%, as is common in the literature. Do the authors know how this hyperparameter might impact the loss landscape and their results?


**Summary Of The Paper:**

This paper examines Noise-Contrastive Estimation (NCE) from an optimization perspective.

The NCE task consists in training a Discriminator to distinguish between data and noise samples. Prior works observe that this task is "too easy" when data and noise distributions are "very distinguishable", which leads to poor estimation. This paper provides a formal explanation for this - the loss landscape becomes flat - along with an analytical formula for a simple model (1D mean-parameterized Gaussians).

To remedy this problem, previous literature has sought to modify the task itself, by breaking it down into a sequence of harder subtasks (GANs, Flow-Contrastive Estimation, Telescoped Density Ratio). This paper proposes to keep the task at hand, but enable the optimization to cope with the difficulties of a flatter landscape. It does so by:

1. Modifying the optimization algorithm: normalize the gradient at every step
2. Modifying the loss: still density-ratio based, but replace the sigmoid-log with a square root

These two contributions are supported theoretically, using the exponential distribution as a study case. Respectively,
1. Provides a polynomial convergence rate in the parameter distance (i.e. how different the data and noise are) and Hessian conditioning number (i.e. local curvature near optimum)
2. Expresses the Hessian conditioning number as polynomial in the parameter distance and dimension, to complete the convergence rate characterization

The effect of either modification is empirically verified on the MNIST dataset.

**Summary Of The Review:**

I have not read the proofs in detail. This optimization angle on NCE seems rather new and the overall expose seems convincing and beneficial to the community.

---

> ### Author Response · Authors · 2021-11-13
> **Thank you for your feedback!**
>
> We appreciate the reviewer for their encouraging feedback!
>
> **Comparison of NGD with other normalized-gradient methods**: This is an interesting question. We do not have an immediate theoretical answer to this: though methods like RMSprop perform certain form of normalization (e.g. per-coordinate normalization), so does Newton's method, and we have shown that it suffers from an exponentially bad convergence rate (at least with standard choices of step size). It's unclear what quantity replaces the condition number of the Hessian for RMSProp, which governs the convergence of NGD.
> Empirically, we will run the experiments with RMSProp and report the results when they are ready.
>
> **The impact of the proportion of noise samples**:
> Following equation (A.2), we can decompose the gradient and Hessian into two terms, one from the data samples, one from the noise samples.
> Denote by $\lambda$ the ratio between the number of noise samples and the number of data samples, then we have:
>
> $2\nabla L(\tau) = -\frac{1}{1+\lambda} \mathbb{E}_\star[\frac{q}{p+q}T(x)] + \frac{\lambda}{1+\lambda}\mathbb{E}_Q[\frac{p}{p+q}T(x)]= \int_x \frac{q}{p+q} (\frac{\lambda}{1+\lambda} p - \frac{1}{1+\lambda}p_\star) T(x) dx$
>
> $2\nabla^2 L(\tau) = \frac{1}{1+\lambda}\mathbb{E}_\star[\frac{pq}{(p+q)^2} T(x)T(x)^\top] + \frac{\lambda}{1+\lambda} \mathbb{E}_Q[\frac{pq}{(p+q)^2} T(x)T(x)^\top] = \int_x \frac{pq}{(p+q)^2} (\frac{1}{1+\lambda} p_\star + \frac{\lambda}{1+\lambda} q) TT^\top$
>
> Given this terminology, the paper presents our results for $\lambda = 1$. For $\lambda \neq 1$, these results still hold up to some scaling factor.
> For example, Theorem 4.1 still holds since its key technical lemma, i.e. Lemma D.1, still holds: the term $p-p_*$ in equation (D.5) will be updated to $\frac{\lambda}{1+\lambda} p - \frac{1}{1+\lambda} p_*$, and the subsequent calculations (e.g. equation D.6) will have terms scaled accordingly, which will be absorbed into the $O(1)$ term in Lemma D.1.

---

> > ### Author Response · Authors · 2021-11-22
> > **RMSprop results added**
> >
> > Dear reviewer,
> >
> > Please note that we added some results comparing RMSprop with GD and NGD on 1d Gaussian mean estimation, which are included in Appendix E.2 of the revised draft.
> > We found that RMSprop performs better than NGD for the NCE loss, but worse than NGD for the eNCE loss. These results suggest that it is unclear how generic normalized optimization methods would perform, and it is an interesting open direction to provide theoretical guarantees on them. Thank you very much again for this interesting question!

---

> > > ### Comment · Reviewer_yXSX · 2021-11-22
> > > **RMSprop results added  [answer]**
> > >
> > > Thank you for adding these results! They provide an interesting comparison.

---

### Decision · Program_Chairs · 2022-01-20

**Decision:**

Accept (Spotlight)

**Comment:**

This contribution investigates and takes a step back on an important problem in recent ML, namely the impact of the noise distribution in density estimation using Noise Contrastive Estimation. The work offers both theoretical insights and convincing experiments.

For these reasons, this work should be endorsed for publication at ICLR 2022.